# Mass balances of Yala and Rikha Samba glaciers, Nepal, from 2000 to 2017

Dorothea Stumm[1,2], Sharad Prasad Joshi[2], Tika Ram Gurung[2], Gunjan Silwal[2,3]

[1]Independent consultant, 8184 Bachenbülach, Switzerland
[2]ICIMOD, G.P.O. Box 3226, Kathmandu, 44600, Nepal
[3]Department of Earth and Atmospheric Sciences, University of Alberta, Edmonton, Alberta, T6G 2E3, Canada

*Correspondence to*: Dorothea Stumm (stummd@gmail.com)

**Abstract.** The glacier mass balance is an important variable to describe the climate system, and is used for various applications like water resource management or runoff modelling. The direct or glaciological and the geodetic method are the standard

methods to quantify glacier mass changes, and both methods are an integral part of international glacier monitoring strategies. In 2011, we established two glacier mass-balance programmes on Yala and Rikha Samba glaciers in the Nepal Himalaya. Here we present the methods and data of the directly measured annual mass balances for the first six mass-balance years for both glaciers from 2011/12 to 2016/17. For Yala Glacier we additionally present the directly measured seasonal mass balance from 2011 to 2017, and the mass balance from 2000 to 2012 obtained with the geodetic method. In additions, we analysed glacier

length changes for both glaciers. The directly measured average annual mass-balance rates of Yala and Rikha Samba glaciers are -0.80 ±0.28 m w.e. a$^{-1}$ and -0.39 ±0.32 m w.e. a$^{-1}$, respectively, from 2011 to 2017. The geodetically measured annual mass-balance rate of Yala Glacier based on digital elevation models from 2000 and 2012 is -0.74 ±0.53 m w.e. The cumulative mass loss for the period 2011 to 2017 for Yala and Rikha Samba glaciers is -4.80 ±0.69 m w.e. and -2.34 ±0.79 m w.e., respectively. The mass loss on Yala Glacier from 2000 to 2012 is -8.92 ±6.33 m w.e. The winter balance of Yala Glacier is

positive and the summer balance is negative in every investigated year. The summer balance determines the annual balance. Compared to regional mean geodetic mass-balance rates in the Nepalese Himalaya, the mean mass-balance rate of Rikha Samba Glacier is in a similar range, and the mean mass-balance rate of Yala Glacier is more negative because of the small and low-lying accumulation area. During the study period, a change of Yala Glacier's surface topography has been observed with glacier thinning and downwasting. The retreat rates of Rikha Samba Glacier are higher than for Yala Glacier. From 1989 to

2013, Rikha Samba Glacier retreated 431 m (-18.0 m a$^{-1}$), and from 1974 to 2016 Yala Glacier retreated 346 m (-8.2 m a$^{-1}$). The data of the annual and seasonal mass balances, point mass balance, geodetic mass balance and length changes are accessible from WGMS (2021): Fluctuations of Glaciers Database. World Glacier Monitoring Service, Zurich, Switzerland, http://dx.doi.org/10.5904/wgms-fog-2021-05.

## 1 Introduction

Glaciers are an essential climate variable (ECV) that contribute to understand and describe the global climate system (IGOS, 2007; Bojinski et al., 2014; Haeberli et al., 2000). The glacier mass balance is one of the seven headline indicators for global climate monitoring (Trewin et al., 2021) and one of the products of the ECV glacier, besides area and glacier thickness changes (GCOS, 2016). Mass-balance monitoring with the glaciological method is an integral part of international glacier monitoring strategies (Gärtner-Roer et al., 2019; Haeberli et al., 2007; Trewin et al. 2021). The glacier mass balance is relevant in various

regards, such as climate indicator, for glacier process understanding, the hydrological cycle and modelling, hazards and contribution to sea-level rise. As an input variable the mass balance is used to model the water availability and its change, and runoff scenarios for glacierized catchments and downstream livelihoods and ecosystems (Huss and Hock, 2018; Immerzeel et al., 2012; Kaser et al., 2010). The World Glacier Monitoring Service (WGMS) manages the database for glacier monitoring data including mass balance and frontal variation data, and runs the Global Terrestrial Network for Glaciers (GTN-G) in

collaboration with partners (IGOS, 2007; WGMS, 2020). GTN-G is the framework for the internationally coordinated monitoring of the ECV glacier, and in support of the United Nations Framework Convention on Climate Change (UNFCCC). In the Fourth Assessment Report of the Intergovernmental Panel on Climate Change (Cruz et al., 2007; Cogely et al., 2010), misinformation was published about an extreme above global average shrinkage of Himalayan glaciers. This led to the question about the actual status and future development of the glaciers in the Himalayas. The current contribution of glaciers in the

Hindu Kush Himalayan (HKH) region to the water availability downstream and sea-level rise still involve large uncertainties (Zemp et al., 2020; Immerzeel et al., 2019; Azam et al., 2018; Lutz et al., 2014; Marzeion et al., 2012; Bolch et al., 2012). Still only few programmes are established to monitor the in situ glacier mass balance and length changes on glaciers in Bhutan, China, India, Nepal and Pakistan (e.g. Azam et al., 2018; Wagnon et al., 2020; Tshering and Fujita, 2016; Dobhal et al., 2013), and only few include seasonal measurements (Wagnon et al., 2013; Azam et al., 2016; Sherpa et al., 2017). On a regional scale

glacier mass balances have been estimated by remote sensing techniques (e.g. Abdullah et al., 2020; Maurer et al., 2019; Gardelle et al., 2013; Vincent et al., 2013; Kääb et al., 2012; Berthier et al., 2007) and modelling (e.g. Fujita et al., 2011; Shea et al., 2015a; Tawde et al., 2017). However, due to the remoteness, high-altitude topography and logistical challenges there is still a lack of in situ measurements to validate and calibrate such studies. Some studies focused on ablation and runoff on a high spatial and temporal resolution on clean and debris covered glaciers (e.g. Litt et al., 2019; Pratap et al., 2019; Pratap et

al., 2015; Immerzeel et al., 2014; Fujita and Sakai, 2014), but rarely measured precipitation and snow accumulation in high altitudes due to challenges such as harsh conditions for precipitation measurements or difficult access to the accumulation zone.

A detailed review on the status and mass changes of Himalayan glaciers has been provided by Azam et al. (2018). They found that up to the year 2000, the mean glacier mass balance was in a similar range as the global average, but likely less negative

after 2000. The longest time series with direct glaciological measurements is found for Chhota Shigri Glacier, India, with measurements since 2002 (Mandal et al., 2020; Wagnon et al., 2007; Azam et al., 2012, 2014 and 2016). Other investigated glaciers in the Indian Himalaya are for example Dokriani, Gara, Gor Garang, Naradu, Neh Nar, Shaune Garang and Tipra Bank (Dobhal et al., 2008; Vincent et al., 2013; Pratap et al., 2015; Azam et al. 2018; WGMS, 2021). In the Chinese Himalaya, geodetic mass-balance data measured with differential global navigation satellite system (dGNSS) surveys are available from

1991 to 1993 and 2007 to 2010 for Kangwure Glacier, north of Mt Shisha Pangma and Langtang Valley, and from 2006 to 2010 on Naimona 'Nyi Glacier, in an upper tributary of the Ganges (Liu et al. 1996; Tian et al., 2014; WGMS, 2021). Additionally, glaciological mass-balance data are available for Kangwure Glacier from 1991 to 1993, and Naimona 'Nyi Glacier from 2006 to 2010. Glaciological and dGNSS mass-balance measurements have been carried out in Bhutan on Gangju La Glacier from 2003 to 2014 (Tshering and Fujita, 2016) and Thana Glacier since 2012 by the National Center for Hydrology

and Meteorology by the Government of Bhutan, and the partners ICIMOD and the Norwegian Water Resources and Energy Directorate. In Afghanistan, point measurements were initiated in 2017 on Pir Yakh Glacier and are continued by the University of Kabul, the Ministry of Energy and Water and supported by ICIMOD (WGMS, 2020).

In the Nepal Himalaya extensive glaciological measurements have been carried out by Japanese researchers on Rikha Samba Glacier in the Hidden Valley and AX010 in Shorong Himal since the 1970s, and on Yala Glacier in the Langtang Valley since

the 1980s (e.g. Ageta and Higuchi, 1984; Fujii et al., 1996; Fujita et al., 1998, 2001; Sugiyama et al., 2013). Mass-balance programmes were established on Mera Glacier in the Hinku Valley, Pokalde and West Changri Nup glaciers in the Khumbu Valley in 2007, 2009 and 2010, respectively (Wagnon et al., 2013; Sherpa et al., 2017). Wagnon et al. (2020) reanalysed the mass-balance data of Mera Glacier by using geodetic mass balances to calibrate the glaciological measurements from 2007 to 2019. Various researchers used the geodetic method with remote sensing products to calculate thickness changes (e.g. Bolch

et al., 2008; Bolch et al., 2011; Nuimura et al., 2012; Lindenmann, 2012; Ragettli et al., 2016).

On Rikha Samba Glacier, the first glaciological fieldwork was carried out in 1974 by Japanese researchers as part of the Glaciological Expedition of Nepal (GEN) (Fujii et al., 1976). Further fieldwork was carried out in October 1995, including

terminus surveys, glacier surface profiles, flow measurements, ice-core drilling and meteorological observations (Fujii et al., 1996; Fujita et al., 1997a; Shrestha et al., 1976). In October 1998 and 1999, stakes were installed and measured for direct point mass-balance measurements (Fujita et al., 2001). Terminus position changes and surface flow velocities were also measured and weather data collected. In 2010, the glacier surface was again surveyed by dGNSS and the geodetic mass balances calculated (Fujita and Nuimura, 2011) and meteorological data collected.

Yala Glacier was selected for the Himalayan Glacier Boring Project based on a GEN reconnaissance flight in Langtang Valley because it was the only one without debris cover and offered easy access to the glacier and the accumulation area (Watanabe et al., 1984). Comprehensive studies were carried out with a wide range of measurements in the field of glaciology, meteorology and geomorphology (e.g. Murakami et al., 1989; Ono, 1985; Yokohama, 1984). Stake measurements were taken in September and October 1982 (Ageta et al., 1984), and from summer 1985 to spring 1986 (Iida et al., 1987). In the accumulation area, Okawa (1991), Iida et al. (1984), Watanabe et al. (1984), and Steinegger et al. (1993) investigated the snow cover, boreholes, crevasses and ice cliffs to better understand the processes including mass balance, hydrology and snow metamorphosis. Fujita et al. (1998) carried out further glaciological measurements in 1994 and 1996 and documented an accelerated retreat and surface lowering of Yala Glacier in the 1990s and decreasing flow velocities. Various studies assessed historic and recent glacier fluctuations at Yala Glacier and in the Langtang Valley (e.g. Shiraiwa and Watanabe, 1991; Ono, 1985; Yamada et al., 1992; Kappenberger et al., 1993). Hydro-meteorological observations were made by Japanese researchers in 1982, 1985 to 1986, 1989 to 1991, and 2008 to 2011 (Yamada et al., 1992; Takahashi et al., 1987a, 1987b; Fujita et al., 1997b; Shiraiwa et al., 1992; unpublished data). Based on sensitivity studies and observational data from Yala and other Himalayan glaciers Fujita (2008a, 2008b) highlights the importance of precipitation seasonality on the climatic sensitivity of the glacier mass balance, besides air temperature changes.

In 2011 the HKH-Cryosphere Monitoring Project was initiated in Nepal by ICIMOD, and its partners the Department of Hydrology and Meteorology of the Government of Nepal, Kathmandu University and Tribhuvan University. The project goal was to improve the knowledge and understanding of the cryosphere in relation to climate change and impact on water resources in the HKH region and capacity building. Within this framework mass-balance monitoring programmes were established on Yala and Rikha Samba glaciers. An integral part of the project was to conduct training courses every year on the easily accessible Yala Glacier for a few dozens of students and professionals from the Himalayan countries, on one hand to build capacity for sustainable and consistent measurements, and on the other hand to promote the development of further mass-balance programmes in other parts of the HKH region. As a result, students from Kathmandu University utilized preliminary mass-balance data for their Master theses (Baral et al., 2014; Gurung et al., 2016; Acharya and Kayastha, 2019).

In this article we focus on the mass balance and glacier length changes of Yala and Rikha Samba glaciers measured within the framework of the HKH-Cryosphere Monitoring Project. At Yala Glacier we measured the mass balance twice a year in the field from 2011 to 2017, with remote sensing from 2000 to 2012, and assessed glacier length changes from 1974 to 2016. Additionally, we recorded supporting information such as flow velocity and direction. On Rikha Samba Glacier we assessed the annual mass balances and glacier length changes from 2011 to 2017, and 1989 to 2013, respectively. The methods are documented for these measurements and data submitted to the WGMS Fluctuations of Glaciers (FoG) database (WGMS, 2021), and for other supporting data beyond the scope of the WGMS FoG database.

## 2 Study areas and climatic setting

Yala Glacier is a small and debris-free glacier in central Nepal in Langtang Valley, and Rikha Samba Glacier is a valley glacier with a moderate altitude range located in western Nepal, in the Hidden Valley in Lower Mustang (Fig. 1, Table 1). Both glaciers are under the influence of the Indian summer monsoon, but Rikha Samba Glacier lies behind the main weather divide in the rain-shadow zone and receives less precipitation. Both glaciers are summer-accumulation-type glaciers (Ageta and

Higuchi, 1984), which are characterized by an overlapping main accumulation and ablation season during the monsoon season

(Fig. S1). A brief description of summer-accumulation-type glaciers and mass-balance measurements is provided in the

Supplement (section S1).

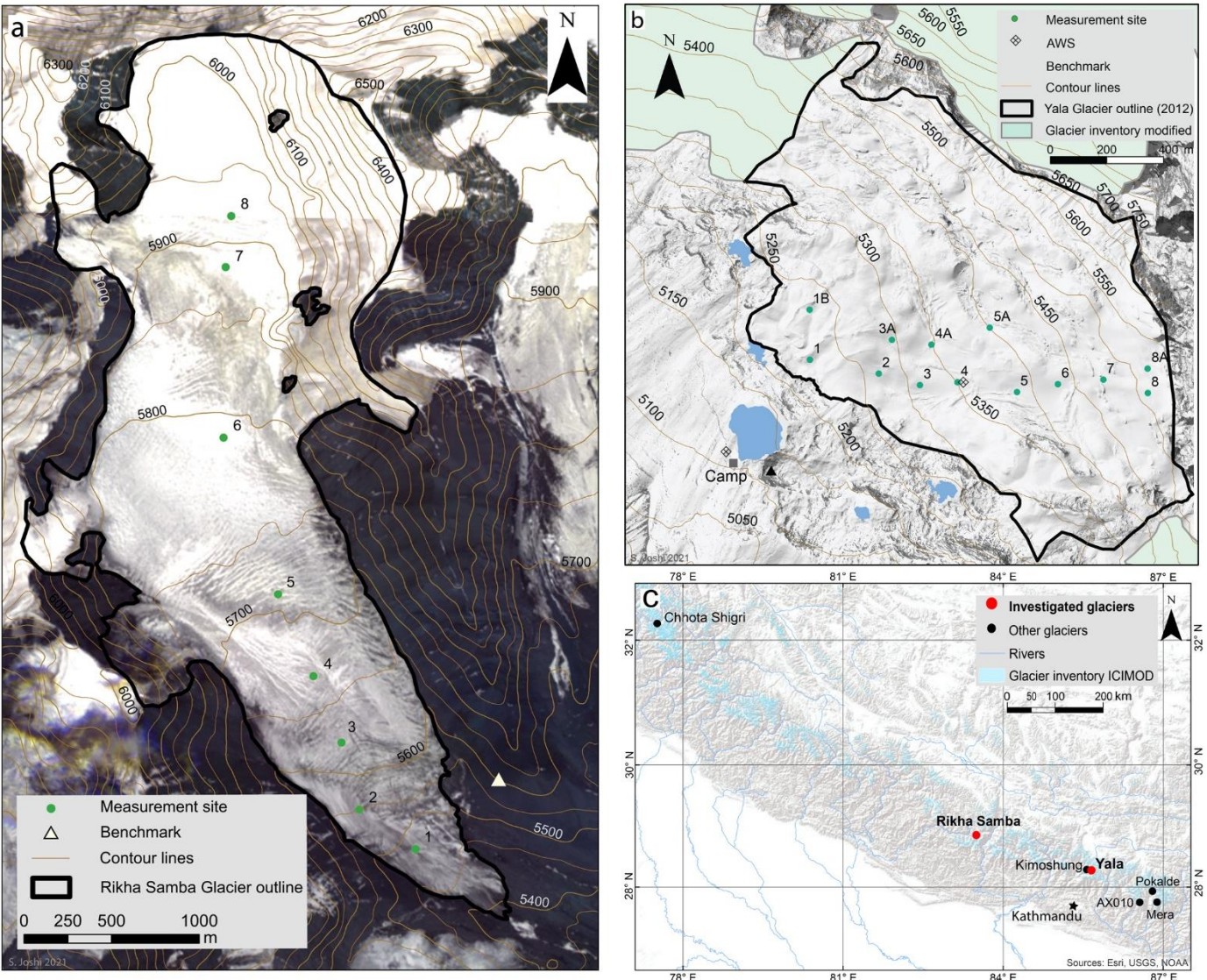

**Figure 1: The study sites Rikha Samba and Yala glaciers showing the measurement sites and their location in the Himalayas. At all measurements sites stakes were installed. Snow pits were dug at the top stakes and at selected lower stakes provided snow was**

**present. (a) For Rikha Samba Glacier RapidEye orthoimages from April 2010 were used for the background image and glacier outlines. The contour lines are derived from the SRTM-3 DEM. (b) For Yala Glacier GeoEye-1 orthoimages from January 2012 were used for the background image and in combination with dGNSS data for the glacier outlines. The contour lines are derived from the DEM2012 generated from the GeoEye-1 stereo images. (c) The overview map shows the location of the two investigated glaciers and other glaciers mentioned in the discussion section. The glacier inventory is from ICIMOD (Bajracharya et al., 2014).**

**Table 1. Geographic and topographic features of Yala Glacier in Langtang Valley and Rikha Samba Glacier in the Hidden Valley. The balanced-budget equilibrium-line altitude and accumulation-area ratio are denoted as $ELA_0$ and $AAR_0$.**

| General features of | Yala Glacier | Rikha Samba Glacier |
|---|---|---|
| Country, region | Nepal, Rasuwa district | Nepal, Mustang district |
| Mountain range | Langtang Himal, Central Nepal Himalaya | Dhaulagiri, Western Nepal Himalaya |
| River system | Trisuli basin, Ganges River | Kali Gandaki basin, Ganges River |
| Climate | Indian monsoon zone | Indian monsoon zone, rain shadow |
| Glacier type | Summer-accumulation type | Summer-accumulation type |
| **Glacier characteristics** | | |
| Latitude/Longitude | 28° 14' N, 85° 37' E | 28° 50' N, 83° 30' E |
| Elevation range | 5168–5661 m a.s.l. | 5416–6515 m a.s.l. |
| Glacier area/length | 1.61 km$^2$/1.4 km (2012, GeoEye-1) | 5.7 km$^2$/5.4 km (2010, RapidEye) |
| Orientation | South-west | South-east |
| Average slope | 25° | 5416–6000 m a.s.l.: 10° |
| | | 6000–6515 m a.s.l.: 36° |
| **Measurement information** | | |
| Maximum number of measurement sites | 14 (between 5175–5483 m a.s.l.) | 8 (between 5437–5900 m a.s.l.) |
| Measurement frequency | Twice a year in May and November (pre- and post-monsoon) | Annually in October (post-monsoon) |
| **Mass-balance information** | | |
| $ELA_0$ | ~5380 m a.s.l. | ~5760 m a.s.l. |
| $AAR_0$ | 0.49 | 0.66 |

## 2.1 Yala and Rikha Samba glaciers

Yala Glacier (28° 14' N, 85° 37' E) is located in the Rasuwa district, Central Nepal about 70 km north of Kathmandu, draining into Langtang River which feeds the Trisuli, and then Ganges rivers. It is a plateau-shaped glacier, ranging from 5168 m to 5661 m a.s.l, and with a length and area of about 1.4 km and 1.61 km$^2$, respectively (Fig 1). The ice body extends further to north-west on a similar elevation range, with steep slopes, ice cliffs and rockfall areas. For the mass-balance analyses, Yala Glacier's drainage basin was separated from the adjacent ice body along the flowline.

The glacier faces mainly south-west and the average slope is 25°. Numerous ice cliffs and steep slopes are distributed over the glacier area, but mainly in the northern part of the glacier. The mean and maximum ice thickness measured by ground penetrating radar (GPR) was 36 m and 61 m in 2009, and the glacier bed topography indicates several small overdeepenings (Sugiyama et al., 2013). The glacier is polythermal (Okawa, 1991; Sugiyama et al., 2013), has clean ice with little debris and small proglacial ponds.

In the 2015 Nepal earthquake, rockfall covered parts of the ice body, which is next to the defined outlines of Yala Glacier. In these parts we find a transition from debris-covered glacier to possible permafrost with refrozen meltwater and buried ice. Yala Glacier sits on a gneiss bedrock shelf, which forms part of the base from which a large landslide slipped (Weidinger et al., 2002; Takagi et al., 2007). Weidinger et al. (2002) suggest that the landslide was a mountain of about 8000 m height, which collapsed about 51 ±13 ka ago (Takagi et al., 2007). The dislocated mass lies south-west of Yala Glacier and has largely been eroded in the most recent glaciation. The landslide left behind an open topography, with Yala Glacier located within and sheltered by the surrounding high mountains of the Langtang range (>6500 m a.s.l.).

Rikha Samba Glacier (28° 50' N, 83° 30' E) is located in the Hidden Valley on the north side of the main range, and is part of Lower Mustang. The Sangda River drains the Hidden Valley and joins the Kali Gandaki River further down. The glacier has an elevation range of 5416 m to 6515 m a.s.l. and a length and area of 5.4 km and 5.7 km$^2$. The ice is polythermal and the maximum ice thickness measured is 178 ±2 m (Gilbert et al., 2020). At about 6000 m at the head of the valley, the glacier is wide and flows down with a gentle slope of ~10° on average, facing mainly south, and south-east at the glacier tongue. Above 6000 m a.s.l, the glacier is steep with a slope of ~36° an average making up 19 % of the glacier area, and flowing down from the sides of the valley.

## 2.2 Climate

The Himalayan mountains are an orographic barrier causing strong north-south but also altitudinal temperature and precipitation gradients. Nepal is under the influence of the Indian summer monsoon that brings the majority of the annual precipitation, and receives some precipitation from westerlies in winter (Bookhagen and Burbank, 2010). The interannual variability of precipitation is much larger in winter than in summer, caused by westerly disturbances and occasional cyclones originating in the Bay of Bengal (Seko and Takahashi, 1991; Fujita et al., 1997b). However, climate information from high elevations in the HKH is sparse. The few high-altitude climate stations are mostly situated in valley floors, and satellite derived products are less reliable (Salerno et al., 2015; Shea et al., 2015b; Ménégoz et al., 2013). Snowfall studies quantifying timing and amounts are sparse but critical (Litt et al., 2019), and automated snowfall measurements are challenging because undercatch can be up to 20 % to 50 % in windy conditions (Rasmussen et al., 2012). Meteorological data from Rikha Samba Glacier, Yala Glacier and other automatic weather stations (AWS) in the Langtang and Dudh Koshi catchments were compared by Shea et al. (2015b). They analysed temperature, incoming radiation, wind, precipitation and other parameters from December 2012 to December 2013, as far as data were available.

Precipitation has been analysed for the Langtang Valley and Rikha Samba Glacier based on reanalysis data and field measurements (Immerzeel et al., 2012; Racoviteanu et al., 2013; Fujita et al., 2001). Immerzeel et al. (2012) found that the upper Langtang catchment received 814 mm of precipitation per year, and 77 % of it during monsoon from June to September based on ERA40 data from 1957 to 2002. The AWS nearest to Yala Glacier with long-term data is in Kyangjing at 3920 m a.s.l., which is about 6 km horizontal distance and south-west from Yala Glacier. Racoviteanu et al. (2013) analysed the AWS data at Kyangjing between 1988 and 2006 and found a mean annual precipitation of 647 mm. Fujita and Nuimura (2011) estimated the long-term annual mean precipitation at Yala Glacier to be 772 mm. From December 2012 to November 2013, Shea et al. (2015) measured 924 mm precipitation in Kyangjing, which includes an extreme precipitation event in October 2013. The conditions at the leeside of the main mountain range at Rikha Samba Glacier are much drier. Precipitation measured with a totalizer and a tipping bucket in the vicinity of the terminus of Rikha Samba Glacier (5267 m a.s.l.) amounted to about 450 mm from October 1998 to September 1999 (Fujita et al., 2001). The precipitation measured from October to April is minimal and likely indicates underrepresented snowfall. Fujita and Nuimura (2011) estimated at least 370 mm of long-term mean annual precipitation at Rikha Samba Glacier, and Shrestha et al. (1976) measured 203 mm of precipitation at 5055 m a.s.l. in the Hidden Valley during monsoon from July to early September 1974.

The mean annual air temperature in Kyangjing was about 4° C from 1988 to 2012. Near Rikha Samba Glacier's terminus, the mean annual air temperatures were -4.6° C and -5° C, at 5267 m a.s.l. in 1999 and at 5310 m a.s.l. in 2014, respectively (Fujita et al., 2001; Gilbert et al., 2020). Temperature lapse rates vary with the season, with largest and smallest lapse rates in winter and summer, respectively (Immerzeel et al., 2015). The diurnal temperature variabilities are smallest during monsoon (Shea et al., 2015b).

The sky in the Nepal Himalaya is generally clear in the post-monsoon and winter season (Fujita et al., 2001). Cloudiness increases during pre-monsoon and reaches a maximum during monsoon. During monsoon, the cloudiness at Yala Glacier is much higher than at Rikha Samba Glacier, which can be explained by the valley circulation and cloud formation patterns in the Langtang Valley and on the leeside location of Rikha Samba Glacier (Fujita et al., 2001; Shea et al., 2015b; Litt et al., 2019). During post-monsoon and winter, the incoming solar radiation is higher at Yala Glacier, which can be explained by the south-west aspect of the glacier and the open topography left behind by the landslide.

The wind directions at the Yala Base Camp AWS show a dominance of bimodal valley winds (Shea et al., 2015b). The Rikha Samba AWS is additionally exposed to synoptic-scale flows. Throughout the year, the wind velocities at Rikha Samba Glacier are higher and with a larger variability than at Yala Glacier. The highest wind speeds are recorded in winter from October to May, with strong wind events of >8 ms$^{-1}$ (Fujita et al., 2001). Winter wind velocities measured at Rikha Samba Glacier are very high and result from the channelling of synoptic-scale winds (Shea et al., 2015b). The winter wind speeds at Yala Glacier

are much smaller, probably because Yala Glacier is better sheltered by surrounding high mountains. During monsoon from June to September the wind speeds at both glaciers are lower with a smaller variability.

## 3 Data and methods

The mass balance of the two glaciers was monitored from 2011 to 2017 with the glaciological method using stakes, snow pits and cores, and for Yala Glacier also with the geodetic method from 2000 to 2012. The frontal variations were evaluated based on satellite images, dGNSS and global positioning system (GPS) data.

### 3.1 Data collection

The in situ measurements started in autumn 2011, and were conducted twice a year on Yala and annually on Rikha Samba glaciers. On Yala Glacier, the annual/summer balance measurements were taken in November. The winter balance was measured in late April or early May, and in 2015 in early June due to the major earthquake in Nepal on 25 April 2015. On Rikha Samba Glacier, in the first years the measurements were carried out in September, which is rather early in the season because it is still under the influence of the monsoon. In the following years, the measurements were carried out in October or November. Generally, October and November are ideal periods for mass-balance measurements in the central Nepal Himalaya, but coincide with the main festival season in Nepal. The festival season is of great religious importance, lasts for several weeks, and varies every year by weeks. This makes it hard to do fieldwork at fixed dates and find people to conduct measurements. In autumn, the expeditions to Yala Glacier were conducted after the last festival ended to allow members from various institutions and universities to participate in the training courses.

#### 3.1.1 In situ mass balance

The in situ mass balance was measured following Kaser et al. (2003), taking into consideration aspects in the ablation and accumulation area specific to summer-accumulation type glaciers (for details see Supplement, section S1). In the ablation area, the mass balance was measured with bamboo stakes. If snow was present, its depth was usually measured at each measurement site, and at selected stakes the snow density and profile were also recorded.

In the accumulation area, snow pits were dug or cores taken, and the snow profile, depth and density recorded. Additionally, several snow probing measurements were taken. Bamboo stakes mainly marked the measurement sites, but in absence of snow-pit data they were also used for the mass-balance calculation, in particular in the case of a negative mass balance. The snow-pit measurements were only reliable if the previous measurement surface could be clearly identified, e.g. when marked with a sawdust layer. Difficulties arose in the accumulation area, if the cumulative ablation temporarily exceeded the cumulative accumulation during the measurement period (Fig. S2). The exceeding ablation is not represented in a snow-pit measurement and likely impacts the sawdust layer. Stake readings were less reliable because the underlying snow and firn layers compact over time and may push or pull the stake up or down.

On Yala Glacier, the measurements stretch along a line established in the past by Japanese researchers (Fujita et al., 1998). In the lower part a few stakes were initially added in a transect. Since the glacier has been shrinking, a second row of stakes was installed parallel to the original line in November 2016, in an attempt to maintain measurements also in future when the glacier retreats beyond the current stake locations. In the northern and highest parts of the glacier no measurements were taken because steep terrain, crevasses and ice cliffs made access difficult.

On Rikha Samba Glacier, eight stakes were installed along the approximate glacier centre line with some deviation, which follow roughly the stake setup of the Japanese researchers (Fujita et al., 2001). In the first year, the lower five stakes were installed, and in 2012 three additional measurement sites were established. Snow depth was probed, and the density measured in snow pits, but sawdust was spread only during few occasions and found only once, making accumulation measurements

challenging. In 2011 and 2014, the conditions on the glacier were very difficult and the higher part of the glacier could not be reached.

### 3.1.2 GNSS surveys

Differential GNSS was used to survey the glacier termini, measurement sites, benchmarks, thickness changes along profiles, and surface velocities (Table S1). The devices were dual frequency dGNSS units from Topcon and Magellan ProMark 3, and

were used in real time kinematic (RTK) mode. The instrument accuracy is within a 10 mm range in RTK mode after post-processing. In the field the antenna was kept vertical in the backpack as much as possible and thus the accuracy is estimated to be ±0.3 m. Yala Glacier's terminus was mapped with a handheld Garmin GPS unit in November 2012 and dGNSS Topcon units in May 2014 and 2016. On Rikha Samba Glacier, the terminus was surveyed with a dGNSS Topcon unit in September 2013.

The glacier surface profiles of Yala and Rikha Samba glaciers were repeatedly surveyed with dGNSS, along a longitudinal profile and three and two cross-profiles, respectively, but only data from May 2012 from Yala Glacier are presented here. Already Sugiyama et al., (2013) surveyed the profile line on Yala Glacier in 2009. The repeated measurements provide the opportunity to further analyse the mass balance with an independent complementing method (Wagnon et al., 2013, 2020). Annual surface velocities were derived from stake displacements between 8 May 2012 and 5 May 2014 on Yala Glacier.

### 3.2 Maps, satellite images and DEMs

For Yala Glacier, various maps were compared and evaluated for their suitability for area, volume and frontal change analysis. The maps included the Survey of India, the Schneider and the Nepal topographical maps published in 1965, 1990 and 1995, the map by the Japanese Glaciological Expedition Nepal (GEN) map (Yokoyama, 1984) and the glacier outlines from the ICIMOD glacier inventory of Nepal (Bajracharya et al., 2014; Table S2). The GEN map and glacier inventory data were used;

however, despite good quality no other maps could be used because of transformation issues and inconsistencies. The GEN map is based on a ground photogrammetric field survey in 1981 (Yokoyama, 1984). The photo point was about 2 km from the glacier terminus in 1981 on a lower location; consequently, the exposing axis is almost parallel to the glacier surface. We found a distortion and mismatches at the ridge and at the south-east and north-west side of the glacier. We georeferenced the map with the GeoEye-1 orthoimage from 2012 to calculate the frontal variations but did not use it for area or geodetic mass-

balance analyses.

Satellite images were used to delineate glacier outlines and termini of both glaciers, and to calculate the geodetic mass balance of Yala Glacier (Table S3). The SRTM-3 DEM (SRTM-3) is the third version of the DEM from the Shuttle Radar Topography Mission (SRTM) and is generated based on data from 2000. The spatial resolution is about 90 m, with an absolute vertical accuracy of ±16 m and a vertical reference to the WGS 84 EGM96 geoid (Rabus et al., 2003). The penetration of the SRTM

C-band beam in snow, firn and glacier ice is an issue that results in a lower accuracy especially in the accumulation area (Kääb et al., 2012; Berthier et al., 2006). SRTM-3 was resampled to 30 m for the geodetic mass-balance calculation of Yala Glacier. The SRTM-1 DEM was used for the mass-balance analysis of Rikha Samba Glacier. It is based on the SRTM-3 data from 2000 but was released with an improved resolution of about 30 m.

The GeoEye-1 is a commercial high-resolution stereo satellite image with 0.5 m spatial and 8 bits per pixel radiometry

resolutions. The stereoscopic images from 15 January 2012 were used to generate a DEM (DEM2012) for Yala Glacier to calculate the glacier-wide geodetic mass balance, and the orthoimage was used to delineate the outlines.

We used Landsat images for various purposes. A Landsat 8 image acquired on 18 November 2013 was used to collect horizontal references (x, y) and the SRTM-3 for the vertical reference (z) for ground control points (GCP) to georeferenced the GeoEye-1 images, and tie points for DEM generation for Yala Glacier. A Landsat 7 Enhanced Thematic Mapper (ETM+)

image from 2000 helped to identify the outlines of Yala Glacier for the geodetic mass balance and to analyse frontal variations.

We analysed terminus changes of Rikha Samba Glacier using a Landsat 4, Landsat 7 ETM+ and two Landsat 5 Thematic Mapper (TM) images from the years 1989, 2001, 2006 and 2011, respectively. RapidEye images from 25 and 27 April 2010 were used to delineate the outlines of Rikha Samba Glacier.

A Hexagon KH-9 image from November 1974 was used for a frontal variation analysis of Yala Glacier. Other Hexagon images were found unsuitable for area and volume analysis because of void areas, or cloud and snow cover in the images. Additionally, it was difficult to delineate the glacier at the north-west and south-east side without contour lines to derive the flowlines at that time.

For this study, we adopted the projection system WGS 1984, UTM Zone 44N and 45N for Rikha Samba and Yala glaciers, respectively. We used the local projection system called Modified Transverse Mercator, with false easting 500,000 m and scale factor of 0.9999 at the central meridian 84° E and 87° E for Rikha Samba and Yala glaciers, respectively.

### 3.3 DEM generation

The DEM generation from GeoEye-1 stereo images from 2012 involved four steps, following Holzer et al. (2015): collection of GCPs, extraction of the DEM, and the two post-processing steps to clean DEM areas of low quality and to co-register the DEM.

Eight GCPs were used to georeference the GeoEye-1 stereo satellite images. The GCPs were obtained from stable terrain and were evenly distributed. The x and y coordinates of the GCPs were measured from a Landsat 8 image from November 2013, and the z-values were taken from the SRTM-3 DEM. All GCPs were cross-checked in Google Earth[TM].

For the DEM extraction from the GeoEye-1 stereo images OrthoEngine from PCI Geomatica 2013 software was used. The DEM was derived using the Rational Function model with first-order RPC adjustments from ephemeral data and GCPs. We applied the Wallis filtering to locally enhance the contrast of the image to improve the image matching. The DEM derived from the forward- and backward-looking images has a resolution of 2 m.

In the next step, DEM areas of low quality were removed. First the SRTM-3 DEM and the GeoEye-1 DEM were resampled from 90 to 30 m, and from 2 to 5 m, respectively, and aligned to a raster grid of same extent and cell alignment. Then the noises in the GeoEye-1 DEM were eliminated applying the expand-sink-expand tool and a median filter (5 x 5 m). With the hillshade of the GeoEye-1 DEM we visually checked the DEM. To evaluate the image matching, PCI produces a score channel image, which we used to identify DEM areas of poor quality and set the values to "no data". Especially a small part of the north-eastern glacier area at Yala ridge had to be discarded due to a very low DEM quality.

In the DEM co-registration process, the SRTM-3 is the reference (master) DEM to which the GeoEye-1 slave DEM is co-registered. For the horizontal DEM co-registration, first we calculated the elevation difference of the GeoEye-1 DEM relative to the SRTM-3. We excluded non-stable terrain such as glaciers and landslide areas and used only terrain with a slope between 10° and 45° in SRTM-3. The SRTM-3 had initially a much coarser resolution than the GeoEye-1 DEM, leading to a resolution-induced bias at topographic extremes with strong curvature (Berthier et al. 2006; Paul, 2008; Gardelle et al., 2012). To account for such curvature effects and most extreme outliers particularly at steep slopes, we identified and removed DEM difference values in the 5 % and 95 % quantiles, as well as pixels outside the two-tailed 1.5 times interquartile range (Pieczonka et al., 2013). The horizontal shift between the two DEMs we corrected manually due to the small study area, followed by a two-dimensional spatial trend correction. For the vertical DEM co-registration of the GeoEye-1 DEM, the flat areas less than 10° of the SRTM-3 were used, avoiding steeper terrain with decreasing accuracy in SRTM-3. The DEM2012 was resampled to a resolution of 5 m for the geodetic method and 30 m for the glaciological method.

### 3.4 Analysis of glacier changes and uncertainties

### 3.4.1 Point and glacier-wide mass balance

The glacier-wide mass balances, the equilibrium-line altitude (ELA) and accumulation-area ratio (AAR) were calculated based on the interpolated mass-balance gradient derived from the point measurements following a similar method used by Wagnon et al. (2013) for Mera and Pokalde glaciers. The mass-balance gradients were derived from the linear regression lines of the point measurements. The elevations of the DEM2012 for Yala Glacier and the SRTM1 for Rikha Samba Glacier were applied

to the regression equations to calculate the glacier-wide mass balance.

For Yala Glacier, characteristic gradients for the ablation area were identified, and separately analysed for the annual and seasonal mass balances, with the winter and summer season starting in November and May or June, respectively. In the accumulation area, there are fewer measurements with large uncertainties because of the challenging measurement conditions described earlier and in Supplement section S1. This inhibited not only to identify characteristic gradients in the accumulation

area, but also to define a fixed mass balance that could be applied in the accumulation area from a defined elevation upwards. As a consequence, a single gradient was used for the glacier-wide mass balance. The interpolation approach is simple and introduces a systematic error for the mass balance in the accumulation area. The part of the accumulation area without measurements for the respective elevations bands makes up 15 % of the glacier area for an elevation range of about 160 m (~5500 m to 5662 m a.s.l.).

For Rikha Samba Glacier two characteristic annual gradients were identified, with a large gradient in the lower ablation area and a medium gradient in the transition between ablation and accumulation area. Based on the assumption that the mass-balance gradients remain very similar in different mass-balance years, gradients were reconstructed for Rikha Samba Glacier for years with limited point measurements (2011/12, 2013/14, and 2014/15). The intersection points of the lower (large) and upper (medium) gradients were identified and reconstructed based on a regression line for sections without measurements. For

the accumulation area, no characteristic gradients could be identified because only few measurements were available. The elevation range without measurements is about 650 m (~5900 m to 6545 m a.s.l.) and makes up 36 % of the glacier area. At about 6000 m, the topography steepens (Fig 1). Using the upper gradient to interpolate the mass balance to the accumulation area would have resulted in much overestimated positive mass balances. Instead we considered it plausible to assume a fixed mass balance at high elevations, based on the steep slopes and the typically small gradient in accumulation areas. We assumed

the lower elevation for a fixed mass-balance value between 5850 m and 5950 m a.s.l., guided by the upper gradient. For the mass-balance year 1998/99, the point measurements collected by Fujita et al. (2001) were used. The ELA and AAR were calculated based on the mass-balance gradients, whereas for Rikha Samba Glacier the upper gradient was used.

The errors of the point measurements were assessed by analysing the random errors for each measurement from density $\sigma_d$, ice surface roughness $\sigma_{rough}$ mainly in the ablation area, varying snow depth $\sigma_{depth}$ mainly in the accumulation area, stake

reading $\sigma_{read}$, errors due to the sawdust spread for snow-pit measurements $\sigma_{sawd}$ and movement of the stake in the firn area $\sigma_{firn}$. The error of an individual point measurement $\sigma_{point}$ was calculated:

$$\sigma_{point} = \sqrt{\sigma_d^2 + \sigma_{rough}^2 + \sigma_{depth}^2 + \sigma_{read}^2 + \sigma_{sawd}^2 + \sigma_{firn}^2} \tag{1}.$$

At few sites with minimal flow, two measurements from older and newer stakes allowed a comparison. In most cases the measurements were within the calculated error. Otherwise, if no explanation was found for differing values, the standard

deviation of the two values was taken as error.

To assess the error $\sigma_{final}$ of the mass balance for the entire glacier and elevation bands of 50 m, the errors of the point measurements $\sigma_{point\_elevb}$ and interpolation method $\sigma_{int}$ were analysed. Due to a lack of updated glacier surface and outline data, the reference-surface balance was calculated (Elsberg et al., 2001), and the systematic errors caused by the changing

glacier geometry were disregarded. Also, the systematic errors caused by stakes placed at unrepresentative locations or even
lack of point measurements were not evaluated due to a lack of respective information.

The overall error $\sigma_{final}$ for the mass balance for the glacier-wide balance and elevation bands was calculated:

$$\sigma_{final} = \sqrt{\sigma_{point\_elevb}^2 + \sigma_{int}^2} \tag{2}.$$

The error of the point measurements for a specific elevation band $\sigma_{point\_elevb}$ was calculated by considering $n$ point
measurements in the respective elevation band:

$$\sigma_{point\_elevb} = \sqrt{\sum_{point=1}^{n} \sigma_{point}^2}/\sqrt{n} \tag{3}.$$

To calculate the systematic error caused by the interpolation method $\sigma_{int}$, we estimated the maximum difference in mass
balance for 50 m elevation bands. The standard deviation of this value and the calculated mass balance was assumed as the
error from the interpolation method.

The error of the cumulative mass balance $\sigma_{cumul}$ for $n$ years was calculated:

$$\sigma_{cumul} = \sqrt{\sum_{years=1}^{n} \sigma_{final}^2} \tag{4},$$

And the error of the mean annual mass-balance rate $\overline{\sigma_{cumul}}$ for $n$ years was calculated:

$$\overline{\sigma_{cumul}} = \sqrt{\sum_{years=1}^{n} \sigma_{final}^2}/\sqrt{n} \tag{5}.$$

The accuracy of the ELA and AAR were estimated by shifting the regression lines based on point measurements deviating
from the initial regression line. For Rikha Samba Glacier the calculation of the ELA and AAR for the years 2011/12, 2013/14
and 2014/15 were omitted due to the very few measurements.

### 3.4.2 Glacier area and length

The area of Yala Glacier was defined based on the GeoEye-1 orthoimage from 15 January 2012, and GPS data of the terminus
from 3 November 2012. On the north-west side, the glacier's drainage basin has been separated from the adjacent ice body
along the flowline, using flow vectors drawn perpendicular to the contour lines derived from the DEM2012 (Cuffey and
Paterson, 2010). A section detached from the main glacier on the south-east side was excluded. For the analysis of the geodetic
mass balance, the glacier outline is based on the Landsat 7 ETM+ image from February 2000 (Table S3).

The frontal variations of Yala Glacier were analysed with satellite images, maps and field-based data (Table S1, S2, S3). Yala
Glacier is very wide and the terminus is not constrained by a valley, hence it is difficult to identify a central flowline of the
glacier. Instead, we delineated the general glacier flow direction with the 'rectilinear box method' described by Lea et al.
(2014) and Koblet et al. (2010). In this method an arbitrary rectangular box is drawn along the flowline. Perpendicular to the
flowline and at the maximum extent of the Hexagon KH-9 1974 glacier outline, a straight arbitrary baseline was drawn.
Perpendicular to the baseline and in flow direction, 26 parallel lines at 50 m intervals were drawn to quantify the glacier
terminus changes. At each parallel line we measured the frontal variation and averaged the values for the final frontal variation
of that period. There are big outliers, and some of the mapped termini were not covered by all 26 parallel lines. Therefore, for
the final calculation only nine parallel lines which covered the lowest parts of the glacier were considered.

For Rikha Samba Glacier, the glacier outline was delineated from RapidEye images from 25 and 27 April 2010. The frontal
variations are quantified along the central glacier flowline that was derived from SRTM-1. The glacier termini are based on
Landsat images from 1989, 2001, 2006, 2011 and a dGNSS survey from 2013 (Table S1). Uncertainties of glacier termini and
outlines are estimated half to one pixel dependent on the quality of the source image or map scale, or according to the dGNSS
settings and field conditions.

### 3.4.3 Geodetic mass-balance calculation

The geodetic mass-balance calculation for Yala Glacier is based on the subtraction of the SRTM-3 from the DEM2012 from the years 2000 and 2012, respectively, which results in a map of elevation differences (Δh). Data gaps smaller than 0.01 km$^2$ in the elevation difference map were filled with a mean filter of surrounding height change (Δh) values. The accumulation and ablation areas were separated by an estimated ELA of 5350 m a.s.l. Outliers and voids larger than 0.01 km$^2$ occurred only in the accumulation area. The largest data gaps were found at the edge of the glacier at Yala ridge, where fresh snow in the GeoEye-1 image compromised the quality of the DEM2012. However, no plausible statistical value could replace the data voids and outliers, therefore, the mode value from the accumulation area was taken, assuming only minor elevation changes in these areas (Schwitter and Raymond, 1993). Assuming an average density of 850 kg m$^{-3}$ (Huss, 2013) for the entire glacier, the elevation change was converted into mass change. Since the accumulation area was small, only a single density value was used. The glacier area was defined by the larger extent from the Landsat 7 image from February 2000. Additionally, the glacier surface elevation changes of Yala Glacier were analysed along the profile line surveyed by dGNSS in May 2012, and compared to SRTM-3.

The SRTM-3 C-band potentially underestimates the glacier elevations because of radar penetration into the upper snow, firn and ice layers on the glacier (Kääb et al., 2012; Gardelle et al., 2012). In winter in the Karakoram, Gardelle et al. (2012) found a penetration on glaciers of a couple of metres below 5300 m, which increases to about 5 m at 5700 m and more above. They emphasise that these values can vary in different regions, decreasing penetration in wetter and warmer snow and dirtier ice. Bolch et al. (2016) use a mean average penetration correction of 2.4 ±1.4 m to address this issue in the Karakoram. The Landsat 7 image from February 2000 showed some snow cover. In this study, we assume that the SRTM-3 DEM represents the glacier surface from early 2000 because on average, we expect only a small snow cover. Additionally, the accumulation area on Yala Glacier is small and on low elevation, reducing the effect of the penetration.

To assess the uncertainty of the thickness change, we estimated the vertical precision of the DEMs by calculating the normalized median absolute deviation (NMAD), which is ±7.41 m (Holzer et al., 2015; Höhle and Höhle, 2009). The uncertainty of the geodetic mass balance is the root of the sum of each squared error term, which consist of the NMAD and the uncertainty for the ice density of ±60 kg m$^{-3}$ (Huss, 2013). Errors due to different spatial scale, sensors, resolutions and area of Yala Glacier were not considered.

## 4 Results

### 4.1 Mass balances, ELA, AAR and gradients

The glacier-wide annual mass balances of Yala and Rikha Samba glaciers were negative for all years, except in 2012/13 when Yala Glacier was almost in balance (-0.01 ±0.29 m w.e.), and Rikha Samba Glacier had a slightly positive balance (0.12 ±0.32 m w.e.), reported in Table 2 and 3, and Fig. 2 and 3. The most negative annual balances on Yala Glacier occurred in 2016/17 and 2014/15 with -1.54 ±0.20 m and -1.18 ±0.26 m w.e. In the years 2011/12, 2013/14 and 2015/16 the values were similarly negative for Yala Glacier (-0.86 ±0.40 m, -0.61 ±0.27 m and -0.61 ±0.23 m w.e.). On Rikha Samba Glacier, 2011/12 was the most negative year (-0.72 ±0.34 m w.e.), followed by 2014/15 (-0.63 ±0.35 m w.e.). In the years 2011/12, 2013/14 and 2014/15, the balances were similarly negative (-0.72 ±0.34 m, -0.55 ±0.34 m and -0.63 ±0.35 m w.e.), followed by less negative years in 2015/16 and 2016/17 (-0.33 ±0.27 m and -0.23 ±0.31 m w.e.). The mean annual mass-balance rate and cumulative balance of Yala and Rikha Samba glaciers from 2011 to 2017 are -0.80 ±0.28 m w.e. a$^{-1}$, -4.80 ±0.69 m w.e., and -0.39 ±0.32 m w.e. a$^{-1}$, and -2.34 ±0.79 m w.e., respectively. The most negative point mass balances of -3.75 ±0.05 m w.e. and -4.12 ±0.04 m w.e., respectively, were measured at the lowest stakes (5175 m and 5437 m a.s.l.) of Yala and Rikha Samba glaciers in 2011/12.

**Table 2: Mass balance (B) measured with the glaciological method, winter balance (Bw), summer balance (Bs), ELA, AAR and mass-balance gradient for Yala Glacier from 2011/12 to 2016/17. The summer balance from 2011/12 and winter balance from 2014/15 (\*) have not been reported to the WGMS and are discussed in subsection 5.1.2 Seasonal mass balance.**

| B year | B (m w.e.) | $B_W$ (m w.e.) | $B_S$ (m w.e.) | $B_W+B_S$ (m w.e.) | ELA (m a.s.l.) | AAR | db/dz (m w.e. $(100\ m)^{-1}$) |
|---|---|---|---|---|---|---|---|
| 2011/12 | -0.86 ±0.40 | 0.16 | -0.20* | -0.03 | 5454 ±30 | 0.28 | 1.14 |
| 2012/13 | -0.01 ±0.29 | 0.36 | -0.35 | 0.01 | 5380 ±20 | 0.48 | 0.99 |
| 2013/14 | -0.61 ±0.27 | 0.27 | -0.99 | -0.73 | 5431 ±20 | 0.35 | 1.18 |
| 2014/15 | -1.18 ±0.26 | 0.54* | -1.12 | -0.59 | 5510 ±40 | 0.13 | 0.90 |
| 2015/16 | -0.61 ±0.23 | 0.19 | -0.79 | -0.60 | 5444 ±20 | 0.31 | 0.93 |
| 2016/17 | -1.54 ±0.20 | 0.20 | -1.75 | -1.54 | 5518 ±20 | 0.12 | 1.10 |
| Mean | -0.80 ±0.28 | 0.29 | -0.87 | -0.58 | 5456 | 0.28 | 1.04 |
| STD | 0.53 | 0.14 | 0.56 | 0.56 | 52 | 0.14 | 0.12 |
| 2011–2017 | -4.80 ±0.69 | 1.72 | -5.21 | -3.48 | | | |


**Table 3: Mass balance (B) measured with the glaciological method, ELA, AAR and the lower and upper mass-balance gradients for Rikha Samba Glacier for the mass-balance years 1998/99, and from 2011/12 to 2016/17. We did not calculate the ELA and AAR for 2011/12, 2013/14 and 2014/15 due to the very few data points. For the mass-balance year 1998/99, the point measurements collected by Fujita et al. (2001) were used.**

| B year | B (m w.e.) | ELA (m a.s.l.) | AAR | db/dz (lower) (m w.e. $(100\ m)^{-1}$) | db/dz at ELA (upper) (m w.e. $(100\ m)^{-1}$) |
|---|---|---|---|---|---|
| 1998/99 | -0.18 | 5790 ±50 | 0.49 | 1.27 | 0.25 |
| 2011/12 | -0.72 ±0.34 | – | – | 1.13 | |
| 2012/13 | 0.12 ±0.32 | 5724 ±20 | 0.75 | 1.57 | 0.37 |
| 2013/14 | -0.55 ±0.34 | – | – | 1.36 | |
| 2014/15 | -0.63 ±0.35 | - | – | 1.48 | |
| 2015/16 | -0.33 ±0.27 | 5872 ±50 | 0.41 | 1.64 | 0.36 |
| 2016/17 | -0.23 ±0.31 | 5862 ±50 | 0.54 | 1.89 | 0.46 |
| Mean | -0.39 ±0.32 | 5807 | 0.55 | 1.48 | 0.36 |
| STD | 0.31 | 63 | 0.15 | 0.25 | 0.09 |
| 2011–2017 | -2.34 ±0.79 | | | | |


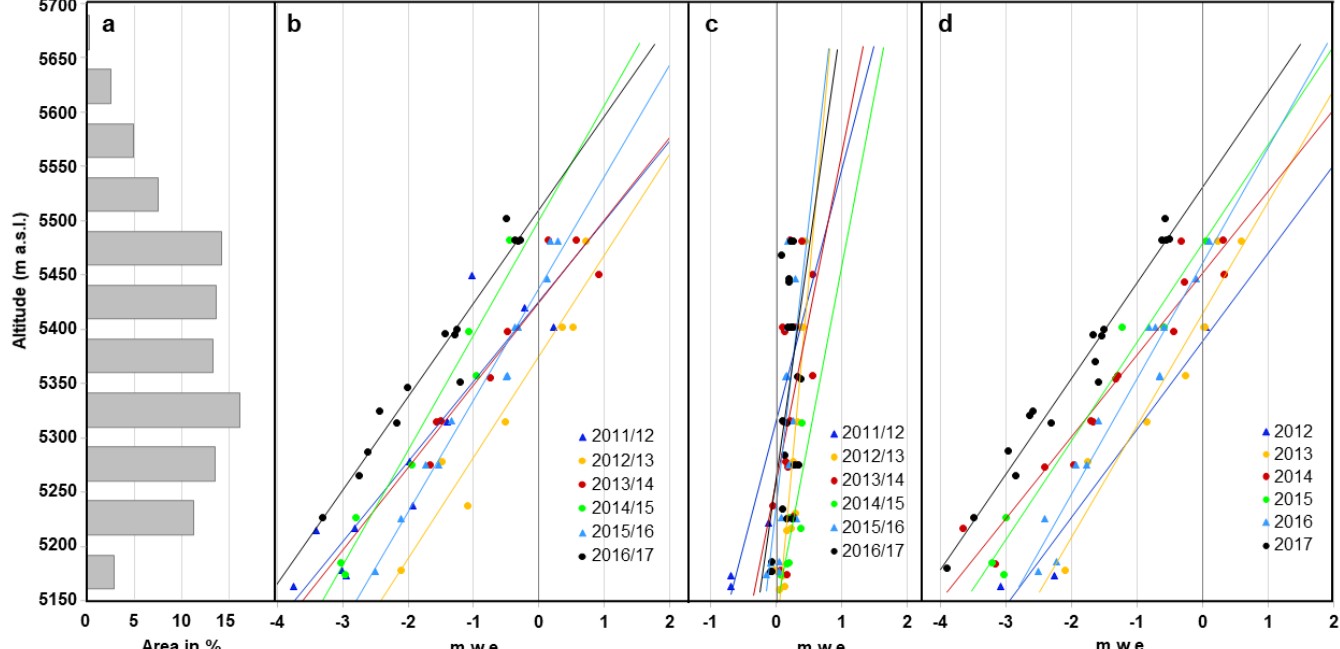

**Figure 2: The glacier hypsography (a), and the mass balances and gradients for the annual, winter and summer mass balance for Yala Glacier from 2011–2017 (b-d).**

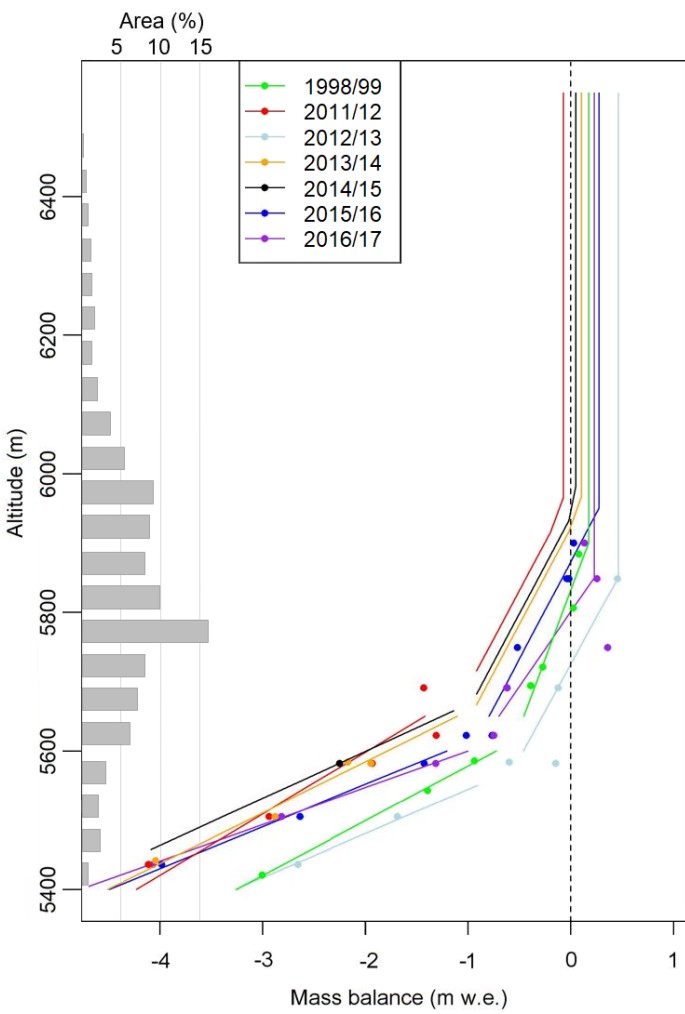


**Figure 3: Point mass balance, gradients and hypsography of Rikha Samba Glacier for the mass-balance years 1998/99, and 2011/12 to 2016/17.**

The seasonal mass balances on Yala Glacier are shown in Table 2 and Fig 4. The average winter and summer balances were 0.29 m and -0.87 m w.e. with standard deviations of 0.14 m w.e. and 0.56 m w.e., respectively. The winter balance is low in

most years, except in 2014/15 when the accumulation was very positive (0.54 m w.e.). The summer balance of 2017 is the most negative balance (-1.75 m w.e.) followed by the summer balances of 2015 and 2014 (-1.12 m and -0.99 m w.e.). In autumn 2012 we calculated the least negative summer balance (-0.35 m w.e.), based on only three measurements. The extreme precipitation events from the cyclones Phailin and Hudhud in October 2013 and 2014, respectively, contributed to the summer balance. The cumulated winter and summer balances largely sum up to the annual balances, except in 2011/12 and 2014/15

when the cumulated winter and summer balances underestimate the annual mass loss by -0.83 m and -0.59 m w.e. These discrepancies are discussed in section 5.1.2 Seasonal mass balance.

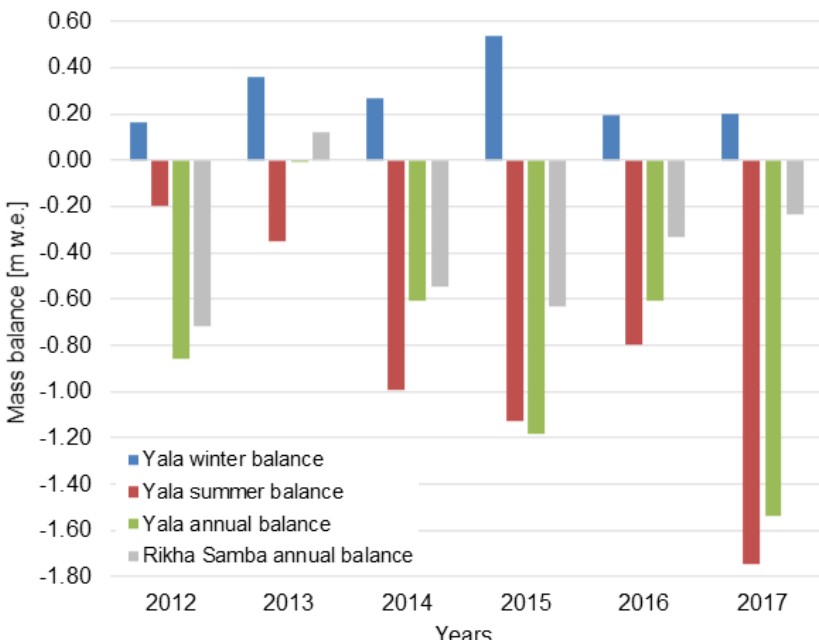

**Figure 4: Winter, summer and annual mass balance of Yala Glacier and annual balance of Rikha Samba Glacier, calculated based**
**on the respective gradients. In the mass-balance years 2011/12 and 2014/15, the sum of winter and summer balances differ significantly from the annual balances, likely due to a lack of data in higher elevations. The summer balance from 2011/12 and winter balance from 2014/15 have not been reported to the WGMS.**

The uncertainties in the accumulation area are larger than in the ablation area because the processes in the snowpack are more complex, influence each other and are difficult to measure (Fig. S3, S4, S5 and S6). The possible causes for these variations
are manifold, from snow/firn compaction, spatial variability of the glacier surface due to varying accumulation and ablation, sawdust promoting local melt, bamboo stakes being pushed up or down and superimposed ice. In some years, the surface roughness was very large in the ablation area, resulting in large errors. Errors for the density of metamorphed snow tended to be larger than for fresh snow because it was harder to measure. At Yala Glacier, the error was largest in the highest elevation bands that make up 15 % of the glacier area because the lack of measurements prevented the calculation of a reliable gradient
in the accumulation area. Similarly, at Rikha Samba Glacier, the sparse measurements in the accumulation area and particularly in its steep slopes (36 % of the area) resulted in large errors that were difficult to estimate.

At Yala Glacier, the measured average densities with standard deviation for snow and firn were 336 kg m$^{-3}$ ($\pm$81 kg m$^{-3}$) and 562 kg m$^{-3}$ ($\pm$128 kg m$^{-3}$). However, harder firn layers were difficult to measure. Dependent on the site and firn conditions, and based on snow-pit profiles and field observations we estimated firn densities between 550 kg m$^{-3}$ and 700 kg m$^{-3}$. At Rikha
Samba Glacier, the average snow density measured was 399 kg m$^{-3}$ with a standard deviation of $\pm$70 kg m$^{-3}$. For ice we assumed a density of 900 kg m$^{-3}$ (Cogley et al., 2011).

The calculated balanced-budget equilibrium-line altitude (ELA$_0$) and balanced-budget accumulation-area ration (AAR$_0$) for Yala and Rikha Samba glaciers are 5378 m a.s.l., 5758 m a.s.l., 0.49 and 0.66, respectively (Fig. 5). From 2011 to 2017 the ELA ranged at Yala Glacier between 5380 m and 5510 m a.s.l. with uncertainties of $\pm$20 m to $\pm$40 m, and at Rikha Samba

Glacier between 5724 m and 5872 m a.s.l. with uncertainties of ±20 m to ±50 m (Fig. 2 and 3, Table 2 and 3). The AAR ranged

from 0.13 to 0.48 and from 0.41 to 0.75 for Yala and Rikha Samba glaciers, respectively.

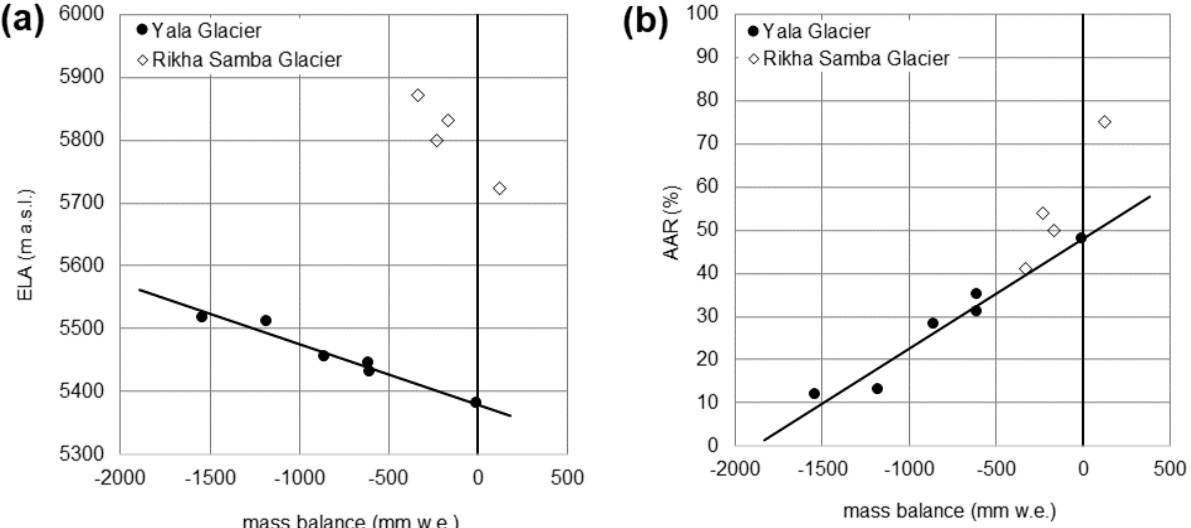

**Figure 5: The ELA (a) and AAR (b) of Yala and Rikha Samba glaciers against the mass balance. The $ELA_0$ and $AAR_0$ for the glaciers are 5377 m a.s.l., 5760 m a.s.l., 0.49 and 0.66 for Yala and Rikha Samba glaciers, respectively.**


The point mass balances are shown in Fig. 2 and 3 as function of elevation and with linear regression lines that are used to derive the mass-balance gradients for Yala and Rikha Samba glaciers, respectively. For Yala and Rikha Samba glaciers, the mean mass-balance gradient at the ELA are 1.04 m and 0.36 m w.e. (100 m)$^{-1}$, respectively (Table 2 and 3). The gradients show a relatively low interannual variability with standard deviations of 0.12 m and 0.9 m w.e. (100 m)$^{-1}$, respectively. In the

lower part of Rikha Samba Glacier, the gradient is much larger with a mean value and standard deviation of 1.48 m and 0.25 m w.e. (100 m)$^{-1}$. Figure 2 shows the characteristic gradients for the annual and seasonal balances of Yala Glacier that remain relatively constant over the investigated time period. However, additional measurements in higher elevations would have allowed to identify a gradient in the accumulation area for the annual and the summer balance. For the winter balance, a small gradient was identified, which is overestimated for years when ablation already set in on the lower part of the glacier.

This is the case for the year 2011/12 when ablation possibly set in earlier, and 2014/15 when the stakes were measured a month later than normally, and in both years without measurements in higher elevations. For these years, the winter mass-balance gradient in the accumulation area is likely smaller than in the ablation area and generally the mass balance is overestimated above about 5500 m a.s.l.

In autumn, often only a very fresh layer of snow was clearly detectable over the entire glacier, and in some years the sawdust

marking the reference surface was removed by ablation before accumulation. Distinct snow and ice layers we identified only after some winters, such as in April 2017 (Fig. 6). In April 2017, sawdust at the bottom of the snow pits or the glacier ice indicated the reference surface. Without the sawdust marking, the lowest layer of darker coarse snow could have been mistaken for snow from the monsoon season. The amount of snow accumulation depended mainly on the elevation, but also aspect, slope and exposure. Maximum accumulation we typically measured at stake S7, which is less exposed than the stakes S6 and

S8. In April 2014, we measured superimposed ice, which formed at the glacier surface below the snow from the cyclone Phailin.

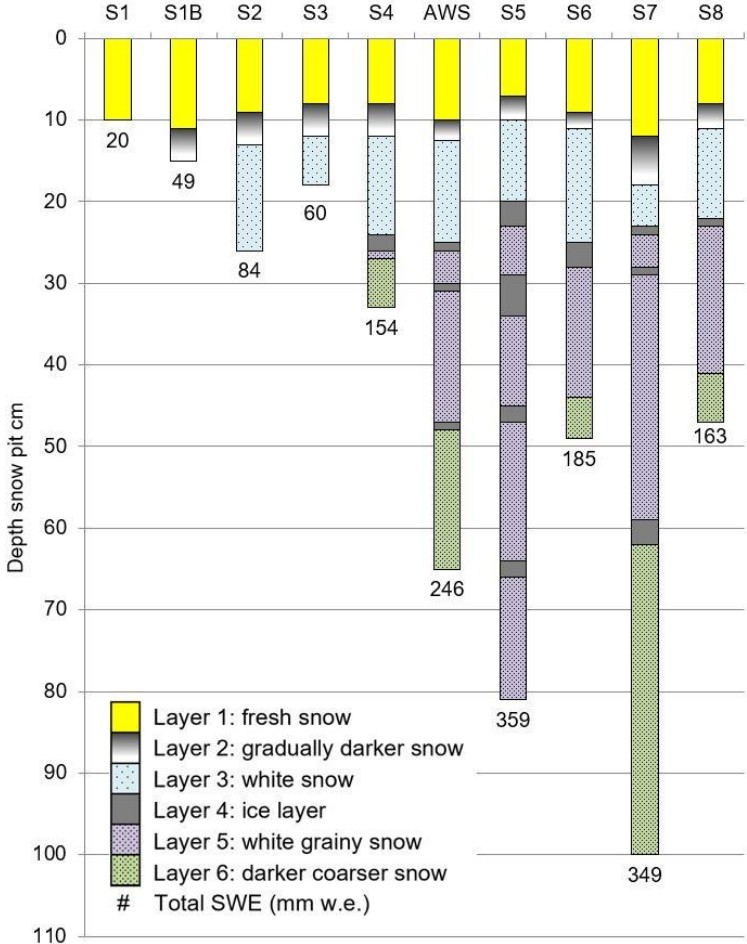

**Figure 6: Snow profiles measured at the stakes on Yala Glacier on 23, 24 and 25 April 2017. At the site AWS, a temporary weather station was set up near stake S4. Distinct snow layers can be identified at all measurement sites. At the stakes S5, S6, S7, and S8 sawdust from 19 and 20 November 2016 was found at the bottom of the snow pit, and glacier ice at all lower sites.**

During the twelve-year period (2000–2012) Yala Glacier's average thinning was -10.49 ±7.41 m, with an annual thinning rate of -0.87 ±0.62 m a$^{-1}$, which corresponds to a total mass loss of -8.92 ±6.33 m w.e., and an annual rate of -0.74 ±0.53 m w.e. a$^{-1}$ (Fig. 7). The mean thinning rate along the profile line was -1.1 ±0.13 m a$^{-1}$. A maximum thickness gain of 17.63 m was measured below ice cliffs, and the biggest ice wastage was measured above the lake and along the glacier terminus, with a value of -50.66 m w.e. Positive mass-balance values lie in the upper part of the glacier. However, when averaging the values over elevation bands, we see a mass gain only in the highest elevation bands, which is filled with the mode value from the accumulation area (Fig. 8). From 2011 to 2017, Yala Glacier's cumulative balance and mean annual rate were -4.80 ±0.69 m w.e. and -0.80 ±0.28 m w.e. a$^{-1}$, respectively. From 2000 to 2017, Yala Glacier lost -12.86 m w.e. with an annual rate of -0.76 ±0.53 m w.e. a$^{-1}$. Rikha Samba Glacier lost -2.34 ±0.79 m w.e from 2011 to 2017, with an annual rate of -0.39 ±0.32 m w.e. a$^{-1}$.

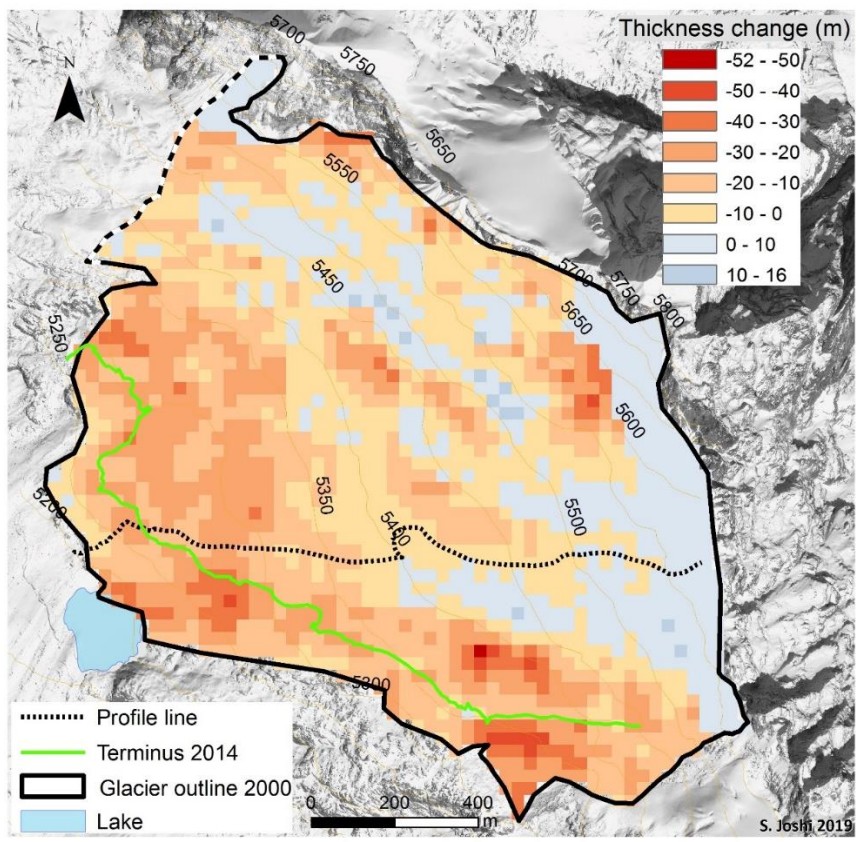

**Figure 7: Thickness changes of Yala Glacier in metres after DEM differencing of GeoEye-1 (Jan 2012) and SRTM3 (Feb 2000) DEM and dGNSS profile in 2012.**

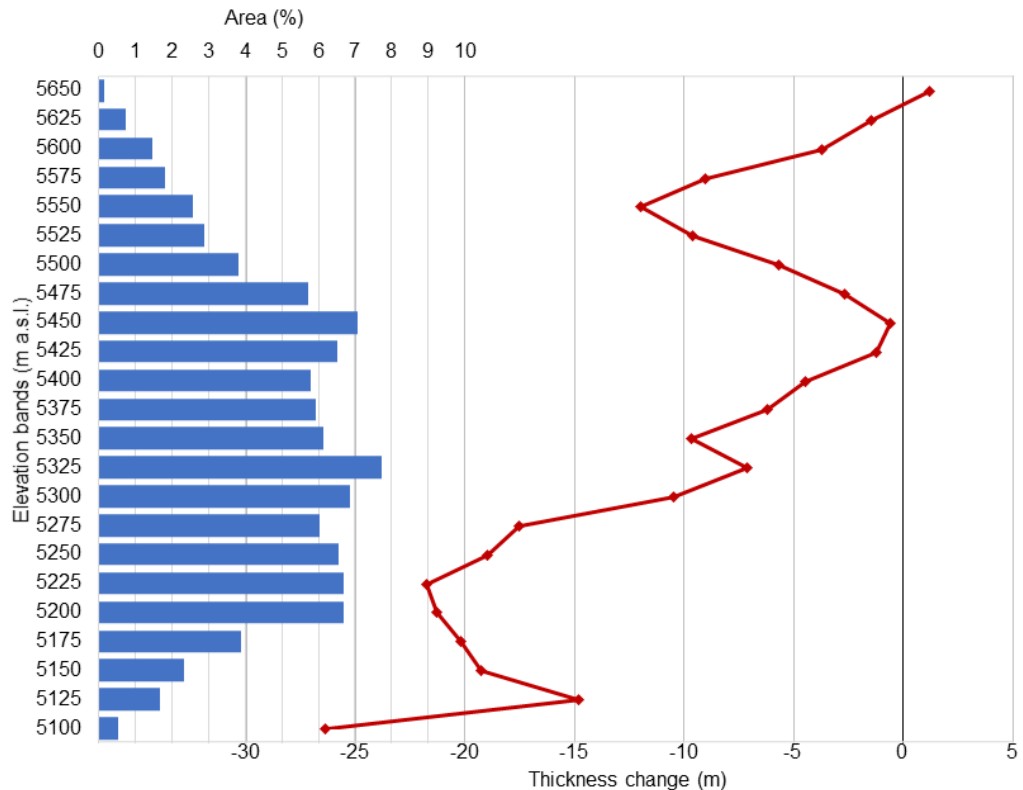

**Figure 8: The mean thickness changes of Yala Glacier for 25 m elevation bands with hypsography, from 2000 to 2012. The reduced thickness change at an elevation of 5125 m is likely a result of the thinner ice thickness in the steeper part of the glacier in 2000. The increased thinning between 5525 and 5575 m a.s.l. might be caused by increased ablation at steep slopes and ice cliffs.**


**4.2 Glacier length changes and flow**

The glacier length changes for Yala and Rikha Samba glaciers are reported in Table 4 and displayed in Fig. 9 and 10. Yala
Glacier retreated by 346 m from 1974 to 2016, with an annual rate of -8.2 m a$^{-1}$. The fastest retreat with a rate of -14.1 m a$^{-1}$
happened between 2000 and 2012, when the glacier retreated 169 m over a large rock step behind the lake. The smallest rates
of -3.8 m a$^{-1}$ and -3.9 m a$^{-1}$ were measured from 1974 to 1981 and 2014 to 2016. For Rikha Samba Glacier, between 1989 and
2013 the average retreat rate and total retreat was -18.0 m a$^{-1}$, and 431 m, respectively. We measured maximum retreat rates
of -31.8 m a$^{-1}$ from 2006 to 2011, when the glacier retreated by 159 m. The smallest retreat rates of -12.4 m a$^{-1}$ were measured
during a retreat of 149 m from 1989 to 2001.

**Table 4: Frontal variations of Yala and Rikha Samba glaciers.**

| Time period | Frontal variation (m) | Uncertainty (m) | Annual rate (m a$^{-1}$) | Source |
|---|---|---|---|---|
| **Yala Glacier** | | | | |
| 1974–1981 | -26.9 | ±5 | -3.8 | Hexagon KH-9 / GEN map |
| 1981–2000 | -129.0 | ±31 | -6.8 | GEN map / Landsat 7 |
| 2000–2012 | -169.1 | ±30 | -14.1 | Landsat 7 / dGNSS |
| 2012–2014 | -13.0 | ±1 | -6.5 | dGNSS / dGNSS |
| 2014–2016 | -7.7 | ±1 | -3.9 | dGNSS / dGNSS |
| **1974–2016** | **-45.8** | **±5** | **-8.2** | Hexagon KH-9 / dGNSS |
| **Rikha Samba Glacier** | | | | |
| 1989–2001 | -149 | ±30 | -12.4 | Landsat 4 / Landsat 7 |
| 2001–2006 | -71 | ±30 | -14.2 | Landsat 7 / Landsat 5 |
| 2006–2011 | -159 | ±30 | -31.8 | Landsat 5 / Landsat 5 |
| 2011–2013 | -52 | ±15 | -26.0 | Landsat 5 / dGNSS |
| **1989–2013** | **-431** | **±34** | **-18.0** | Landsat 4 / dGNSS |

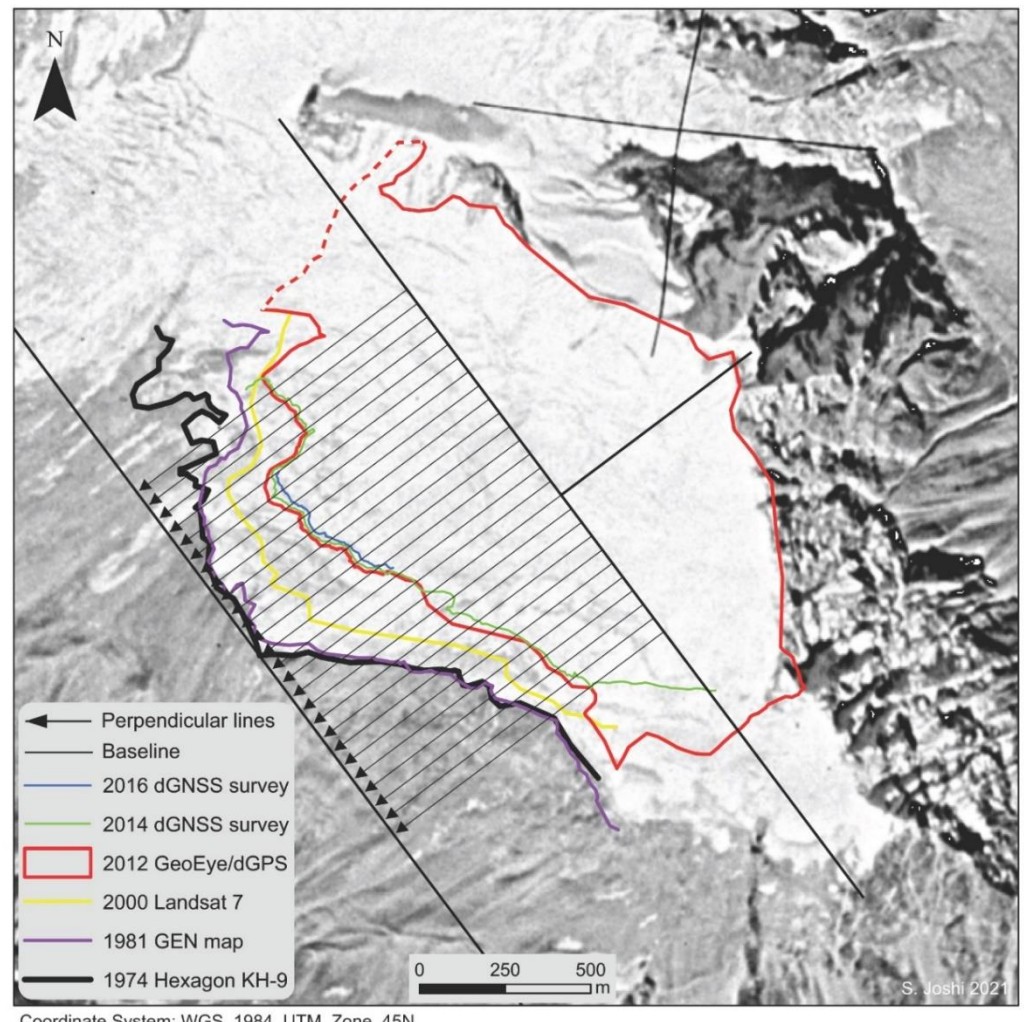

Coordinate System: WGS_1984_UTM_Zone_45N

**Figure 9: Frontal variations of Yala Glacier from 1974 to 2016. The general flow direction is indicated by a straight black line starting at the highest point of the glacier (north-east corner). An arbitrary baseline marks the maximum extent of 1974. Twenty-six parallel arrows in flow direction at 50 m intervals were used to calculate average frontal variations but to exclude outliers only the 9 lines crossing the terminus from 2016 were used for the analysis. The background image is the Hexagon KH-9 from 1974.**

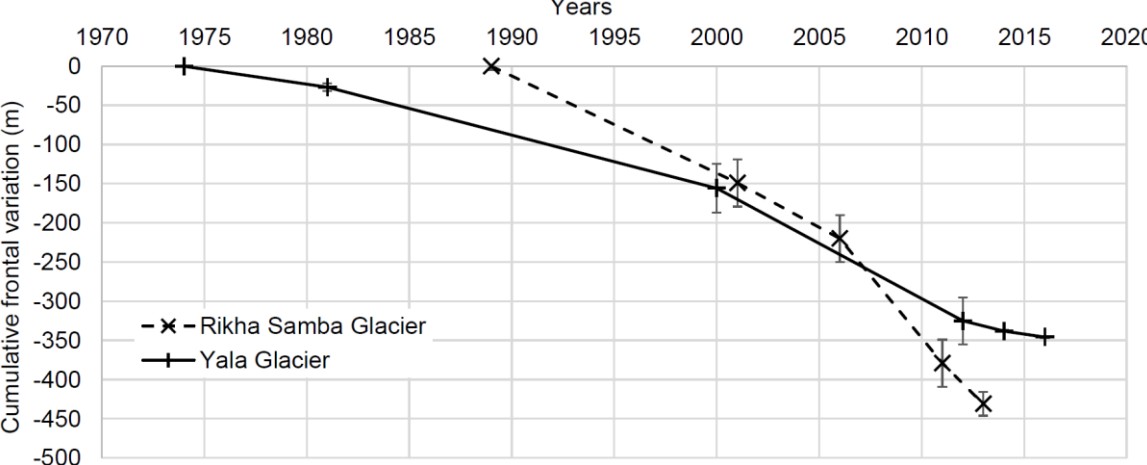

**Figure 10: Cumulative glacier retreat of Yala and Rikha Samba glaciers, with uncertainty range.**

Glacier flow was measured on Yala Glacier between 8 May 2012 and 5 May 2014. The mean horizontal flow was 5.8 ±0.4 m a$^{-1}$, with a minimum and maximum velocity of 4.6 ±0.4 m a$^{-1}$ and 7.8 ±0.4 m a$^{-1}$, respectively (Fig. 11, Table 5). The glacier surface lowered at each measured stake, on average by 3.4 ±0.4 m a$^{-1}$. While reinstalling stakes in the lowest part of the glacier, it was observed that flow velocities were typically less than 5 m a$^{-1}$.


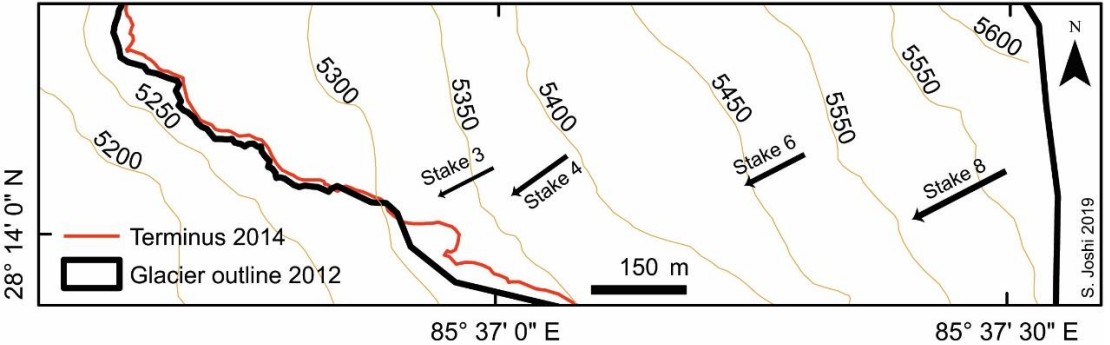

Figure 11: Glacier flow from 8 May 2012 to 5 May 2014 at the stakes 3, 4, 6 and 8, with annual rates between 4.6 and 7.8 m a$^{-1}$. The black arrows show the flow direction and the lengths indicate the annual speed of glacier surface flow, which is depicted 10 times longer than the real flow.


Table 5: Glacier flow in metres and direction of Yala Glacier at the stakes S3, S4, S6 and S8 from 8 May 2012 to 5 May 2014.

| Stake | Horizontal flow (m) | Annual horizontal flow (m a$^{-1}$) | Flow direction | Altitude in 2012 | Altitude in 2014 | Vertical flow (m) | Annual vertical flow (m a$^{-1}$) |
|---|---|---|---|---|---|---|---|
| 3 | 9.1 | 4.6 ±0.4 | S63W | 5249 | 5242 | 7.0 | 3.5 ±0.4 |
| 4 | 11.2 | 5.6 ±0.4 | S56W | 5286 | 5279 | 7.1 | 3.6 ±0.4 |
| 6 | 10.0 | 5.0 ±0.4 | S62W | 5373 | 5366 | 7.1 | 3.6 ±0.4 |
| 8 | 15.6 | 7.8 ±0.4 | S63W | 5457 | 5450 | 6.2 | 3.1 ±0.4 |
| **Average** | | **5.8 ±0.4** | | | | | **3.4 ±0.4** |

**5 Discussion**

**5.1 Yala Glacier**

**5.1.1 Annual mass-balance rates**

Yala Glacier's annual geodetic mass-balance rate is -0.74 ±0.53 m w.e. a$^{-1}$ from 2000–2012 (Table 6). The thinning rate along the profile line is -1.1 ±0.13 m a$^{-1}$, which is higher but within the uncertainty range of the DEM thinning rate, most likely because the mass balance above 5571 m a.s.l. is excluded from the calculation. The profile line has been surveyed repeatedly, the first time by Sugiyama et al. (2013) in 2009 and in subsequent years by our team. The future analysis of the geodetic mass balances along the profile lines and transects is planned as supporting and independent method for the analysis of the mass 575 balance (Wagnon et al., 2020, 2013). The average annual rate of the in situ mass balance from 2011 to 2017 is -0.80 ±0.28 m w.e. a$^{-1}$, which is larger than the geodetic mass-balance rate from 2000 to 2012. From 2000 to 2017, Yala Glacier lost -12.86 m w.e. with an annual rate of -0.76 ±0.53 m w.e. a$^{-1}$. For Yala Glacier, Ragettli et al. (2016) calculated a mass-balance rate of -0.76 ±0.24 m w.e. a$^{-1}$ from DEM differencing for 2006 to 2015, which is within the uncertainty range calculated in this study. Brun et al., (2017) calculated an annual geodetic mass-balance rate of -0.47 ±0.25 m w.e. a$^{-1}$, from 2000 to 2016, 580 which is lower than what we measured, but withing the uncertainty range. Fujita and Nuimura (2011) and Sugiyama et al. (2013) calculated geodetic mass-balance rates of -0.80 ±0.16 m and -0.64 ±0.20 m w.e. a$^{-1}$, respectively, from 1996 to 2009, which are within the uncertainty ranges but for different time periods. Based on a modelling study Fujita and Nuimura (2011) find that Yala Glacier will disappear over time.

**Table 6: Comparison of glacier surface lowering and in situ mass-balance measurements from various studies. Conversions of thickness change (\*) calculated assuming a density of 850 kg m$^{-3}$ and annual uncertainties calculated based on authors' values and Zemp et al. (2013).**

| Duration | Total years | Glacier | Annual thickness change (m a$^{-1}$) | Annual mass-balance rate (m we a$^{-1}$) | Method | Source |
|---|---|---|---|---|---|---|
| 2000–2012 | 12 | Yala | -0.87 ±0.62 | -0.74 ±0.53* | DEM differencing | This study |
| 2000–2012 | 12 | Yala profile | -1.1 ±0.13 | -0.94 ±0.11* | DEM differencing | This study |
| 2011–2017 | 6 | Yala | | -0.80 ±0.28 | Glaciological method | This study |
| 2006–2015 | 9 | Yala | -0.89 ±0.23 | -0.76 ±0.24 | DEM differencing | Ragettli et al., (2016) |
| 2000–2016 | 16 | Yala | -0.56 ±0.30 | -0.47 ±0.25* | DEM differencing | Brun et al., (2017); WGMS 2021 |
| 1996–2009 | 13 | Yala profile | -0.75 ±0.24 | -0.64 ±0.20* | dGNSS and GPR Survey | Sugiyama et al., 2013 |
| 1996–2009 | 13 | Yala | | -0.80 ±0.16 | DEM differencing | Fujita and Nuimura, 2011 |
| 2006–2015 | 9 | 7 glaciers in Langtang | -0.45 ±0.18 | -0.38 ±0.17 | DEM differencing | Ragettli et al. (2016) |
| 2000–2016 | 16 | 3 glaciers in Langtang | | -0.58 ±0.08 | DEM differencing | Maurer et al., 2019 |
| 2011–2017 | 6 | Rikha Samba | | -0.39 ±0.32 | Glaciological method | This study |
| 2000–2016 | 16 | Rikha Samba | -0.44 ±0.27 | -0.37 ±0.23* | DEM differencing | Brun et al., (2017); WGMS 2021 |
| 1998–2010 | 12 | Rikha Samba | | -0.48 ±0.10 | DEM differencing | Fujita and Nuimura, 2011 |
| 2011–2017 | 6 | Mera | | -0.31 ±0.17 | Glaciological method | Wagnon et al., 2020 |
| 2011–2017 | 6 | Pokalde | | -0.75 ±0.28 | Glaciological method | Wagnon et al., 2020 |
| 2000–2011 | 11 | Everest Region | | -0.26 ±0.13 | DEM differencing | Gardelle et al., 2013 |
| 2000–2008 | 8 | Everest Region | | -0.45 ±0.60 | DEM differencing | Nuimura et al., 2012 |
| 2002–2007 | 5 | Everest Region | | -0.79 ±0.52 | DEM differencing | Bolch et al., 2011 |
| 2011–2017 | 6 | Chhota Shigri | | -0.43 ±0.40 | Glaciological method | Mandal et al., 2020 |
| 2000–2016 | 16 | Himalayan glaciers clean | | -0.38 ±0.08 | DEM differencing | Maurer et al., 2019 |

### 5.1.2 Seasonal mass balance

On Yala Glacier the negative summer balance determines the annual balance. For every winter season we measured positive mass balances, and during summer only little or no net accumulation in higher elevations (Fig. 2, 4 and Table 2). The slight mass gain in winter mainly happened between January and May when snowfall set in. In early October 2013 and 2014, the Central Himalayas received large amounts of precipitation brought by the cyclones Phailin and Hudhud (Shea et al. 2015b; Necker et al., 2015). These precipitation events in the form of snow contributed to the summer balance since the measurements

were taken after the cyclones passed, making the summer balance less negative.

In winter 2014/15 an exceptional amount of precipitation was measured at various AWSs. Local people in Langtang reported many Yaks dying in the snow, and during the Nepal earthquake in April 2015 extreme avalanches with anomalous amounts of snow were triggered (Fujita et al., 2017). For this winter, above average accumulation (0.54 m w.e.) was measured and calculated, despite a delay of measurements by a month in early June. Triggered by the earthquake and aftershocks, the

Langtang Valley was heavily affected by snow and ice avalanches, landslides, as well as rockfalls on the glacier in the immediate vicinity of the study area (Kargel et al., 2016; Fujita et al., 2017). Direct effects of the earthquake on the glacier could not be measured, however AWSs on and near the glacier were destroyed likely because of air blasts from ice avalanches. The effect of the air blasts on the snow cover of Yala Glacier is not known, however, it is possible that snow was blown away

and partly sublimated. The air in the valley was filled with dust and it is probable that more dust than usual settled on Yala Glacier, increasing ablation particularly in summer 2015. The seasonal mass-balance measurements in June 2015 were taken under precarious conditions, and only stake measurements could be taken up to an elevation of 5217 m a.s.l., resulting in a higher uncertainty for the seasonal mass balances in 2014/15 and a possibly underrepresented accumulation in winter 2014/15. Hence, the winter balance for the mass-balance year 2014/15 has not been reported to the WGMS. These circumstances explain the discrepancy in the cumulative seasonal and the annual mass balance by -0.59 m w.e. in the mass-balance year 2014/15 (Fig. 4). In autumn 2012, we calculated the least negative summer balance (-0.35 m w.e.), based on only three measurements and likely underestimating ablation. This could explain the underestimated annual mass loss of -0.83 m w.e. in the cumulative seasonal balance compared to the annual balance of 2011/12. Consequently, the summer balance from the mass-balance year 2011/12 has not been reported to the WGMS. Measurements taken in autumn were generally more reliable because less snow was present on the glacier surface, reducing the uncertainty related to the snow cover. Although Yala Glacier is a summer-accumulation-type glacier, most of the accumulation was measured in the winter season because the accumulation area is too small and at a too low elevation to benefit from snowfall during the monsoon months. Together with the overall negative balances it indicates that Yala Glacier is out of balance and shrinking.

### 5.1.3 Glacier length changes, flow and downwasting

At Yala Glacier, Ono (1985) dated Little Ice Age moraines and documented annual ice push moraines, and Yamada et al. (1992) and Kappenberger et al. (1993) observed terminus retreat since the 1970s with a minor advance in the early 1980s and stagnation, respectively, followed by retreat. In the 1990s Fujita et al. (1998) noted an accelerated retreat. From 2000 to 2012, we measured the highest retreat rate of -14.1 m a$^{-1}$ when the glacier retreated over a steep rock step from about 5100 m to 5175 m a.s.l. From 2012 to 2016, Yala Glacier retreated with a slower annual rate of -5.2 m a$^{-1}$ in mostly flat terrain, partly in shallow water.

Horizontal flow was measured with a theodolite from 28 September to 27 October 1982 (Ageta et al., 1984), and from 22 May to 7 October 1996 (Fujita et al., 1998) and a decreasing velocity was observed (Fig. 12). In both studies, the annual flow rate was assumed to be the same as for the measurement periods, despite varying seasons. Sugiyama et al. (2013) measured the top three stakes on 26 September 2008 and 31 October 2009, and the lower two stakes for four days from 31 October to 4 November 2009 with dGNSS, which were presumably extrapolated to calculate the annual rate, assuming a constant flow. The flow velocity and direction measured in this study from 2012 to 2014 compares to the measurements from 2008 to 2009. However, the glacier flow is slower than in the 1980s and 1990s, and the direction slightly varied, as already shown by Sugiyama et al. (2013).

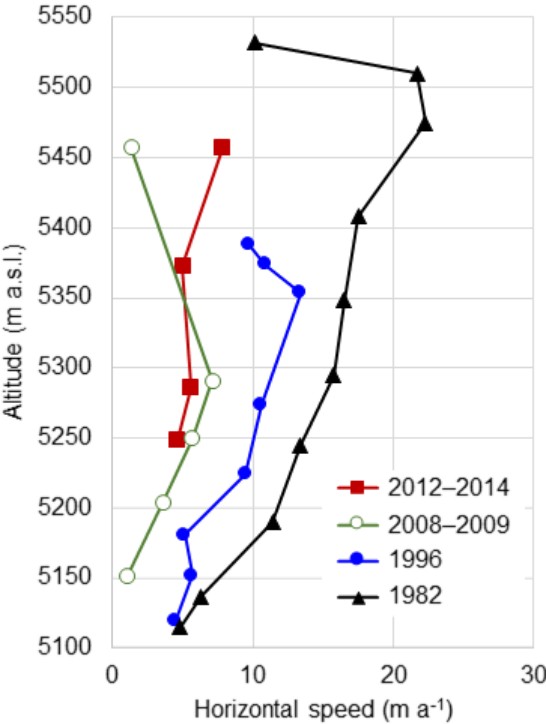

**Figure 12: Altitudinal distribution of the surface flow speeds of Yala Glacier, surveyed in 1982 by Ageta et al. (1984), 1996 by Fujita**
**et al. (1998), 2008 to 2009 by Sugiyama et al. (2013) and from 2012 to 2014 in this study.**

From 2011 onwards, we observed that concave shapes on Yala Glacier's surface have become more pronounced, and that the glacier surface was downwasting, as observed at other glaciers (Ragettli et al., 2016; Sommer et al. 2020). Both the downwasting and enhanced concave shapes are a consequence of the decreased ice velocities, and indicate changes in the

glacier dynamics. The downwasting of Yala Glacier can affect the mass balance and its monitoring in several ways, such as locally enhanced ablation and compromised representativeness of stake measurements. Ablation can be locally enhanced in bowl-shaped areas, where radiation is reflected, resulting in a positive feedback and higher ablation than in the surrounding area (Hock, 2005). Such concave surfaces with transitions to steep slopes became more pronounced, for example, between stakes S1 and S1B and near S5. Usually, stakes represent a characteristic type of glacier area. However, the representativeness

of stake measurements is compromised over time when the glacier surface topography changes from an even surface to a very concave surface with steep slopes. The bias induced by reduced stake representativeness should be corrected later with help of complementing geodetic mass-balance analyses for the same timeframe (Zemp et a., 2013).

### 5.1.4 Steep slopes and ice cliffs

The ice cliffs and steep slopes at Yala Glacier are mainly exposed to south-west, occur over the entire glacier range, and likely

experience increased melt due to their orientation and large surface area. Already Ageta et al. (1984) described the ice cliffs, and old photos document part of the glacier terminus as ice cliff, at times with an apron (Shiraiwa, 1993). The effect of vertical ablation through melt, sublimation and ice breaking off could be substantial, as observed at glacier ice cliffs in the Antarctic McMurdo Dry Valleys (Chinn, 1987; Lewis et al., 1999), on Kilimanjaro (Winkler et al., 2010), and debris-covered glaciers (e.g. Sakai et al., 2002; Steiner et al., 2015). However, ice-cliff and steep-slope ablation cannot be quantified with the

conventional glaciological method and ablation might be underestimated. Additionally, it is difficult to quantify the relevance of steep slopes in terms of area because the slope surface area is not well represented in the map view of a DEM, and increases with steepness (Supplement section S4). At Yala Glacier, assessed with a DEM of 30 m resolution, the area of slopes in average steeper than 50° make up 5 % of the total glacier area in map view. But these steep slopes only represent slopes of at least 36 m height (Table S4 and S5, Fig. S8), and the actual surface area exposed to ablation is much larger than represented by the

DEM (Table S6 and Fig. S9). Analysed with the SRTM-3 DEM, Bajracharya et al. (2014) found that more than 50 % of the glacier area in Nepal is oriented south-west, south, or south-east. Yet, to quantify steep slopes with a DEM with a resolution of 90 m, slopes with angles equal or larger than 48° must have a minimum slope height of 100 m, and steeper slopes of smaller height cannot be represented (Table S4 and S5, Fig. S8). Hence, the surface area of Nepal's ice cliffs and steep ice slopes is underrepresented and cannot be quantified in such DEM analyses.

Complementing geodetic mass-balance measurements for the same timeframe help to correct the glacier-wide annual mass balances of Yala Glacier for biases such as introduced by steep slopes and ice cliffs (Zemp et al., 2013; Wagnon et al., 2020). To better understand and assess specifically the influence of the steep slopes and ice cliffs of the mass balance, geodetic thickness-change analyses based on high-resolution surface elevations for short time intervals could be used, in combination with energy-balance models (Joerg and Zemp, 2014).

**5.2 Rikha Samba Glacier**

For Rikha Samba Glacier, Fujita and Nuimura (2011) and Brun et al. (2017) calculated geodetic mass-balance rates of -0.48 m w.e. $a^{-1}$ (1998–2010) and -0.37 ±0.23 m w.e. $a^{-1}$ (2000–2016). These values are close to the annual average rate of -0.39 ±0.32 m w.e. $a^{-1}$ (2011–2017) calculated in this study, however, the time periods vary and Fujita and Nuimura (2011) largely excluded elevations above 6000 m a.s.l. From 1974 to 1994, Fujita et al. (2001) measured a retreat of 216 m with a slow retreat

rate of -10.8 m $a^{-1}$. The rate accelerated to -18.2 m $a^{-1}$ from 1994 to 1998 when the glacier retreated 73 m. From 1989 to 2006, we measured a glacier retreat of 220 m in total, with retreat rates of -12.4 m $a^{-1}$ and -14.2 m $a^{-1}$ from 1989 to 2001 and 2001 to 2006, respectively (Table 4, Fig. 10). From 2006 to 2011 and 2011 to 2013 the terminus retreated rapidly by 159 m and 52 m, with rates of -31.8 m $a^{-1}$ and -26.0 m $a^{-1}$, respectively. In a modelling study, Fujita and Nuimura (2011) found that Rikha Samba Glacier will not disappear under the current climate.

**5.3 Comparison of in situ glacier mass balances in the Himalaya**

In Nepal, the mean annual mass-balance rates of the small low-lying Yala and Pokalde glaciers (Fig 1) from 2011 to 2017 are similar (-0.80 ±0.28 m and -0.75 ±0.28 m w.e. $a^{-1}$, Table 6). Rikha Samba and Mera glaciers are both higher lying glaciers with a larger elevation range and smaller mass-balance rates (-0.39 ±0.32 m and -0.31 ±0.17 m w.e. $a^{-1}$; Wagnon et al., 2020). These tendencies are reflected in the cumulative mass balances that are negative for Mera and Rikha Samba glaciers, and even

more negative for Yala and Pokalde glaciers (Fig. 13). Mera Glacier has a large elevation range (4940–6420 m a.s.l.) and similar upper limits as Rikha Samba Glacier (5416–6515 m a.s.l), but a lower $ELA_0$ (~5515 m a.s.l.), and a large accumulation area with an $AAR_0$ of about 0.60. Rikha Samba Glacier has a smaller elevation range (1100 m vs. 1460 m), and a smaller average mass-balance gradient at the ELA than Mera Glacier (0.36 m vs. 0.45 m w.e. $(100 m)^{-1}$), which indicates the more continental conditions on the north side of the Himalayan main divide, opposed to Mera Glacier on the south side of the main

divide. Fujita and Nuimura (2011) calculated so-called preferable ELAs for the glacier extents of Yala and Rikha Samba glaciers in 2009 and 2010, which are 5260 m and 5545 m a.s.l., respectively, and are lower than the calculated $ELA_0$ of 5378 m and 5758 m a.s.l. in this study. Varying glacier areas and elevation ranges are likely reasons for the differences.

In winter, wind and sublimation are important ablation processes on the glaciers. Wagnon et al. (2013) address the high wind speeds from westerly winds at Mera Glacier (5360 m a.s.l on glacier AWS) in winter, which in combination with sublimation

causes a substantial part of the winter ablation. Stitger et al. (2018) and Litt et al. (2019) assessed sublimation on Yala Glacier and confirm its strong ablating influence, especially during favourable conditions such as high wind speed, low atmospheric vapour pressure and low near-surface vapour pressure. The study of Shea et al. (2015b) shows similarly high winter wind speeds at Rikha Samba Glacier (5310 m a.s.l, off-glacier AWS) as at Mera Glacier, but at Yala Glacier (5060 m a.s.l., off-glacierAWS) only slightly higher wind speeds than on annual average. It seems reasonable that wind and sublimation are

important ablation processes for Rikha Samba Glacier in winter. At Yala Glacier, in winter when accumulation dominates over

ablation the effect of wind and sublimation is probably smaller compared to Mera and Rikha Samba glaciers. Fujita et al. (1997b) point out that winter precipitation is more important in Langtang than in Khumbu, which is confirmed by the AWS data described by Shea et al. (2015b) and could partly explain the winter accumulation on Yala Glacier. Shiraiwa (1993) highlights the influence of both the summer monsoon and westerly winter circulation on the annual balance. To better understand the relationship between the climate and the mass balance of Yala and Rikha Samba glaciers, the analysis of homogenised climate data from nearby weather stations or reanalysis data would be useful.

Chhota Shigri Glacier (Fig. 1) is a glacier in the Western Himalaya under the influence of the Indian summer monsoon in summer, and western disturbances in winter, with a relatively long in situ mass-balance series (Mandal et al., 2020). The cumulative mass balance and the annual mass-balance rate of the glacier (-2.59 m w.e. and -0.43 ±0.40 m w.e.a$^{-1}$) from 2011 to 2017 are in a similar range like Rikha Samba and Mera glaciers. Chhota Shigri Glacier also has a large elevation range of about 1760 m, but lies on a lower elevation (4072 m to 5830 m a.s.l.). The mean ELA and AAR of 5047 m a.s.l. and 0.49, respectively, indicate that Chhota Shigri Glacier is relatively healthy despite the lower elevation range, due to the colder climate and winter precipitation from westerly disturbances.

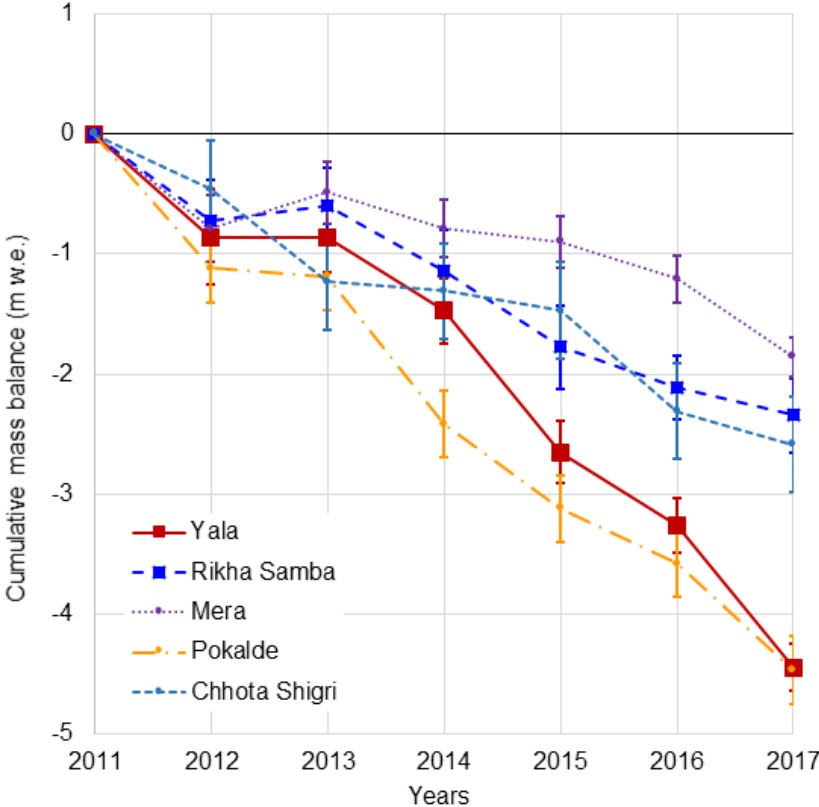

**Figure 13: Cumulative mass balances of Yala, Rikha Samba, Mera, Pokalde and Chhota Shigri glaciers. The data for Mera, Pokalde and Chhota Shigri glaciers is from Wagnon et al. (2013), Sherpa et al. (2017), WGMS (2021), Wagnon et al. (2020) and Mandal et al. (2020).**

**5.4 Bias by small low-lying glaciers**

Yala and Pokalde glaciers are low-lying glaciers with a small elevation range, and demonstrate a bias towards negative mass balances in terms of representativeness for the mass balance of a region. Both glaciers are small, lie on a low altitude with a small elevation range (Yala: 5168 m–5661 m, Pokalde: 5430 m–5690 m a.s.l.) similar to AX010 Glacier in the Shorong Himal, Nepal (Fig. 1), and are very sensitive to temperature especially in the monsoon season (Fujita and Nuimura, 2011; Ragettli et al., 2016). Immerzeel et al. (2012), found that from 1957 to 2002 in Langtang 77 % of precipitation fell between June and September, and Ageta and Higuchi (1984) reported about 80 % of the annual precipitation in the same months for east Nepal. Shea et al. (2015b) estimated the height of the 0° C isotherm in Langtang between 3000 m a.s.l. in winter and 6000 m a.s.l. during monsoon. Hence, glaciers at lower altitudes receive precipitation predominantly in the form of rainfall during the

monsoon season and snow accumulation is minimal. The very negative balances of Yala and Pokalde glaciers can be explained by the small amount of accumulation during the main precipitation season in monsoon.

In comparison, Ragettli et al. (2016) calculated a balanced geodetic mass balance of -0.02 ±0.13 m w.e. a$^{-1}$ for the debris-free Kimoshung Glacier (Fig. 1) in close vicinity of Yala Glacier about 3.5 km away, and explain the difference with the very different hypsometry. Compared to Yala Glacier, Kimoshung Glacier has a steep narrow tongue and a large accumulation area (AAR of 0.86) at high altitude, which is less exposed to air temperatures above 0° C and making the glacier less sensitive to temperature. The accumulation area is probably sheltered from strong westerly winter winds by a mountain ridge running from north-west to south-east, reducing ablation by wind and sublimation, but receiving precipitation largely in the form of snow.

Geodetic mass-balance analyses from the Himalayan region show heterogenous patterns, with average values less negative than for Yala Glacier, although mostly within the uncertainty ranges. Ragettli et al. (2016) assessed the geodetic mass balances of two clean and five debris-covered glaciers in Langtang and found a very heterogeneous distribution and a mean annual mass-balance rate of -0.38 ±0.17 m w.e. a$^{-1}$ from 2006 to 2015, which is lower than Yala Glacier's annual geodetic mass-balance rate of -0.74 ±0.53 m w.e. a$^{-1}$ from 2000 to 2012. Maurer et al. (2019) calculated a median geodetic balance of about -0.54 m w.e. a$^{-1}$ for the clean glaciers in a subregion including Langtang, and a mean rate of -0.58 ±0.08 m w.e. a$^{-1}$ for three debris covered glaciers in Langtang from 2000 to 2016, which is a bit more negative than calculated for the same glaciers by Ragettli et al. (2016). The average geodetic mass-balance rates measured in the Everest Region by Gardelle et al. (2013; 2000–2011: -0.26 ±0.13 m w.e. a$^{-1}$) and Nuimura et al. (2012; 2000–2008: -0.45 ±0.60 m w.e. a$^{-1}$), are lower than measured at Yala Glacier. Bolch et al. (2011) found a slightly higher mass-balance rate (2002–2007: -0.79 ±0.52 m w.e. a$^{-1}$) but within the uncertainty ranges of the other studies. For 18 Himalayan glaciers, Azam et al. (2018) assessed a mean rate of -0.49 m w.e. a$^{-1}$ for directly measured glacier mass balance for the period from 1975 to 2015. Maurer et al. (2019) calculated a Himalayan-wide geodetic mass balance of -0.38 ±0.08 m w.e. a$^{-1}$ for clean ice from 2000 to 2016. The mass-balance rate of Rikha Samba Glacier is within a similar range, however, the one of Yala Glacier is more negative.

The bias introduced by small low-lying glaciers like Yala Glacier result in the overestimation of negative mass balances in the region (Gardner et al., 2013). It highlights the importance of investigating glaciers with large elevation ranges, and measuring mass balances in the accumulation areas and precipitation data in high altitudes.

**5.5 Interannual variability of winter precipitation and long-term trends of accumulation**

Climate data indicate a large interannual variability of winter precipitation but long-term trends of solid and liquid precipitation on high elevations are not well known, and winter mass balances measurements are still rare in the Nepal Himalaya. The interannual variability of winter precipitation is much larger than of summer precipitation, and affects the seasonal mass balances on Yala Glacier. Derived from precipitation data from the Indian Embassy and the Airport in Kathmandu, Seko and Takahashi (1991) found that during the period from 1911 to 1986, winter precipitation (October–April) exceeded summer precipitation (May–September) in 10 years . Since 1985, the interannual variability was largest in the month of October (Fujita et al., 1997b) and extreme snowfall has been reported from cyclones in October for several years, such as in 1985 (Seko and Takahashi, 1991; Iida et al., 1987), Phailin in 2013 (Shea et al., 2015b), Hudhud in 2014 (Neckel et al., 2015), and the 1995 India cyclone in November 1995 (Kattelmann and Yamada, 1996). This precipitation variability has a significant effect on the mass balance of glaciers in the Nepal Himalaya (Seko and Takahashj, 1991). Early or large amounts of winter snowfall protect the glacier from ablation by the high albedo, like the snowfall from the cyclones Phailin and Hudhud in October 2013 and 2014. In early 2015, exceptional amounts of precipitation likely dampened the effects of the extremely negative summer balance with less than average precipitation.

On Yala Glacier positive point mass-balance data from the 1980s and 1990s are more positive than those measured in this study (Fig. 14), but the related precipitation trends are unknown. Positive annual point balances were measured above 5400 m a.s.l. in all years except 2014/15 and 2016/17. Steinegger et al. (1993) measured deposited snow in a crevasse at

5580 m a.s.l. and identified annual layers from 1981 to 1989 based on the dirt layers, and converted them to water equivalent.
Iida et al. (1987) studied snow and dirt layer formation processes, analysed a snow profile at 5333 m a.s.l. and used precipitation data to assign clean and dirt layers to specific periods in the mass-balance years 1983 and 1984. Ozawa and Yamada (1989) and Yamada (1991) evaluated snow profiles from various elevations to calculate the net accumulation for the years 1985/86 and 1986/87, and Yoshimura et al. (2006) retrieved an ice core at 5350 m a.s.l. and identified annual layers from 1984 to 1994 with help of snow algae. Shiraiwa et al. (1992) analysed snow profiles at various elevations, identified surface balances from monsoon 1990 and the following winter balance up to May 1991. Even though the measurements are difficult to compare because of varying methodologies, it can be seen that accumulation was highest in the 1980s, and also measured at lower elevations. In the 1990s the accumulation decreased, however, accumulation was still measured at elevations where in this study no positive balances were measured. The authors of the earlier studies identified annual layers with confidence, and only Iida et al. (1987) discussed additional dirt layers formed after strong winter snowfall events. In this study the accumulation measurements were challenging because often sawdust layers were gone or older layers hard to assign. In the winter snow at Yala Glacier, we often observed white and grey snow layers, with ice lenses or layers in between (Fig. 6). The ice layers and lenses, superimposed ice and occasional ice fingers indicated melt and refreezing processes, which likely already start in March when incoming solar radiation and temperature increase and in April when solar radiation is close to its maximum (Takahashi et al., 1987a; Shea et al., 2015b). Snow from monsoon was usually more metamorphed with darker and coarser grains. Watanabe et al. (1984) reported from April to June melting up to at least 5500 m a.s.l., and an abundance of water from rain and melt in the temperate accumulation area during the Himalayan Glacier Boring Project 1981–1982, which promotes the snow metamorphosis process. In some years we observed icicles hanging from distinct layers in ice cliffs, indicating melt and refreezing processes and impermeable ice layers in the snowpack.

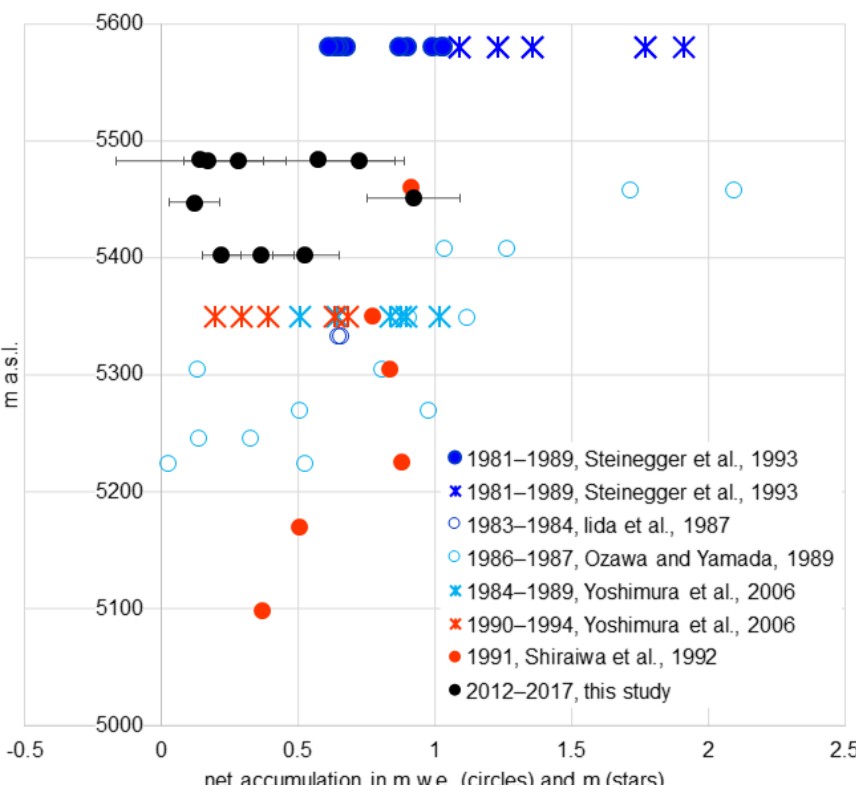

**Figure 14: Positive point mass balances in the accumulation area from mass-balance years in the 1980s (blue), 1990s (red) and from this study (black). The data was compiled from annual snow-pit measurements, multiannual snow profiles, ice cores and crevasses, using dirt, algae or ice layers to distinguish annual layers. Most measurements were converted into water equivalents (circles), and some are only available as snow depth (stars).**

The decreased accumulation over the past decades is likely due to the rising air temperatures, and possibly a decrease in precipitation as observed in the Everest region by Salerno et al. (2015). On the south slopes of Mt. Everest above 5000 m a.s.l., they found that the minimum temperature increased outside of the monsoon season and liquid precipitation decreased significantly from 1993 to 2013. Provided this also applies to other parts of the Central Himalaya, the impact of reduced snowfall could possibly contribute to the negative mass balances of Yala, Rikha Samba and other glaciers.

**5.6 Extrapolation of mass-balance data to unmeasured areas**

In the ablation area of Yala and Rikha Samba glaciers sufficient in situ measurements largely allowed the interpolation of the data by using elevation dependent mass-balance gradient. In the accumulation area, measurements were often challenging and associated with higher uncertainties. The main issues were difficult access, and cumulative ablation that temporarily exceeded the cumulative accumulation (Supplement section S1). On one hand this ablation removed the marked reference surfaces for
the accumulation measurements, and on the other hand the uncertainty is increased for ablation measured with stakes installed in an unstable firn and snow underground. Additionally, no accumulation data could be collected at the highest elevations.

To extrapolate the mass balance to higher elevations, we made a few considerations: the glacier mass-balance programmes were running only within the first decade, and a re-evaluation and possible correction of the glacier-wide mass balance with help of other methods is likely in the future (Zemp et al., 2013; Cullen et al., 2016; Wagnon et al., 2020). Therefore, we chose
simple extrapolation approaches.

At Yala Glacier, extrapolating the ablation gradient to the accumulation area introduced a systematic error for a small glacier area (15 % of the total area) with a small elevation range (~160 m). The largest errors are expected in the highest elevation bands, where accumulation is overestimated (Fig. S3 and S4). At the steep south-west oriented slopes of Yala Glacier, the ablation is likely increased and underestimated in the glacier-wide mass balance. At Rikha Samba Glacier, using the same
extrapolation method like at Yala Glacier would have very much overestimated the accumulation in a large area (36 % of the total area) with a large elevation range (~650 m). Instead, we estimated a fixed value for the accumulation area, which introduced a random error. Geodetic mass-balance analyses complementing in situ mass-balance data for the same time interval help reducing uncertainties and are an integral part of glacier mass-balance programmes following the international glacier monitoring strategy (WGMS, 2020; Haeberli et al., 2000).

**6 Conclusions**

We measured the in situ mass balance of Yala and Rikha Samba glaciers for the mass-balance years 2011/12 to 2016/17. Additionally, we measured the seasonal in situ mass balance of Yala Glacier, and analysed the geodetic mass balance from 2000 to 2012. Glacier length changes have been analysed for both glaciers based on field measurements, maps and satellite images. We conclude:

- Both Yala and Rikha Samba glaciers shrank and retreated in the last couple of decades. The geodetic mass balance of Yala Glacier showed a mass loss of -8.92 ±6.33 m w. e. from 2000 to 2012, at an annual rate of -0.74 ±0.53 m w.e. a$^{-1}$. The cumulative in situ mass balances for Yala and Rikha Samba glaciers were -4.80 ±0.69 m w.e. and -2.34 ±0.79 m w.e., and the annual mass-balance rates -0.80 ±0.28 m w.e. a$^{-1}$ and -0.39 ±0.32 m w.e. a$^{-1}$, respectively. From 1974 to 2016, Yala Glacier retreated by 346 m, and from 1989 to 2013 Rikha Samba Glacier retreated by 431 m. Under
the recent climate it can be expected that Yala Glacier will disappear over time but not Rikha Samba Glacier (Fujita and Nuimura, 2011).

- For both investigated glaciers, the measurements in the ablation area were sufficient to calculate mass-balance gradients. However, a lack of reliable measurements in the high-elevation areas prevented the calculation of accumulation gradients. On one hand, parts of the accumulation areas were not accessible, on the other hand the in

situ measurements in the accumulation area had higher uncertainties. The related uncertainties can be addressed in future with complementing geodetic mass-balance analyses for the same time interval.

- The mass balance of the steep south-west-facing slopes on Yala Glacier could not be measured but have been quantified based on the linear regression equations from the in situ measurements. However, the ablation on steep slopes is possibly underestimated due to the orientation and the steepness of the slopes. This bias can be addressed

with geodetic mass-balance analyses using the same time period as for the in situ measurements. The relevance of the steep glacier slopes in terms of area cannot be quantified neither for Yala Glacier, nor the glaciers in Nepal in general with DEMs of 30 m and 90 m resolution, respectively.

- Yala Glacier experienced downwasting, indicated by the observed changes in the surface topography between 2011 and 2017 and decreasing ice flow velocities. Over the course of the years, most of the stakes could not be reinstalled

at the original coordinates, either because of new crevasses, or significant changes of the surface features at the original site. The downwasting and the small accumulation area at low elevation compromise the long-term monitoring of Yala Glacier.

- The mean annual mass-balance rate of Yala Glacier is more negative compared to regional geodetic mass-balance analyses. The reasons are the small area and elevation range of Yala Glacier and the setting on a low elevation.

The glacier mass-balance programmes for the two glaciers have been designed using a comprehensive monitoring strategy following the international glacier monitoring strategy within GTN-G (WGMS, 2020; Haeberli et al., 2000). Provisions have been made for future geodetic mass balance analyses by acquiring stereo images for DEM generation early on. AWSs at both study sites collect data to further assess the relationship between the mass balance and the climate, and modelling studies are ongoing for Rikha Samba Glacier.

**7    Recommendations**

The mass-balance programmes at Yala and Rikha Samba glaciers are set up for a long-term sustainable continuation. Based on this study we recommend a focus on the following points:

- The long-term monitoring of glaciers with a high and large elevation range is important. Rikha Samba Glacier is such a glacier and its long-term survival is better compared to the small low-lying Yala Glacier.

- More measurements are needed in accumulation areas. At Rikha Samba Glacier measurements up to 6000 m a.s.l. are feasible with the glaciological method. However, at Yala Glacier possibilities are limited.

- Geodetic mass-balance analyses overlapping the time interval of the glaciological measurements of Yala and Rikha Samba glaciers are needed (Zemp et al., 2013). The complementing approach assures keeping the annual signal of the glaciological measurements, and reduce uncertainties introduced for example by unmeasured parts of the

accumulation area or steep glacier slopes.

- The comparison of mass-balance data with climate data is needed to better understand the climate signal of the mass-balance data. Homogenised data from AWSs or reanalysis climate data could be used for that purpose.

**Data availability**

The data are available from the Fluctuations of Glaciers Database http://dx.doi.org/10.5904/wgms-fog-2021-05 (WGMS,
2021). The Supplement contains additional information related to this article.

**Contributions**

DS designed the study and monitoring programme, collected and analysed data, and wrote the manuscript as main author. SPJ collected data and analysed the geodetic mass balance, velocities and frontal variations, and wrote the respective sections. TRG and GS collected and analysed the in situ mass-balance data and contributed to text editing.

**Competing interests**

The authors declare that they have no conflict of interest

**Acknowledgements**

We would like to thank the Government of Norway for supporting and funding this research, as well as ICIMOD's national partners of Nepal, including the Department of Hydrology and Meteorology, Kathmandu University, and Tribhuvan University. This study was partially supported with core funds of ICIMOD contributed by the governments of Afghanistan, Australia, Austria, Bangladesh, Bhutan, China, India, Myanmar, Nepal, Norway, Pakistan, Switzerland, and the United Kingdom. Thanks to all field assistants, trainees and the Trekking Agencies Glacier Safari Treks, Himalayan Research Expeditions and Guides for all Seasons for their support for the glacier measurements and logistic support. We thank Koji Fujita for his support and the dGNSS Magellan ProMark 3 data from May 2012, and the GEN map of Yala Glacier. Thanks to Joe Shea, Inka Koch and Santosh Nepal from ICIMOD and Suresh C. Pradhan from DHM for providing temperature and precipitation information from the Langtang Valley. We thank Christa Stephan for providing the orthomap of Langtang, and Pushpalal Ball from the Department of Survey for help to digitize existing topographic data and maps. We thank Tino Pieczonka, Nicolai Holzer and Tobias Bolch for the training and support on the geodetic mass balance analysis. Thanks to Patrick Wagnon for supporting the field measurements in May 2013 and taking measurements in June 2015 together with Joe Shea, his support for dGNSS measurements and the geodetic profile analysis, and reviewing an earlier version of this article. We thank Matthias Huss for his support for the uncertainty assessment of the glaciological measurements, and Martin Hoelzle for reviewing the article thoroughly and providing support. We thank the scientific editor Reinhard Drews, reviewer Argha Banerjee and an anonymous reviewer for taking the time and help improve the article with constructive feedback.

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
