# Peer review of "Mass balances of Yala and Rikha Samba glaciers, Nepal, from 2000 to 2017"

_Earth System Science Data, 2020_

## Referee Comment (RC1) · Argha Banerjee (Referee) · 25 Dec 2020

Stumm and coauthors present and analyse glaciological mass balance and other re-
lated data measured at Yala and Rikha Samba (RS) glaciers. These data sets spanning
seven balance years may be invaluable contributions towards a better understanding of
the high-Himalayan glaciers and climate. However, there are a few points listed below
where I believe there are some scope for improvements.

**Major comments:**

1. I am concerned about the two very different strategies employed to extrapolate
the winter/annual balance over the upper reaches of the accumulation zones of RS
and Yala glaciers: An elevation-independent constant accumulation rate on RS, and a

constant elevation-gradient based interpolation on Yala. While some discussions are presented with reference to Yala, a more detailed discussion of the rationale behind adpoting these the two methods, and a quantitative analysis of the corresponding uncertainties may be necessary.

As far as I can tell looking at the fig 2 (left most panel), the balance measured at the upper few stakes/pits may be better described with a constant elevation-independent balance rate. (Let me mention here that I am not sure if the present choice of horizontal range for the left-most panel of fig 2, which seriously compromises the readability, is a good one. I request you to please restrict the horizontal range to something reasonable, like [-1,1] m w.e.)

2. I think the paper may benefit from a more careful presentation of the geodetic analysis in general.

While I am not that familiar with intricacies of geodetic techniques involved, I wonder if obtaining horizontal (vertical) location of the GCPs from Landsat (SRTM) products having a relatively lower resolution (and vertical accuracy) may seriously compromise the accuracy of the generated GeoEye-1 DEM that has nominal horizontal resolution of 2-5 m. In general, the rationale behind switching between SRTM1 and SRTM3 for various pieces of analysis done needs to be made clear. Also, why not use SRTM1 here?

Could you not check the accuracy of the DEM making use of any GNSS data from the stable terrain that may be available? A comparison between your May 2012 GNSS elevation profile, and the same extracted from Jan 2012 DEM could be very useful here. I missed such a plot in the present manuscript. In fact, it appears that you do not make much use the highly accurate elevation profile from GNSS survey, apart from comparing it with 2000 SRTM3 data.

It may be useful to plot the distribution of differences between DEM2000 and DEM2012 on the off-glacier stable terrain.

3. You have invoked subtropical jet stream which operates on a planetary scale to explain strong winds on certain glaciers during the winter season while referring to Ding & Sikka (2000) and Wagnon et al. (2013). You have not discussed this issue in depth, but I find it hard to imagine how a planetary-scale system influences two glaciers from the same region in a differnt manner. It seems that the connection between jet stream and winds speed on glacier scale that Wagnon et al. (2013) make is entirely based on observation at a single glacier/location. Also, the authors didn't discuss this issue in that detail. Unfortunately I could not access the article that they referred to or Ding & Sikka (2000) that you refer to, so not sure if these references talk about some relevant mechanism in the context of Himalayan glaciers. To me your data from Yala may instead be a pointer that local effects can have a strong impact on spatiotemporal variability of winds. I believe a one-to-one correspondence between local scale and planetary scale winds will always be tricky - more so with the complex Himalayan topography.

I suggest that you please revise your discussions on the effects of the jet stream on local winds. Please provide sufficient arguments about the corresponding mechanism and/or cite relevant references.

4. If I am not mistaken, the paper refers to more detailed data from the two glaciers than that can be accessed from FoG database (for example, stake displacements, winter balance, digital elevation models etc.). It would be a great if all such data used in the study can be made accessible to the community.

**Minor comments:**

* You have discussed the difficulties in identifying the lower boundary of the annual accumulation of snow in L513 and also in section 5.1. This an important issue that has implications for the data collected in the past on Himalayan glaciers, and may help improve similar measurements in the future. Therefore this point may be highlighted with the corresponding discussions moved to a dedicated subsection within the Discussions section.

[Figure]

* I request a carefully revision the manuscript to edit the language. Please identify and replace/rephrase sentences that are either difficult to understand and/or do not conform to standard English usage. I have tried to point out a few such instances, but there may be a few more.

* Please improve the readability of figure 1. Often the contour lines are all but invisible (particularly for RS). Elevation labels are generally hard to read as the text overlays cannot be distinguished from the background. Please explain the part the boundary marked with dashed lines in the caption. Please indicate the locations of the snow pits in this figure.

* Table 1: Please include slope information.

* L115 Please expand the sentence with some more details if possible, and if necessary split it into multiple sentences.

* sect 2.2: Is it possible to include plots showing observed precipitation, wind, radiation time series for at least one of the years where there are some data?

* L220 Can the corresponding uncertainty be quantified?

* L230 Can the bias be quantified using geodetic data?

* L231 The stakes don't seem to be following a flowline as far as I can tell from fig 1.

* L235 Is the corresponding contribution to the uncertainty of mass balance quantified?

* L238 Unclear

* L 245 How are the hand-held 'GPS' data is combined with RTK data - their accuracies are quite different? In fact it is not clear if you actually use 'GPS' data anywhere. Also, please be careful about using GPS as opposed to GNSS.

* L253 Where do you discuss this data?

* L255 Why not from all 'DGPS' data?

* 266 "to do other tasks" — please specify what other tasks you are referring to.

* L269 Do you have access to multiple image-pairs?

* L270 Why not use SRTM1 for both the glaciers?

* L285 Why not used GNSS surveyed points to georeference or at least to validate the DEM?

* L289-294 This part may be a bit redundant as it appears from L292-293 that you have not really used any of the data from ICIMOD inventory in the end. Also, is there any reason for not doing the same analysis for RS?

* L295 Are you talking about one image or you have access to multiple images? Please rephrase the sentence accordingly.

* L302 The central meridians are 81E and 87E (https://epsg.io/32644)

* L 336 It is unclear what are you doing here and how.

* L342 I might have missed it, but I did not see the values you assigned for the various uncertainty contributions on the right-hand side of eq 1. A table containing the values used may be included (in the supplementary if you wish).

* L345 Do you use the error estimated for each elevation band (eq 2)? Are they considered in the fits shown in Figs 2 & 3. It would be good to show the error bars in Figs 2 &3 at least for some representative years.

You probably did not define the elevation bands up to this point.

Also, why not simply plot individual stake data available for a period with elevation and do the regression? What necessitates the binning of data from individual stakes into elevation bands? Or, is it that the the summation in eq 2 is over the old and new stakes at the same location, and not really over all the stakes in an elevation band?

* L348 It is unclear what do you mean when you say mass-balance gradients "were
applied to the DEMs".

* L357 rephrase (You mean to say that above 5850 m balance was assumed to be elevation independent?)

* L364 This may be addressed by analysing random subsets of the data where some stakes have been removed.

* eq 3: To compute sigma_final which value of sigma_point_elevb is used? I would expect sigma_point_elevb to vary between years or between elevation bands.

* L490 Not sure about your use of the word 'consistent'. Are you referring to a relatively low interannual variability of the balance gradient? If so, you may rephrase the sentence accordingly. However, please note that the corresponding coefficient of variation is about 2.5 times larger on RS than on Yala - so the two glacier are not really that similar here.

* Sect. 5.1

I do not find the opening paragraph that relevant. I suggest that you either remove this paragraph or save it for later.

I don't understand what is rationale of including Chota Shigri glacier, which is from a different region with distinctly different climate setting, in your discussion/table etc.

In this subsection, two separate topics are being discussed: 1. a comparison of the net balance of Yala with those of some nearby glaciers, and 2. a comparison with past data on Yala. It may be better organise these two topics into two different subsections. Also, if possible, please include some related discussions on RS (or point the reader to relevant references).

* L743 I am not familiar with the usage "the 70ies" and so on. I am not entirely sure about it, but "the '70s" may be a more common usage.

* L 755 I see that the "Conclusions" section contains both conclusions based on the

rest of the paper, and future plans/recommendations that are not really based on things that are discussed in the rest of the paper. For example, the points about the need for homogenized climate data. While this is surely an important point, I wonder if moving this to the 'Discussions' section might have been better.

\* L794 Rephrase the sentence.

---

## Referee Comment (RC2) · Anonymous Referee #2 · 7 Feb 2021

First of all, I would like to congratulate the authors for their efforts in establishing the new glaciological mass-balance programmes at two glaciers in Nepal. Direct measurements are still scarce in the entire Himalayas, and as also written by the authors in their paper, there is an urgent need for such key data for climate monitoring but also for applications such as water resources or hazard management. Moreover, such data are indispensable for model calibration and hence for projections of future developments in this regard. The presentation and discussion of direct glaciological measurements at Yala and Rikha Samba glaciers are the centerpiece of the presented paper (this focus could actually be emphasized even more explicitly in the paper). The data are complemented by other relevant glaciological information such as geodetic mass-balance data, glacier length changes, and glacier velocities. This very nicely fits to the scope

of "Earth System Science Data" and I recommend the paper to be published after addressing the following comments and suggestions.

Overall, the description of data and methods is very detailed and applied methods are sound. Some clarifications and specifications might be needed at some instances, see detailed comments below. Although the presented series of glacier mass balance are still rather short, results are plausible and in line with other studies (using other methods). This is also confirmed by the mass-balance gradients observed in this study, which are consistent over the entire observation years. Regarding the mass-balance gradients, I was though wondering about the different approach to estimate the gradient at high elevations (Yala vs. Rikha Samba glaciers, Figure 3). Although the authors give some indications in the paper and refer to other studies, too, it is not fully clear why there are two different approaches chosen. I suggest that the authors address this point in the Discussion section. Or, if the effect is neglectable for the overall mass balances, it would be useful to have some quantification about the effect.

Regarding the data, I was wondering why the data are presented until 2017 only. The authors refer to the "Global Glacier Change Bulletin No. 3" by WGMS, where we can find the data up to the balance year 2016/17. On the other hand, the authors also refer to the latest version of the "Fluctuations of Glaciers" database by WGMS from August 2020, which includes data until 2018, also Figure 4 includes some data until 2018. This might cause some confusion. Regarding the available datasets, it should also be specified which data are actually available from the WGMS. If I am right, only the glaciological mass-balances are included, but the geodetic data are not (yet?) included and for instance velocity data are currently not stored by the WGMS.

The paper is very nicely structured and clearly presented. As mentioned above, the description is very detailed, which is useful for readers that are looking for comprehensive information about mass-balance programmes (almost in the sense of a review). On the other hand, I am still wondering whether the paper can be shortened and streamlined, such as the Data section and in particular the Discussion as well as the Conclusions

sections. The discussion is very comprehensive and includes many interesting aspects (e.g., climatic influence such as monsoon and jet stream, local weather conditions), which however call for further answers that are not provided in the paper (and which are not in the scope of the paper). I therefore strongly recommend to limit and stream-line the discussion to facts that are directly linked and most relevant to the data of the present paper, and to clearly highlight the importance and the value of the presented mass-balance series. In turn, the Conclusions should concisely focus on the key points from the Discussion.

More specific comments and suggestions are as follows:

Line 9: Should be formulated more precisely, because "essential variable" might be associated with the ECV Glacier according to GCOS/WMO terminology. Moreover, I would separate between the climate aspects (sensitive response of glaciers to climate change) and applications of the data, such as for runoff modelling.

Line 11: This sentence could be deleted as you anyway state at the end of the abstract that the data are available from the WGMS. One might also ask if the main purpose was the ingestion of the data into the WGMS database (which is of course important but probably not the main motivation).

Line 14: It is not fully clear which series refers to the glaciological method and which one to the geodetic method. General comment regarding the dates: although it be-comes mostly clear from the context, it could be helpful to specify the balance years for the glaciological measurements (e.g. 2011/12 to 2016/17). Otherwise the year could refer either to the start of the measurements or to the moment when the observation period is finished.

Line 15: cumulative mass loss

Line 16: missing verb

Line 18: logically not clear, the mass balance of Yala is compared to the mass balance

of another glacier. The same also applies to line 19.

Lines 18-20: Was this really investigated in the present paper? Some aspects are discussed but there are many other aspects that could be considered, and given the fact that there are only very few other glaciological mass-balance series available, I would be careful with such general or relative statements (such as "mostly because of..." or attribution to "low-lying" areas etc.).

Line 23: Specify that the glaciological mass-balance datasets are available.

Lines 28/29: This is a bit too simplified and could be further explained: the glaciological method is indeed a key in international monitoring strategies and is needed for model calibration etc., but for global sea-level rise, the largest ice masses measured by satellite remote sensing are contributing most. Moreover, you could specify what kind of local hazards you refer to.

Line 30: input variable

Line 30: water availability and the change in its availability

Line 34: or: "in support of the United Nations Framework..."

Lines 32-35: 1-2 key references should be given here

Line 37: still involve

Line 44: Some studies have focused (?)

Line 45: Not fully clear, did these studies only address the lower parts of the glaciers (where there is debris cover)?

Line 51: the longest series ... is found (?)

Line 51/52: with measurements since 2002

Line 52: why partly?

Line 55: are glaciological mass-balance records available for both periods?

Line 57: What do you mean with in situ geodetic mass balance measurements?

Line 60: rather point mass-balance measurements than index measurements

Line 62: check tense: have these measurements been continued until today?

Line 63: 1970s, 1980s

Line 65: I suggest writing Pokhalde and Changri Nup Glaciers (in plural), to make clear that you refer to two different glaciers. The same spelling should be used throughout the entire paper, incl. Yala and Rikha Samba Glaciers – I leave it to the authors what they prefer.

Line 67: you could specify what they calibrated against. Geodetic mass-balance data?

Line 68: what do you mean with in situ surface surveys?

Line 69: where exactly was this applied?

Line 78: What was the glacier selected for?

Line 79: or: it offered / enabled easy access

Line 80: what do you mean with glacier processes?

Line 82: delete the comma (last character in line)

Line 87: documented

Line 87: "for Langtang valley": did they also observe other glaciers in the valley?

Line 90/91: "highlights the importance of...": you could elaborate a bit more on this very important aspect.

Line 92: Cogley et al. 2011?

Line 96: Add break and start new paragraph with the main objective.

Lines 93ff: Here I am wondering about the structure: after a rather long introduction, the motivation of the present study is given here again (compare with second paragraph of the Introduction), which leads to some redundancy. Moreover, you refer to the motivation of establishing a sustainable and long-term mass balance programme. This is a very important aspect; however, the paper is about the mass-balance data themselves and not about the monitoring strategy behind. This should be clarified here. In the same paragraph, you mention the training of students and professional; another key aspect in maintaining the monitoring programme on the long-term, however, the paper does not deal with this aspect either. These aspects could be given at the end of the Conclusions in the sense of an outlook.

Line 102: Master theses

Line 104: The focus is not only on Yala glacier.

Line 104: seasonal? Bi-annual could be interpreted ambiguously (twice a year or every two years).

Line 107: For general clarification, I would just refer to the latest version of the database.

Line 109: The questions is also whether the experiences are applicable to other remote areas, too. A short notice in the Discussion or as an outlook at the end could be useful.

Line 112: Altitude range of Yala glacier?

Line 114/115: What kind of data is already available for Yala glacier, and what do you mean with long gaps?

Line 116: mass-balance monitoring programmes?

Line 120, Figure 1: In the overview map, include the name Nepal. As you refer to the different rivers in the text, it could be useful to show them also in the map (I leave this up to the authors how much effort they want to put into the overview map). Regarding

[Figure]

the glacier maps, I am though wondering whether the stake locations also include snow pits? For Yala glacier, I am also wondering why there are no stakes in the upper part of the glacier (i.e. in the northern half of the glacier). Is the area too steep or inaccessible? Are the contour lines from the DEM from 2012 for both glaciers? The hill-shaded DEM seems to apply to Yala, the background for Rikha Samba is an ortho image?

Table 1: Order of characteristics? Glacier locations could be placed first; glacier type information rather below

Line 142: not clear: if the glacier extents further to north-west, why is this area not included in the analyses? Also, not clear: how can the glacier be separated along the flowline? What do you mean exactly?

Line 144: Do the 5% of the area also include the ice cliffs or only the steep slopes? What is the influence of the ice cliffs? (it is addressed in the Discussion, maybe add make a short separate section there, because it is a special feature).

Line 146: based on radar or other measurements?

Line 148: if rockfall covered parts of the glacier, why is it outside the defined outlines? Line 148/149: this transition zone from debris-covered ice to buried ice (dead ice?) and permafrost ground with ice should be shown on the map. And, how is a clear delineation of the glacier margin possible? Does this situation influence the mass-balance observations?

Line 149: Add a break and start new paragraph.

Line 150: Earth's

Line 150: the largest landslide so far documented (?)

Line 152: ka

Line 153: What do you mean with high glaciation? Glaciation during the ice age(s) or glacier coverage until today.

Line 155: why possibly? From your own field experience, you can maybe directly assess the local meteorological conditions?

Lines 154-162: Move the aspects about the climate to the next section.

Line 162: compared to Yala glacier?

Line 169: is sparse

Line 173: "received": check tense

Line 175: the meteo stations you refer to could be added to the map

Line 177: rather mean annual precipitation (also line 181)

Line 178: start new paragraph for Rikha Samba glacier

Line 189/190: not clear: what do you mean here? How is it indicated? From the observed radiation?

Lines 189-200 and entire sub-chapter 2: Regarding the overall structure, I suggest to bring first the more general facts about the setting incl. climate, and then specifically address the two glaciers.

Line 199/200: does this apply in general or especially at Yala glacier?

Line 200: wind speeds

Line 202: was monitored

Line 202: rather write "from autumn 2011 to. . ."

Line 206: avoid abbreviation and "biannually"

Line 207: measurements

Line 218: stakes

Line 219/220: unclear, should be further explained. Also, in line 221/222 you refer to

ablation and accumulation occurring during the same time. For a better general understanding of the seasonal mass-balance evolution at the glaciers, it would be useful to have (1) a general statement about potential accumulation/ablation seasons and (2) how this affects the mass-balance measurements (and (3) how mass-balance measurements are performed in practice).

Line 221: can be clearly identified

Line 225: why is the network stretched along one line? See also my comment regarding Figure 1.

Line 229/230: The effect of the ice cliffs and steep slopes should be placed in the Discussion.

Line 234: Snow depths and snow pits incl. snow density...

Line 238: kg m-3

Line 240ff: This section is very detailed (too detailed). You could just directly refer to Table S1.

Line 245: at best?

Line 255: What are the velocity measurements used for?

Line 258: Explain that "Schneider" is a particular map name (Arbeitsgemeinschaft für vergleichende Hochgebirgsforschung?)

Section 3.2 is interesting to read but very detailed. Maybe some of the information could be placed in a table instead to shorten the section.

Line 305: not clear, rewrite the last sentence

Line 332: Abbreviations should be written out when they appear for the first time.

Line 358: based on the measurements by whom? According to Figure 3, there are no data.

Line 358: balance year 1998/99?

Line 360ff: Are there large changes in the area over the observation period? If not, the difference between conventional and reference-surface balance is small and a corresponding note should be added in the text.

Line 375/376: not fully clear what you mean with regression lines caused by outlying points

Line 380: not clear: what has been separated? Why based on the contour line method? It is not evident from Figure 1 (map).

Line 402: Not clear: earlier in the text you state that the ELA is derived from the mass-balance gradient; is it also observable in the field? "balanced ELA": write ELA0 as in other occasions in the paper and state at some point that ELA0 (AAR0) is the balanced ELA (AAR).

Line 410: What is/was the purpose of this additional profile line? Was it used to calibrate the geodetic data? Was there a correction applied? Or was there any benefit for the in situ glaciological measurements?

Line 434: climate stations?

Line 445: gradient for Yala Glacier from 2011/12 to 2016/17. Delete row with "1999"

Line 450: Not clear whether there was actually a measurement at Rikha Samba in 1999 or if this should be 2011. However, if the latter applies, the caption should read 2010 to 2017, or better 2010/11 to 2016/17.

Line 463: The differences between seasonal and annual balances should be further explained (in the Discussion).

Lines 474-477: Repetition of what has previously been stated.

Line 486: m a.s.l.

Line 488: linear regression?

Line 493: seasonal and annual balances The consistent gradient for the glacier: an interesting finding that should be mentioned in the abstract.

Line 499: Your statements here raise the question of why there are no measurements in the higher areas of the glacier(s). Are these areas too high? Too steep? Too dangerous? This should be specified and clarified at some point.

Lines 500-509: move this paragraph to the Discussion

Line 528: The thinning rate along the profile line is not clear (-1.1 m a-1); is it the mean along the line? Would a mean even make sense? What is the purpose of this profile line?

Line 529: m a.s.l.

Line 538: Table 4 should be moved to the Discussion.

Line 540: In Table 4, you refer to WGMS (2019). The reference is missing and anyway the question is what the original source is. In the table, I would explicitly write "glaciological method".

Lines 545-552: should be moved to the Discussion

Line 550: Maybe also add Chhota Shigri glacier (cf. Discussion)

Line 551: regarding Mera and Pokalde glaciers: where are these glaciers exactly located? They could/should be added to the overview map.

Line 558/559: The comparison of the retreat rates of both glaciers is relative because the glaciers are different in size and form. For Yala glacier, it might also be more straightforward to assess area changes instead of length changes (?). Or to add area changes in addition to the length changes.

Line 579: Figure 12 should be moved to the Discussion.

Line 583: The displacement of the stakes is interesting and allows the link to the velocity measurements.

Line 585: why is the profile line from Sugiyama et al. added? And is this the same profile line as mentioned before?

Line 595: All glaciers and benchmarks as used in the Discussion could be shown in the overview map. For clarification, it should also explicitly be stated

Line 614: The paragraph here as well as the subsequent ones refer always to geodetic mass balances; this should be clearly stated or structed in a clear way so that both observations types are not mixed up.

Line 637ff: What does this mean for assessing glacier mass balances over the entire Himalaya? Is it possible from the available data to draw general conclusions? The authors might elaborate a bit more on this question, which also allows to place the mass-balance data from Yala and Rikha Samba into a larger context and to streamline the Discussion (cf. my general comment above about the Discussion).

Line 640: Such a bias

Line 641: here the term mass budget is used, at other occasions the term mass balance is used. You may check for a consistent use throughout the paper. The same applies to the term net balance (e.g. in line 649 or Figure 14), which should be avoided (cf. the mass-balance glossary by Cogley et al., UNESCO, 2011).

Line 651/652: delete sentence

Line 660/661: 1980s, 1990s (see also line 680 and other occasions)

Paragraph from line 649: It should be better worked out why this context is important for the discussion of the measurements presented in this study.

Line 686: If you specifically address particular stakes with names, then these should also be labelled in the map.

Line 700: There is a break and hence a new paragraph should be started here.

Line 701: why are the measurements on Yala glacier representative? Could you explain that a bit more in detail?

Line 707: The general character of Yala (and Rikha Samba) glacier(s) should be stated at the beginning of the paper (it also relates to my previous comment regarding the ablation/accumulation season, see lines 219/220). In line 708, there is now a contradiction to the what has just been stated before. This should also be clarified.

Line 715: during favorable conditions

Line 723: Is this statement really needed for the discussion?

Line 738ff: rather results, can be deleted here or just refer to Figure 11, but then also an explicit link to the mass-balance data should be made. On the other hand, the Little Ice Age extent is mentioned here for the first time, however, the location is not shown on the length change plot (either add or delete entirely).

Line 756: the mass-balance data shown only cover a few years, so it is probably more appropriate to write about an overall retreating. Or can you exclude that there were more balanced (or maybe even positive) years over the last couple of decades?

Line 760/line 770: Avoid new references or rather move to Discussion

Lines 764ff: Does this refer to Yala and/or Rikha Samba?

Line 772: m a.s.l.

Line 773: This aspect is not mentioned in the Discussion, isn't it? It is an important aspect because it also influences the glacier's mass balance (and there is also another comment above about conventional vs. reference-surface mass balance), therefore I would discuss it before and draw one conclusion that is then presented here at the end.

Line 784: Can you give a time horizon regarding the survival of the glacier? Or are

there any projections for the region?

Lines 786-793: In my view, these sentences can be deleted.

Lines 794ff: In fact, the inclusion of geodetic mass-balance data is an integral part of the monitoring strategy within the Global Terrestrial Network for Glaciers (GTN-G), to compare and eventually calibrate both glaciological and geodetic time series (which is not the case for Yala glacier). This should be clearly stated here. In turn, I would rather conclude how in your case the geodetic method helped to interpret the glaciological mass-balance measurements (and will eventually help to calibrate the series at a later stage).

Line 803/804: are available and are published

Line 805: delete "World Glacier Monitoring Service, Zurich, Switzerland"

Line 810: mass balance data
* * *

---

## Author Comment (AC1) · 18 Apr 2021

**Response to Referee Argha Banerjee**

*We would like to thank Argha Banerjee for the careful review and constructive comments that helped improve the manuscript. We responded to all comments raised by the referee point by point in blue italic font.*

**General changes**

*In February 2021, we resubmitted data to the World Glacier Monitoring Service (WGMS) for the Fluctuations of Glaciers (FoG) database. The data included mass balance data for the annual and seasonal mass balance 2016/17 of Yala Glacier which we corrected due to a shift in the elevation records. Additionally, we submitted seasonal mass balance data for Yala Glacier. The updated FoG database is scheduled to be published in the coming weeks and until then the respective data can be requested from the WGMS. All related values have been corrected in the manuscript.*

*Besides responding to the referees' comments, the sections "3 Data and methods" and "4 Results have been reorganised". Section "5 Discussion" has been restructured and streamlined, the section "6 Conclusions" has been revised and a section "7 Recommendations" has been added.*

*In the Supplement, we added a section "S1 A brief description of summer-accumulation type glaciers and related mass balance measurements". Inspired by a comment by referee 2. We briefly describe summer-accumulation type glaciers, the related accumulation and ablation seasons, how it affects the mass-balance measurements and how the measurements are performed.*

*In section S3 we present Figures showing the mass balances and uncertainties for elevation bands at Yala and Rikha Samba glaciers, based on suggestions of referee Argha Banerjee.*

*In section S4 we added Figures and Tables to illustrate issues related to the representation of steep slopes in DEMs. Steep slopes and ice cliffs occur on Yala Glacier but we could neither quantify the mass balance, nor the relevance of steep slopes in terms of area in the DEM.*

**Comments Referee Argha Banerjee**

Stumm and coauthors present and analyse glaciological mass balance and other related data measured at Yala and Rikha Samba (RS) glaciers. These data sets spanning seven balance years may be invaluable contributions towards a better understanding of the high-Himalayan glaciers and climate. However, there are a few points listed below where I believe there are some scope for improvements.

**Major comments**

1. I am concerned about the two very different strategies employed to extrapolate the winter/annual balance over the upper reaches of the accumulation zones of RS and Yala glaciers: An elevation-independent constant accumulation rate on RS, and a constant elevation-gradient based interpolation on Yala. While some discussions are presented with reference to Yala, a more detailed discussion of the rationale behind adopting these the two methods, and a quantitative analysis of the corresponding uncertainties may be necessary.

As far as I can tell looking at the fig 2 (left most panel), the balance measured at the upper few stakes/pits may be better described with a constant elevation-independent balance rate. (Let me mention here that I am not sure if the present choice of horizontal range for the left-most panel of fig 2, which seriously compromises the readability, is a good one. I request you to please restrict the horizontal range to something reasonable, like [-1,1] m w.e.)

*The two different strategies to extrapolate the annual mass balance are now addressed in more detail in section "3 Data and methods", and in section "5 Discussion" (revised text below).*

*The uncertainties have been calculated for all elevation bands of 50 m intervals (eq. 3) for the annual mass balance and can be retrieved from the WGMS Fluctuations of Glaciers (FoG) database. For visualisation, the mass balance for each elevation band has now been plotted together with the uncertainties in the Supplement (Fig S3, S4, S5, S6). As described in the manuscript, the uncertainties are larger in the accumulation area.*

*Figure 2 has been revised. The design of Figure 2 follows the format used by the WGMS in the Global Glacier Change Bulletin (GGCB), in section "3 Detailed information". The GGCB is freely accessible from www.wgms.ch/ggcb. In the GGCB data related to the FoG database is published.*

*In section 3 the subsection "3.1.1 Point and glacier-wide mass balance" the revised text now reads:*
*"For Yala Glacier, characteristic gradients for the ablation area were identified, and separately analysed for the annual and seasonal mass balances, with the winter and summer season starting in November and May or June, respectively. In the accumulation area, there are fewer measurements with large uncertainties because of the challenging measurement conditions described earlier and in Supplement section S1. This inhibited not only to identify characteristic gradients in the accumulation area, but also to define a fixed mass balance that could be applied in the accumulation area from a defined elevation. As consequence, a single gradient was used for the glacier-wide mass balance. The interpolation approach is simple and introduces a systematic error for the mass balance in the accumulation area. The part of the accumulation area without measurements for the respective elevations bands makes up 15 % of the glacier area for an elevation range of about 160 m (~5500 m to 5662 m a.s.l.).*
*For Rikha Samba Glacier two characteristic annual gradients were identified, with a large gradient in the lower ablation area and a medium gradient in the transition between ablation and accumulation area. Based on the assumption that the mass-balance gradients remain very similar in different mass-balance years, gradients were reconstructed for Rikha Samba Glacier for years with limited point measurements (2011/12, 2013/14, and 2014/15). The intersection points of the lower (large) and upper (medium) gradients were identified and reconstructed based on a regression line for sections*

*without measurements. For the accumulation area, no characteristic gradients could be identified because only few measurements were available. The elevation range without measurements is about 650 m (~5900 m to 6545 m a.s.l.) and makes up 36 % of the glacier area. At about 6000 m, the topography steepens (Fig 1). Using the upper gradient to interpolate the mass balance to the accumulation area would have resulted in much overestimated positive mass balances. Instead we considered it plausible to assume a fixed mass balance at high elevations, based on the steep slopes and the typically small gradient in accumulation areas. We assumed the lower elevation for a fixed mass-balance value between 5850 m and 5950 m a.s.l., guided by the upper gradient."*

*In section 5 the subsection "5.6 Extrapolation of in situ measurements to the accumulation area" has been added and now reads:*
*"In the ablation area of Yala and Rikha Samba glaciers sufficient in situ measurements largely allowed the interpolation of the data by using elevation dependent mass-balance gradient. In the accumulation area, measurements were often challenging and associated with higher uncertainties. The main issues were difficult access, and cumulative ablation that temporarily exceeded the cumulative accumulation (Supplement section S1). On one hand this ablation removed the marked reference surfaces for the accumulation measurements, and on the other hand the uncertainty is increase for ablation measured with stakes installed in an unstable firn and snow underground. Additionally, no accumulation data could be collected at the highest elevations.*
*To extrapolate the mass balance to higher elevations, we made a few considerations: the glacier mass-balance programmes were running only within the first decade, and a re-evaluation and possible correction of the glacier-wide mass balance with help of other methods is likely in the future (Zemp et al., 2013; Cullen et al., 2016; Wagnon et al., 2020). Therefore, we chose simple extrapolation approaches.*
*At Yala Glacier, extrapolating the ablation gradient to the accumulation area introduced a systematic error for a small glacier area (15 % of the total area) with a small elevation range (~160 m). The largest errors are expected in the highest elevation bands, where accumulation is overestimated (Fig. S3 and S4). At the steep south-west-oriented slopes of Yala Glacier, the ablation is likely increased and underestimated in the glacier-wide mass balance. At Rikha Samba Glacier, using the same extrapolation method like at Yala Glacier would have very much overestimated the accumulation in a large area (36 % of the total area) with a large elevation range (~650 m). Instead, we estimated a fixed value for the accumulation area, which introduced a random error. Consequently, complementing independent methods to assess the glacier mass balance for the same time interval help reducing uncertainties and should be part of any comprehensive glacier mass-balance programme following the international glacier monitoring strategy (WGMS, 2020b; Haeberli et al., 2000)."*

2. I think the paper may benefit from a more careful presentation of the geodetic analysis in general.

While I am not that familiar with intricacies of geodetic techniques involved, I wonder if obtaining horizontal (vertical) location of the GCPs from Landsat (SRTM) products having a relatively lower resolution (and vertical accuracy) may seriously compromise the accuracy of the generated GeoEye-1 DEM that has nominal horizontal resolution of 2-5 m. In general, the rationale behind switching between SRTM1 and SRTM3 for various pieces of analysis done needs to be made clear. Also, why not use SRTM1 here?

Could you not check the accuracy of the DEM making use of any GNSS data from the stable terrain that may be available? A comparison between your May 2012 GNSS elevation profile, and the same extracted from Jan 2012 DEM could be very useful here. I missed such a plot in the present manuscript. In fact, it appears that you do not make much use the highly accurate elevation profile from GNSS survey, apart from comparing it with 2000 SRTM3 data.

It may be useful to plot the distribution of differences between DEM2000 and DEM2012 on the off-glacier stable terrain.

*We agree with the suggestions to make best use of dGNSS data and use up to date data, as far as constraints allow. In winter 2011/12 we ordered GeoEye-1 stereo images covering the area of Yala Glacier. We intended to have stereo images at an early stage of the glacier monitoring programme to generate a DEM and to analyse the geodetic mass balance. In February 2014 under the guidance of Tobias Bolch, Tino Pieczonka and Nicolai Holzer we were able to calculate the geodetic mass balance of Yala Glacier between 2000 and 2012. However, in early 2014 we did not have sufficient dGNSS data to georeference the GeoEye-1 images, and used the GCPs based on Landsat images and STRM-3 as instructed.*

*When we ordered the GeoEye-1 images, we chose a restricted rectangle around Yala Glacier because of the costs. When generating the DEM, we learnt that a larger area would have allowed for better comparison of stable terrain data and selection of GCPs. However, at that time we couldn't acquire larger GeoEye-1 images anymore.*

*For Yala Glacier we analysed the geodetic mass balance in 2014 when only SRTM-3 was available. Later we did not have the capacity to recalculate the DEM and geodetic mass balance with SRTM-1. For Rikha Samba Glacier we recalculated the glaciological mass balance in 2017 and used the newer SRTM-1.*

*For the future, there is a need to reanalyse the glaciological mass balance as recommended by Zemp et al. (2013) and done e.g. by Wagnon et al. (2020) for Mera Glacier. New geodetic mass balance analyses will be necessary to cover the same time interval as the glaciological measurements. In this process the repeated surface profile lines from 2009 (Sugiyama et al. 2013), 2012, 2014 and 2017 will be supporting data generated with an independent method. Here we present the data currently available.*

3. You have invoked subtropical jet stream which operates on a planetary scale to explain strong winds on certain glaciers during the winter season while referring to Ding & Sikka (2000) and Wagnon et al. (2013). You have not discussed this issue in depth, but I find it hard to imagine how a planetary-scale system influences two glaciers from the same region in a differnt manner.

It seems that the connection between jet stream and winds speed on glacier scale that Wagnon et al. (2013) make is entirely based on observation at a single glacier/location. Also, the authors didn't discuss this issue in that detail. Unfortunately I could not access the article that they referred to or Ding & Sikka (2000) that you refer to, so not sure if these references talk about some relevant mechanism in the context of Himalayan glaciers.

To me your data from Yala may instead be a pointer that local effects can have a strong impact on spatiotemporal variability of winds. I believe a one-to-one correspondence between local scale and planetary scale winds will always be tricky - more so with the complex Himalayan topography.

I suggest that you please revise your discussions on the effects of the jet stream on local winds. Please provide sufficient arguments about the corresponding mechanism and/or cite relevant references.

*The paragraph in the section "2.2 Climate" was revised and is based on the study by Shea et al. (2015b) and Fujita et al. (2001). Shea et al. (2015b) measured, described and compared the winds at Rikha Samba, Yala and Mera Glacier. The entire section "5 Discussion" has been revised and restructured and the sentence formerly referring to the jet stream has been edited. The study of the winds was not the focus; however, the winds play a role for the glacier mass balance and have been discussed accordingly. The reference Ding & Sikka (2006) has been delete.*

*The revised paragraph in section "2.2 Climate" reads now: "The wind directions at the Yala Base Camp station show a dominance of bimodal valley winds (Shea et al., 2015b). The Rikha Samba station is additionally exposed to synoptic-scale flows. Throughout the year, the wind velocities at Rikha Samba*

*Glacier are higher and with a larger variability than at Yala Glacier. The highest wind speeds are recorded in winter from October to May, with strong wind events with >8 ms$^{-1}$ (Fujita et al., 2001). Winter wind velocities measured at Rikha Samba Glacier are very high and result from the channelling of synoptic-scale winds (Shea et al., 2015b). The winter wind speeds at Yala Glacier are much smaller, probably because Yala Glacier is better sheltered by surrounding high mountains. During monsoon from June to September the wind speeds at both glaciers are lower with a smaller variability."*

*The revised sentence in the discussion subsection "5.3 Comparison of in situ glacier mass balances in the Himalaya" now reads: "Wagnon et al. (2013) address the high wind speeds from westerly winds at Mera Glacier (5360 m a.s.l on glacier station) in winter, which causes in combination with sublimation a substantial part of the winter ablation."*

4. If I am not mistaken, the paper refers to more detailed data from the two glaciers than that can be accessed from FoG database (for example, stake displacements, winter balance, digital elevation models etc.). It would be a great if all such data used in the study can be made accessible to the community.

*In the abstract the available data has been further specified: "The data of the annual and seasonal mass balances, point mass balance, geodetic mass balance and length changes are accessible from WGMS (2021): Fluctuations of Glaciers Database, World Glacier Monitoring Service, Zurich, Switzerland, http://dx.doi.org/10.5904/wgms-fog-2021-xx"*

*The Fluctuations of Glaciers (FoG) Database of the World Glacier Monitoring Service is open access and the data can be freely downloaded also from: https://wgms.ch/data_databaseversions/.*

*The guidelines for the reported attributes are documented in "General Guidelines for Data Submission and Notes on the Completion of Data Sheets" and can be freely downloaded from: http://wgms.ch/downloads/WGMS_AttributeDescription.pdf*

*The FoG database uses a standardized format for specified data. The mass balance data for this article has been made available in the respective datasheets. Some of the data has been submitted as early as in 2012. Additional data as well as corrected data for the annual and summer balance in 2017 has been submitted to the WGMS in February 2021. This data is scheduled to be published in spring 2021. The revised manuscript refers to this updated database for data up to 2017. More recent mass balance data is part of other studies. The updated data can be requested by reviewers from the WGMS until the data is ingested into the database and freely available from the website.*

*In the FoG database, general information can be found in the sheets A and B. Length changes and geodetic mass balances are provided in the sheets C and D, respectively. Data related to the glaciologically measured mass balances can be found in sheet E, the glacier wide mass balance and mass balance for elevation bands are documented in sheet EE, and point mass balance data in sheet EEE. Other data such as digital elevation models or stake displacement data are outside the scope of the FoG database and may be requested from co-authors working at ICIMOD.*

**Minor comments:**

* You have discussed the difficulties in identifying the lower boundary of the annual accumulation of snow in L513 and also in section 5.1. This an important issue that has implications for the data collected in the past on Himalayan glaciers, and may help improve similar measurements in the future. Therefore this point may be highlighted with the corresponding discussions moved to a dedicated subsection within the Discussions section.

*In section "3 Data and methods", subsection "3.1.1 In situ mass balance" the text has been revised, and the described issue is further elaborated in the Supplement section 1. The described issue mainly occurs at sites with strong winds or close to the ELA but not necessarily everywhere. It is also difficult to improve the measurements. Instead, complementing geodetic measurements for the same time interval are important and a practical approach to correct this bias.*

*In section "5 Discussion", the importance of complementing geodetic mass balance analyses for the same timeframe are emphasised, as well as the need for a later re-evaluation of the glacier-wide mass balance to address uncertainties and biases such as related to the accumulation measurements.*

*The revised text in subsection "3.1.1 In situ mass balance" now reads: "The in situ mass balance was measured following Kaser et al. (2003), taking into consideration aspects in the ablation and accumulation area specific to summer-accumulation type glaciers (for details see Supplement, section S1). […] In the accumulation area, snow pits are dug or cores taken, and the snow profile, depth and density recorded. Additionally, several snow probing measurements are taken. Bamboo stakes mainly mark the measurements sites, but in absence of snow pit data they are also used for the mass-balance calculation, in particular in the case of a negative mass balance. The snow pit measurements are only reliable if the previous measurement surface can be clearly identified, e.g. when marked with a sawdust layer. Difficulties arise in the accumulation area, if the cumulative ablation temporarily exceeds the cumulative accumulation during the measurement period (Fig. S2). The exceeding ablation is not represented in a snow pit measurement and likely impacts the sawdust layer. Stake readings are less reliable because the underlying snow and firn layers compact over time and may push or pull the stake up or down."*

\* I request a carefully revision the manuscript to edit the language. Please identify and replace/rephrase sentences that are either difficult to understand and/or do not conform to standard English usage. I have tried to point out a few such instances, but there may be a few more.

*The language of the manuscript has been revised and suggestions accepted as indicated in the response to the comments. Sentences and paragraphs have been revised as indicated in the manuscript with track changes. As guideline British English was used and the Oxford Dictionary of English.*

\* Please improve the readability of figure 1. Often the contour lines are all but invisible (particularly for RS). Elevation labels are generally hard to read as the text overlays cannot be distinguished from the background. Please explain the part the boundary marked with dashed lines in the caption. Please indicate the locations of the snow pits in this figure.

*The maps and caption of Fig. 1 were revised, and following changes were made:*

*Both glacier maps:*

- *Contour lines and elevation labels visibility improved*
- *Legends: Labelled "measurement sites" instead "stakes" because also snow measurements were taken if snow was present.*
- *Maps: Measurements sites labelled.*

*Yala Glacier:*

- *The adjacent glaciers from the ICIMOD glacier inventory (modified) were added to give the context of Yala Glacier and the dashed line is removed.*

*Overview map:*

- *The Himalayan range, rivers and glacier area are shown, including the investigated glaciers and other glaciers mentioned in the discussion as requested by referee 2. Country boundaries are removed to adhere to the journal's guidelines.*

*Caption revised: "Figure 1: The study sites Rikha Samba and Yala glaciers showing the measurement sites and their location in the Himalayas. At all measurements sites stakes were installed. Snow pits were dug at the top stakes and at selected lower stakes provided snow was present. (a) For Rikha Samba Glacier RapidEye orthoimages from April 2010 were used for the background image and glacier outlines. The contour lines are derived from the SRTM-3 DEM. (b) For Yala Glacier GeoEye-1 orthoimages from January 2012 were used for the background image and in combination with dGNSS data for the glacier outlines. The contour lines are derived from the DEM2012 generated from the*

*GeoEye-1 stereo images. (c) The overview map shows the location of the two investigated glaciers and other glaciers mentioned in the discussion section. The glacier inventory is from ICIMOD (Bajracharya et al., 2014)."*

\* Table 1: Please include slope information. → *Slope information added in Table 1*

\* L115 Please expand the sentence with some more details if possible, and if necessary split it into multiple sentences. → *Sentence deleted; literature review already provided in the introduction.*

\* sect 2.2: Is it possible to include plots showing observed precipitation, wind, radiation time series for at least one of the years where there are some data?

*Shea et al. (2015) published such data. Sentences were added referring to this source in the section "2.2 Climate": "Meteorological data from Rikha Samba Glacier, Yala Glacier and other automatic weather stations (AWS) in the Langtang and Dudh Koshi catchments were compared by Shea et al. (2015b). They analysed temperature, precipitation, incoming radiation, wind and other parameters from December 2012 to December 2013, as far as data were available."*

\* L220 Can the corresponding uncertainty be quantified?

*We assessed the uncertainty of the point measurements individually, estimating the errors according to eq. 1. The resulting uncertainties of the point measurements are larger in the accumulation area than in the ablation area. Consequently, the uncertainties of the mass balances for the elevation bands (eq. 3) is also larger in the accumulation area.*

*The uncertainties of the point measurements and the elevation bands is documented in the WGMS FoG database in sheets EE and EEE (WGMS-FoG-2020-08-EE-MASS-BALANCE.csv and WGMS-FoG-2020-08-EEE-MASS-BALANCE-POINT.csv; Please see also the response to major comment #4.*

*The uncertainty of some type of errors can be relatively easily quantified (e.g. density). Other types of error are difficult to quantify and additional measurements with other methods would be required, which is sometimes not practical. The standard approach to address such uncertainties in glaciological measurements is to use geodetic mass-balance analyses to correct the overall annual mass balance series as done by Wagnon et al., 2020 (see also Zemp et al. 2013).*

\* L230 Can the bias be quantified using geodetic data?

*With high resolution geodetic mass balance data for time intervals of only a few years e.g. based on lidar data (Joerg and Zemp, 2014), the ablation at the ice cliffs can probably be quantified. However, we don't have this data to estimate the bias.*

\* L231 The stakes don't seem to be following a flowline as far as I can tell from fig 1.

*Corrected: "On Rikha Samba Glacier, eight stakes are installed along the approximate glacier centre line with some deviation, which follow roughly the stake setup of the Japanese researchers (Fujita et al., 2001).:*

\* L235 Is the corresponding contribution to the uncertainty of mass balance quantified?

*Please see the uncertainties in the WGMS FoG database sheets EE and EEE. We quantified the uncertainties based on our data, knowledge and experience (eq. 1), and discussed the uncertainty assessment with M. Huss. Please see also the response to major comment #4 and comment L220.*

\* L238 Unclear

*The paragraph has been deleted here. The revised paragraph is mentioned in section "4 Results": in subsection "4.1 Mass balances, ELA, AAR and gradients.*

*The revised paragraph now reads: "At Yala Glacier, the measured average densities with standard deviation for snow and firn were 336 kg m$^{-3}$ (±81 kg m$^{-3}$) and 562 kg m$^{-3}$ (±128 kg m$^{-3}$). However, harder firn layers were difficult to measure. Dependent on the site and firn conditions, and based on snow pit*

*profiles and field observations we estimated firn density between 550 kg m$^{-3}$ and 700 kg m$^{-3}$. At Rikha Samba Glacier, the average snow density measured was 399 kg m$^{-3}$ with a standard deviation of ±70 kg m$^{-3}$. For ice we assumed a density of 900 kg m$^{-3}$ (Cogley et al., 2011)."*

* L 245 How are the hand-held 'GPS' data is combined with RTK data - their accuracies are quite different? In fact it is not clear if you actually use 'GPS' data anywhere. Also, please be careful about using GPS as opposed to GNSS.

*We revised the entire manuscript and wrote "GNSS" instead of "GPS". In case of differential GNSS we used the abbreviation dGNSS. Now, we use the term "GPS" only for the handheld Garmin GPS unit.*

*We revised the sentence regarding the RTK mode: "The devices were dual frequency dGNSS units from Topcon and Magellan ProMark 3, and used in real time kinematic (RTK) mode."*

* L253 Where do you discuss this data?

*The results from the profile line survey at Yala Glacier is presented in the results section, and discussed in the section "5 Discussion". The revised text in subsection "5.1.1 Annual mass-balances rates" now reads: "Yala Glacier's annual geodetic mass-balance rate is -0.74 ±0.53 m w.e. a$^{-1}$ from 2000–2012 (Table 6). The thinning rate along the profile line is with -1.1 ±0.13 m a$^{-1}$ higher but within the uncertainty range of the DEM thinning rate, most likely because accumulation above 5571 m a.s.l. is excluded from the calculation."*

* L255 Why not from all 'DGPS' data?

*In the two mass-balance programmes, the new stakes are installed at the original coordinates rather than at the melt-out location of old stakes. To measure velocities, the same stakes must be surveyed. However, no stake lasted from 2014 to 2017.*

* 266 "to do other tasks" — please specify what other tasks you are referring to.

*"to do other tasks" deleted*

* L269 Do you have access to multiple image-pairs?

*Sentence deleted.*

*Background: We ordered a commercial GeoEye image for winter 2011/12. During this period, we received offers with preview images to select suitable stereo images within a restricted timeframe. We also evaluated Hexagon and Aster stereo-images, but they were unsuitable for the small area of Yala Glacier.*

* L270 Why not use SRTM1 for both the glaciers?

*The geodetic mass balance of Yala Glacier we analysed in February 2014 when only the SRTM-3 DEM was available. The data we submitted to the WGMS in early 2015. Due to limited resources at a later stage we didn't recalculated the geodetic mass balance with SRTM-1.*

*For the glacier mass balance analysis, it is better to have a more up to date DEM if available. In case of Yala Glacier, the DEM generated from the GeoEye-1 images was from 2012, and has a higher accuracy than the SRTM-1 DEM.*

* L285 Why not used GNSS surveyed points to georeference or at least to validate the DEM?

*We had one benchmark site that was established earlier by Japanese researchers, which wasn't enough for georeferencing or validation of the DEM. Up to February 2014 (generation of DEM with help of T. Bolch and colleagues), we only had limited access to dGNSS devices, and only limited time to conduct such surveys besides the mass balance measurements and trainings of about 25 participants. Initially the dGNSS data from May 2012 had a post-processing error, which could be corrected only later, and did not include GCPs.*

* L289-294 This part may be a bit redundant as it appears from L292-293 that you have not really used any of the data from ICIMOD inventory in the end. Also, is there any reason for not doing the same analysis for RS?

*Sentence adjusted and later sentence about the inventory outline data deleted. The geodetic mass balance of Rikha Samba Glacier has also been calculated and is part of a different study. The revised sentence now reads: "We used a Landsat 7 Enhanced Thematic Mapper (ETM+) image from 2000 to identify the outlines of Yala Glacier for the geodetic mass balance and frontal variation analyses."*

* L295 Are you talking about one image or you have access to multiple images? Please rephrase the sentence accordingly.

*Sentence rephrased: "A Hexagon KH-9 image from November 1974 was used for a frontal variation analysis of Yala Glacier. Other Hexagon images were found…"*

* L302 The central meridians are 81E and 87E (https://epsg.io/32644)

*Sentence clarified: "We used the local projection system called Modified Transverse Mercator,…"*

* L 336 It is unclear what are you doing here and how.

*Deleted, context now clarified in section "3.1.1 In situ mass balance"*

* L342 I might have missed it, but I did not see the values you assigned for the various uncertainty contributions on the right-hand side of eq 1. A table containing the values used may be included (in the supplementary if you wish).

*The error of each individual point measurement was assessed by analysing each error in eq. 1, provided it was relevant. The final error for the point measurement is reported in the WGMS FoG database sheet EEE (WGMS-FoG-2020-08-EE-MASS-BALANCE.csv).*

* L345 Do you use the error estimated for each elevation band (eq 2)? Are they considered in the fits shown in Figs 2 & 3. It would be good to show the error bars in Figs 2 &3 at least for some representative years.

You probably did not define the elevation bands up to this point.

Also, why not simply plot individual stake data available for a period with elevation and do the regression? What necessitates the binning of data from individual stakes into elevation bands? Or, is it that the the summation in eq 2 is over the old and new stakes at the same location, and not really over all the stakes in an elevation band?

*For clarity, the text has been slightly restructured and eq. 2 and 3 have been swapped. Equation 3 (now: error of the point measurements for a specific elevation band $\sigma_{point\_elevb}$) is needed is to calculate equation 2 (now: overall error $\sigma_{final}$ for the mass balance for elevation bands and glacier-wide mass balance).*

*In the WGMS FoG database sheet EE (WGMS-FoG-2020-08-EE-MASS-BALANCE.csv), the mass balance and uncertainty for the annual glacier-wide mass balance is reported, as well as for the elevation bands with the upper and lower limits. For both glaciers the elevation bands are 50 m for the entire glacier except for the elevation bands at the lowest and highest glacier elevations. The uncertainties have been assessed for each point measurements, elevation band, as well as for every year. There can be considerable variations dependent on the conditions of the glacier at the time of the measurements.*

*The uncertainties for all annual balance are documented in Table 2 and 3, and for the elevation bands and point measurements in the FoG database. Figure 2 and 3 don't show the uncertainties of the point measurements or elevation bands to maintain the readability of the Figures. The uncertainties of the mass balance of the elevation band for all six years are now plotted in the Supplement in Fig. S3, S4, S5 and S6.*

*The revised text in subsection "3.4 Analysis of glacier changes and uncertainties" now reads:*

*"To assess the error of the mass balance for the entire glacier and elevation bands of 50 m, the errors of the point measurements $\sigma_{point\_elevb}$ and interpolation method $\sigma_{int}$ were analysed. […]*

*The overall error $\sigma_{final}$ for the mass balance for the glacier-wide balance and elevation bands was calculated:*

$$\sigma_{final} = \sqrt{\sigma_{point\_elevb}^2 + \sigma_{int}^2} \qquad\qquad (2).$$

*The error of the point measurements for a specific elevation band $\sigma_{point\_elevb}$ was calculated by considering n point measurements in the respective elevation band:*

$$\sigma_{point\_elevb} = \sqrt{\sum_{point=1}^{n} \sigma_{point}^2} / \sqrt{n} \qquad\qquad (3).$$

*To calculate the systematic error caused by the interpolation method $\sigma_{int}$, we estimated […]"*

\* L348 It is unclear what do you mean when you say mass-balance gradients "were applied to the DEMs".

*Sentence rephrased: "The mass-balance gradients were derived from the linear regression lines of the point measurements. The elevations of the DEMs of Yala and Rikha Samba glaciers were applied to the regression equations to calculate the glacier-wide mass balance."*

\* L357 rephrase (You mean to say that above 5850 m balance was assumed to be elevation independent?)

*The rephrased sentences now read: "[…] Instead we considered it plausible to assume a fixed mass balance at high elevations, based on the steep slopes and the typically small gradient in accumulation areas. We assumed the lower elevation for a fixed mass-balance value between 5850 m and 5950 m a.s.l., guided by the upper gradient."*

*For context please also note the response to major comment #1.*

\* L364 This may be addressed by analysing random subsets of the data where some stakes have been removed.

*The subset of stake data in a year is not big enough to do that.*

\* eq 3: To compute sigma_final which value of sigma_point_elevb is used? I would expect sigma_point_elevb to vary between years or between elevation bands.

*Yes, the uncertainties vary between years and elevation bands and are documented in FoG sheet EE. The value of $\sigma_{point\_elevb}$ is not a fixed value and depends on $\sigma_{point}$, which is documented in FoG sheet EEE. Please also see response for Line 345.*

\* L490 Not sure about your use of the word 'consistent'. Are you referring to a relatively low interannual variability of the balance gradient? If so, you may rephrase the sentence accordingly. However, please note that the corresponding coefficient of variation is about 2.5 times larger on RS than on Yala - so the two glaciers are not really that similar here.

*The sentences have been revised: "For Yala and Rikha Samba glaciers, the mean mass-balance gradient at the ELA are 1.04 m and 0.36 m w.e. (100 m)$^{-1}$, respectively (Table 2 and 3). The gradients show a relatively low interannual variability with standard deviations of 0.12 m and 0.9 m w.e. (100 m)$^{-1}$, respectively."*

\* Sect. 5.1

I do not find the opening paragraph that relevant. I suggest that you either remove this paragraph or save it for later.

I don't understand what is rationale of including Chota Shigri glacier, which is from a different region with distinctly different climate setting, in your discussion/table etc.

In this subsection, two separate topics are being discussed: 1. a comparison of the net balance of Yala with those of some nearby glaciers, and 2. a comparison with past data on Yala. It may be better organise these two topics into two different subsections. Also, if possible, please include some related discussions on RS (or point the reader to relevant references).

*The section "5 Discussion" has been reorganised and streamlined. The opening paragraph has been shortened and is used later. The comparison of the mass balances of Yala and other glaciers is now in the subsection "5.3 Comparison of in situ glacier mass balances in the Himalaya" and subsection "5.4 Bias by small low-lying glaciers". The comparison of past data from Yala Glacier is now discussed in subsection "5.5 Interannual variability of winter precipitation and long-term trends of accumulation".*

*We agree, Chhota Shigi Glacier is from a different region and climatic setting in the Western Himalayas. We added Chhota Shigri Glacier because it is one of very few glaciers with a longer series of in situ mass-balance data from the Himalayas and also under the influence of the Indian summer monsoon although at higher latitude and more influenced by westerly disturbances. Inspired by section "2 Regional Information" in the WGMS Global Glacier Change Bulletin we considered it reasonable to include Chhota Shigri as a Himalayan glacier in the comparison. Based on the suggestion of referee 2 we added Chhota Shigi also in Fig 13. with the cumulative mass balance.*

*There is a large range of studies for Langtang but only few studies for Rikha Samba Glacier. Hence, the related discussion for Rikha Samba is much shorter.*

* L743 I am not familiar with the usage "the 70ies" and so on. I am not entirely sure about it, but "the '70s" may be a more common usage.

*The entire manuscript has been revised, to consistently use 1960s, 1970s, 1980s, 1990s.*

* L 755 I see that the "Conclusions" section contains both conclusions based on the rest of the paper, and future plans/recommendations that are not really based on things that are discussed in the rest of the paper. For example, the points about the need for homogenized climate data. While this is surely an important point, I wonder if moving this to the 'Discussions' section might have been better.

*The section "6 Conclusions" has been revised and the structure improved. In the section "5 Discussion", subsection "5.3 Yala and Rikha Samba glaciers in a Nepalese and Himalayan context", climatic aspects influencing the mass balance have been addressed. A sentence has been added: "To better understand the relationship between the climate and the mass balance of Yala an Rikha Samba glaciers, the analysis of homogenised climate data from nearby weather stations or reanalysis data would be useful."*

*In the new section "7 Recommendations", sentences have been added: "The comparison of mass balance data with climate data is needed to better understand the climate signal of the mass-balance data. Homogenised data from AWSs or reanalysis climate data could be used for that purpose."*

* L794 Rephrase the sentence.

*The rephrased sentence now reads: "Geodetic mass-balance analyses overlapping the time interval of the glaciological measurements of Yala and Rikha Samba glaciers are needed (Zemp et al., 2013)."*

---

## Author Comment (AC2) · 18 Apr 2021

**Response to anonymous referee #2**

*We would like to thank referee 2 for the careful and detailed review and constructive comments that helped improve the manuscript. We responded to all comments raised by the referee point by point in blue italic font.*

**General changes**

*In February 2021, we resubmitted data to the World Glacier Monitoring Service (WGMS) for the Fluctuations of Glaciers (FoG) database. The data included mass balance data for the annual and seasonal mass balance 2016/17 of Yala Glacier which we corrected due to a shift in the elevation records. Additionally, we submitted seasonal mass balance data for Yala Glacier. The updated FoG database is scheduled to be published in the coming weeks and until then the respective data can be requested from the WGMS. All related values have been corrected in the manuscript.*

*Besides responding to the referees' comments, the sections "3 Data and methods" and "4 Results have been reorganised". Section "5 Discussion" has been restructured and streamlined, the section "6 Conclusions" has been revised and a section "7 Recommendations" has been added.*

*In the Supplement, we added a section "S1 A brief description of summer-accumulation type glaciers and related mass balance measurements". Inspired by a comment by referee 2. We briefly describe summer-accumulation type glaciers, the related accumulation and ablation seasons, how it affects the mass-balance measurements and how the measurements are performed.*

*In section S3 we present Figures showing the mass balances and uncertainties for elevation bands at Yala and Rikha Samba glaciers, based on suggestions of referee Argha Banerjee.*

*In section S4 we added Figures and Tables to illustrate issues related to the representation of steep slopes in DEMs. Steep slopes and ice cliffs occur on Yala Glacier but we could neither quantify the mass balance, nor the relevance of steep slopes in terms of area in the DEM.*

**Comments anonymous referee #2**

First of all, I would like to congratulate the authors for their efforts in establishing the new glaciological mass-balance programmes at two glaciers in Nepal. Direct measurements are still scarce in the entire Himalayas, and as also written by the authors in their paper, there is an urgent need for such key data for climate monitoring but also for applications such as water resources or hazard management. Moreover, such data are indispensable for model calibration and hence for projections of future developments in this regard. The presentation and discussion of direct glaciological measurements at Yala and Rikha Samba glaciers are the centerpiece of the presented paper (this focus could actually be emphasized even more explicitly in the paper). The data are complemented by other relevant glaciological information such as geodetic mass-balance data, glacier length changes, and glacier velocities. This very nicely fits to the scope of "Earth System Science Data" and I recommend the paper to be published after addressing the following comments and suggestions.

Overall, the description of data and methods is very detailed and applied methods are sound. Some clarifications and specifications might be needed at some instances, see detailed comments below. Although the presented series of glacier mass balance are still rather short, results are plausible and in line with other studies (using other methods). This is also confirmed by the mass-balance gradients observed in this study, which are consistent over the entire observation years. Regarding the mass-balance gradients, I was though wondering about the different approach to estimate the gradient at high elevations (Yala vs. Rikha Samba glaciers, Figure 3). Although the authors give some indications in the paper and refer to other studies, too, it is not fully clear why there are two different approaches chosen. I suggest that the authors address this point in the Discussion section. Or, if the effect is neglectable for the overall mass balances, it would be useful to have some quantification about the effect.

Regarding the data, I was wondering why the data are presented until 2017 only. The authors refer to the "Global Glacier Change Bulletin No. 3" by WGMS, where we can find the data up to the balance year 2016/17. On the other hand, the authors also refer to the latest version of the "Fluctuations of Glaciers" database by WGMS from August 2020, which includes data until 2018, also Figure 4 includes some data until 2018. This might cause some confusion. Regarding the available datasets, it should also be specified which data are actually available from the WGMS. If I am right, only the glaciological mass-balances are included, but the geodetic data are not (yet?) included and for instance velocity data are currently not stored by the WGMS.

The paper is very nicely structured and clearly presented. As mentioned above, the description is very detailed, which is useful for readers that are looking for comprehensive information about mass-balance programmes (almost in the sense of a review). On the other hand, I am still wondering whether the paper can be shortened and streamlined, such as the Data section and in particular the Discussion as well as the Conclusions sections. The discussion is very comprehensive and includes many interesting aspects (e.g., climatic influence such as monsoon and jet stream, local weather conditions), which however call for further answers that are not provided in the paper (and which are not in the scope of the paper). I therefore strongly recommend to limit and streamline the discussion to facts that are directly linked and most relevant to the data of the present paper, and to clearly highlight the importance and the value of the presented mass-balance series. In turn, the Conclusions should concisely focus on the key points from the Discussion.

*Regarding the approaches chosen to extrapolate in situ mass balance data to high elevations, a subsection "5.6 Extrapolation of in situ measurements to the accumulation area" has been added in the section "5 Discussion". Additionally, the approaches are described in more detail in the subsection "3.4.1 Point and glacier-wide mass balance".*

*Data up to 2017 is presented in this manuscript. Mass balance data from 2017/18 onwards are part of other studies and are not presented here. The data entry for 2017/18 in the manuscript has been*

*deleted. In the abstract and in the last paragraph of the introduction, the data submitted to the FoG is now specified. The authors submitted glacier data to the WGMS since 2012, including geodetic and glaciological mass balance data and glacier length change data, which are integrated in the newest version of the FoG database.*

*Parts of the section "3 Data and methods" have been shortened, few have been extended in order to address the referees' comments. The sections "5 Discussion" and "6 Conclusions" have been restructured and streamlined. In the study region the data basis (e.g. DEMs, maps, survey base stations) has varying quality and the environmental setting is challenging. Some of the resulting issues and challenges have been addressed in the manuscript because they impact the monitoring programmes and data collection. The mass balance is directly influenced by the climate and we considered it relevant to address how the weather and climate may impact the mass balance.*

More specific comments and suggestions are as follows:

Line 9: Should be formulated more precisely, because "essential variable" might be associated with the ECV Glacier according to GCOS/WMO terminology. Moreover, I would separate between the climate aspects (sensitive response of glaciers to climate change) and applications of the data, such as for runoff modelling.

*"essential" replaced by "important"*

*Sentence rephrased: "The glacier mass balance is an important variable to describe the climate system, and is used for various applications like water resource management or runoff modelling. The direct or glaciological and the geodetic method are the standard methods to quantify glacier mass changes, and both methods are an integral part of international glacier monitoring strategies."*

Line 11: This sentence could be deleted as you anyway state at the end of the abstract that the data are available from the WGMS. One might also ask if the main purpose was the ingestion of the data into the WGMS database (which is of course important but probably not the main motivation).

*Statement deleted about WGMS database, sentence rephrased and connected with following sentence: "Here we present the methods and data for glacier length changes and the directly measured annual mass balances for the first six mass-balance years for both glaciers."*

Line 14: It is not fully clear which series refers to the glaciological method and which one to the geodetic method. General comment regarding the dates: although it becomes mostly clear from the context, it could be helpful to specify the balance years for the glaciological measurements (e.g. 2011/12 to 2016/17). Otherwise the year could refer either to the start of the measurements or to the moment when the observation period is finished.

*Previous sentences rephrased and methods clarified: "Here we present the methods and data for glacier length changes and the directly measured annual mass balances for the first six mass-balance years for both glaciers. For Yala Glacier we additionally present the directly measured seasonal mass balance, and the mass balance from 2000 to 2012 analysed with the geodetic method."*

*Mass balance years have been changed to the format using the start and end year. The original notation using only a single year is based on the definition of "year" by Cogely et al., 2011, p 99: "… The practice when brevity is desirable, regardless of hemisphere, is to identify the hydrological year, mass-balance year or measurement year by the calendar year in which it ends. For example the mass-balance year 2000 began in calendar year 1999 and ended in calendar year 2000." The WGMS uses this notation in the FoG database.*

Line 15: cumulative mass loss → *corrected*

Line 16: missing verb → *corrected*

Line 18: logically not clear, the mass balance of Yala is compared to the mass balance of another glacier. The same also applies to line 19.

*Sentence revised: "Compared to regional mean geodetic mass-balances rates in the Nepalese Himalaya, the mean mass-balance rate of Rikha Samba Glacier is in a similar range, and the mean mass-balance rate of Yala Glacier is more negative because of the small and low-lying accumulation area. During the study period, a change of Yala Glacier's surface topography has been observed with glacier thinning and downwasting."*

Lines 18-20: Was this really investigated in the present paper? Some aspects are discussed but there are many other aspects that could be considered, and given the fact that there are only very few other glaciological mass-balance series available, I would be careful with such general or relative statements (such as "mostly because of. . ." or attribution to "low-lying" areas etc.).

*The sentence has been revised (see comment above). Yes, in the discussion the mass balances of Yala and Rikha Samba Glacier have been compared to regional geodetic mass balances. In the restructured section "5 Discussion", subsection "5.4 Bias by small low-lying glaciers" addresses now the role of Yala glaciers low-lying and small accumulation area and related bias compared to mean geodetic mass-balance rates in the region.*

Line 23: Specify that the glaciological mass-balance datasets are available.

*Available data specified: "The data of the annual and seasonal mass balances, point mass balance, geodetic mass balance and length changes are accessible from WGMS (2021)"*

*For clarification, besides general information (sheets A, B) and glaciologically measured mass balances (sheets E and EE), also point mass balance (sheet EEE), length changes (sheet C) and geodetic balance (sheet D) have been submitted. In February 2021, additionally the seasonal mass balance (EE, EEE) as well as the corrected mass balance from 2017 (shifted elevations and resulting mass balances corrected) have been submitted.*

Lines 28/29: This is a bit too simplified and could be further explained: the glaciological method is indeed a key in international monitoring strategies and is needed for model calibration etc., but for global sea-level rise, the largest ice masses measured by satellite remote sensing are contributing most. Moreover, you could specify what kind of local hazards you refer to.

*Sentence clarified and reworded: "The glacier mass balance is one of the seven headline indicators for global climate monitoring (Trewin et al., 2021) and one of the products of the ECV glacier, besides area and glacier thickness changes (GCOS, 2016). Mass-balance monitoring with the glaciological method is an integral part of international glacier monitoring strategies (Haeberli et al., 2007; Trewin et al. 2021). The glacier mass balance is relevant in various regards, such as climate indicator, for glacier process understanding, the hydrological cycle and modelling, hazards and contribution to sea level rise."*

Line 30: input variable → *"input" replaces "important"*

Line 30: water availability and the change in its availability

*Sentence revised: "As an input variable the mass balance is used to model the water availability and its change, and runoff scenarios for glacierized catchments and downstream livelihoods and ecosystems (Huss and Hock, 2018; Immerzeel et al., 2012; Kaser et al., 2010)."*

Line 34: or: "in support of the United Nations Framework. . ." → *adjusted*

Lines 32-35: 1-2 key references should be given here

*References added: "IGOS, 2007; WGMS, 2020b"*

Line 37: still involve → *corrected*

Line 44: Some studies have focused (?) → *corrected*

Line 45: Not fully clear, did these studies only address the lower parts of the glaciers (where there is debris cover)?

*Sentence clarified: "Some studies focused on ablation and runoff on a high spatial and temporal resolution on clean and debris covered glaciers (Litt et al., 2019; Pratap et al., 2019; Pratap et al., 2015; Immerzeel et al., 2014; Fujita and Sakai, 2014), but rarely measured precipitation and snow accumulation in high altitudes due to challenges such as harsh conditions for precipitation measurements or difficult access to the accumulation zone."*

Line 51: the longest series . . . is found (?) → *verb corrected "is found" instead "are found"*

Line 51/52: with measurements since 2002 → *corrected*

Line 52: why partly?

*Rephrased, "partly with ongoing monitoring" deleted: "Other investigated glaciers in the Indian Himalaya are for example Dokriani, Gara, Gor Garang, Naradu, Neh Nar, Shaune Garang and Tipra Bank (Dobhal et al., 2008; Vincent et al., 2013; Pratap et al., 2015; Azam et al. 2018; WGMS, 2020a)."*

*Background: Uncertain status regarding continuation of monitoring programmes. Dokriani Glacier likely continuation, for other listed programmes no evidence for continuation found.*

Line 55: are glaciological mass-balance records available for both periods?

*Sentence revised: "In the Chinese Himalaya, geodetic mass-balance data measured with differential global navigation satellite system (dGNSS) surveys are available from 1991 to 1993 and 2007 to 2010 for Kangwure Glacier, north of Mt Shisha Pangma and Langtang Valley, and from 2006 to 2010 on Naimona 'Nyi Glacier, in an upper tributary of the Ganges (Liu et al. 1996; Tian et al., 2014; WGMS, 2020a). Additionally, glaciological mass-balance data are available for Kangwure Glacier from 1991 to 1993, and Naimona 'Nyi Glacier from 2006 to 2010."*

*The term dGPS has been replaced by dGNSS in the entire manuscript. The term dGNSS is mentioned here the first time and the abbreviation is written out. For readability, the above-underlined phrase was used instead of "…the differential global navigation satellite system (dGNSS) mass balance data…".*

Line 57: What do you mean with in situ geodetic mass balance measurements?

*Sentence revised: "Glaciological and dGNSS mass balance measurements ..."*

*The geodetic method is used to determine the volume balance of a glacier based on glacier surface elevation changes (Cogley et al., 2011). Currently, dGNSS is the most common method to measure the glacier surface elevation in situ.*

Line 60: rather point mass-balance measurements than index measurements → *corrected*

Line 62: check tense: have these measurements been continued until today?

*Measurements continue, e.g. Yala in 2009, 2012, on Rikha Samba approx. in 2010, 2019.*

Line 63: 1970s, 1980s → *corrected, entire manuscript checked and corrected*

Line 65: I suggest writing Pokhalde and Changri Nup Glaciers (in plural), to make clear that you refer to two different glaciers. The same spelling should be used throughout the entire paper, incl. Yala and Rikha Samba Glaciers – I leave it to the authors what they prefer. → *done*

Line 67: you could specify what they calibrated against. Geodetic mass-balance data?

*Sentence rephrased and information specified: "Wagnon et al. (2020) reanalysed the mass-balance data of Mera Glacier by using geodetic mass balances to calibrate the glaciological measurements from 2007 to 2019."*

Line 68: what do you mean with in situ surface surveys?

*«or in situ surface surveys» deleted*

*Currently, dGNSS surveys are the most common method to survey a surface in situ.*

Line 69: where exactly was this applied?

*"or in situ surface surveys" deleted*

*Fujita and Nuimura (2011) conducted in situ surface surveys with theodolite and dGNSS zigzag lines on Rikha Samba, Yala and AX010 glaciers. In Bhutan, Tshering and Fujita (2016) used dGNSS to repeatedly surveyed the surface of Gangju La Glacier with a dense zigzag pattern. Other researchers conducted repeated surveys along profile lines and transects (e.g. Wagnon et al., 2020).*

Line 78: What was the glacier selected for? → *Specified: "for the Himalayan Glacier Boring Project"*

Line 79: or: it offered / enabled easy access

*corrected "offered" instead "had"*

*Sentence revised: "Yala Glacier was selected for the Himalayan Glacier Boring Project based on a GEN reconnaissance flight in Langtang Valley because it was the only one without debris cover and offered easy access to the glacier and the accumulation area (Watanabe et al., 1984)."*

Line 80: what do you mean with glacier processes?

*Sentence revised: "Comprehensive studies were carried out with a wide range of measurements in the field of glaciology, meteorology and geomorphology (e.g. Murakami et al., 1989; Ono, 1985; Yokohama, 1984)."*

*Background: Japanese scientists and partners conducted research in various parts of Nepal, including the Langtang Valley and the Hidden Valley. Many articles are published on glaciological and related research. The research is comprehensive and on a wide range of topics and for Yala and Rikha Samba Glacier dozens of articles are available. The literature review here aims at giving a glimpse of the work done with a focus on the two glaciers. Work by Japanese researchers relevant for this manuscript are discussed in more detail at other places (totally >30 articles by Japanese scientists cited).*

Line 82: delete the comma (last character in line) → *corrected*

Line 87: documented → *"document" replaced by "assessed"*

Line 87: "for Langtang valley": did they also observe other glaciers in the valley?

*Yes. Sentence revised: "Various studies assessed historic and recent glacier fluctuations at Yala Glacier and in the Langtang Valley" (e.g. Shiraiwa and Watanabe, 1991; Ono, 1985; Yamada et al., 1992; Kappenberger et al., 1993).*

Line 90/91: "highlights the importance of. . .": you could elaborate a bit more on this very important aspect.

*The precipitation seasonality is addressed in the section "2.2 Climate" and further discussed in the section "5 Discussion", subsection "5.5 Interannual variability of winter precipitation and long-term trends of accumulation"*

Line 92: Cogley et al. 2011?

*The reference is correctly cited: Cogley et al. Tracking the source of glacier misinformation, Science, 327(5965), 522, https://doi.org/10.1126/science.327.5965.522-a, 2010.*

Lines 93ff: Here I am wondering about the structure: after a rather long introduction, the motivation of the present study is given here again (compare with second paragraph of the Introduction), which leads to some redundancy. Moreover, you refer to the motivation of establishing a sustainable and long-term mass balance programme. This is a very important aspect; however, the paper is about the mass-balance data themselves and not about the monitoring strategy behind. This should be clarified

here. In the same paragraph, you mention the training of students and professional; another key aspect in maintaining the monitoring programme on the long-term, however, the paper does not deal with this aspect either. These aspects could be given at the end of the Conclusions in the sense of an outlook.

*The paragraph has been reworded and restructured. To eliminate redundancy, the first two sentences or the original paragraph were shifted to the second paragraph of the "Introduction" section, and the third and fourth sentence were deleted. The paragraph on the motivation to establish long-term mass-balance programmes has been revised and the background and context for the glacier monitoring programmes is given instead. The glacier monitoring programmes of Yala and Rikha Samba glaciers are influenced by strategic decisions for the benefit of long-term sustainable measurements. The measurements on Yala Glacier were an integral part of the training activities. The field campaigns had to accommodate with the training requirements, which had an impact on glacier monitoring. From 2011 to 2017 about 50 trainees from various cultural backgrounds and countries participated in 9 trainings in an extreme high-altitude environment. Glacier monitoring campaigns with main focus on scientific measurements can be managed more efficiently and flexible regarding measurements. E.g. the extent of measurement, or the dates of the data collection are partly influenced by the trainings (section "3.1 Data collection": in autumn expeditions starts approx. 1 week after the last festival finished).*

*The restructured and revised paragraph now reads: "In 2011 the HKH-Cryosphere Monitoring Project was initiated in Nepal by ICIMOD, and its partners the Department of Hydrology and Meteorology of the Government of Nepal, Kathmandu University and Tribhuvan University. The project goal was to improve the knowledge and understanding of the cryosphere in relation to climate change and impact on water resources in the HKH region and capacity building. Within this framework mass-balance monitoring programmes were established on Yala and Rikha Samba glaciers. An integral part of the project were trainings every year on the easily accessible Yala Glacier for a few dozens of students and professionals from the Himalayan countries, on one hand to build capacity for sustainable and consistent measurements, and on the other hand to promote the development of further mass-balance programmes in other parts of the HKH Region. Students from Kathmandu University utilized preliminary mass-balance data for their Master theses (Baral et al., 2014; Gurung et al., 2016; Acharya and Kayastha, 2019)."*

Line 96: Add break and start new paragraph with the main objective.

*Obsolete after revisions, see comments for 93ff.*

Line 102: Master theses → *corrected*

Line 104: The focus is not only on Yala glacier.

*Rephrased: "Here we focus on the mass balance of Yala and Rikha Samba glaciers".*

Line 104: seasonal? Bi-annual could be interpreted ambiguously (twice a year or every two years).

*Sentence rephrased: "At Yala Glacier we measured the mass balance twice a year in the field from 2011 to 2017, with remote sensing from 2000 to 2017, and assessed glacier length changes from 1974 to 2016."*

Line 107: For general clarification, I would just refer to the latest version of the database.

*Sentence revised: "The methods are documented for these measurements and data submitted to the WGMS Fluctuations of Glaciers (FoG) database (WGMS, 2021), and other supporting data beyond the scope of the WGMS FoG database."*

Line 109: The questions is also whether the experiences are applicable to other remote areas, too. A short notice in the Discussion or as an outlook at the end could be useful. *→ Sentence deleted*

Line 112: Altitude range of Yala glacier?

*This is an introductory paragraph. More details including the altitude range are provided in section "2.2 Yala and Rikha Samba glaciers", and in Table 1.*

Line 114/115: What kind of data is already available for Yala glacier, and what do you mean with long gaps?

*Sentence deleted; brief literature review already provided in the introduction. Statement about "long gaps" is irrelevant for mass balance research.*

Line 116: mass-balance monitoring programmes?

*Sentence modified and integrated in section "Introduction"*

Line 120, Figure 1: In the overview map, include the name Nepal. As you refer to the different rivers in the text, it could be useful to show them also in the map (I leave this up to the authors how much effort they want to put into the overview map). Regarding the glacier maps, I am though wondering whether the stake locations also include snow pits? For Yala glacier, I am also wondering why there are no stakes in the upper part of the glacier (i.e. in the northern half of the glacier). Is the area too steep or inaccessible? Are the contour lines from the DEM from 2012 for both glaciers? The hill-shaded DEM seems to apply to Yala, the background for Rikha Samba is an ortho image?

*The maps and caption were revised:*

*Both glacier maps:*

- *Legends: Labelled "measurement sites" instead "stakes" because also snow measurements were taken if snow was present.*
- *Maps: Measurements sites labelled.*
- *Contour lines visibility improved*

*Yala Glacier:*

- *The adjacent glaciers from the ICIMOD glacier inventory (modified) were added to give the context of Yala Glacier (please also see comment for Line 142).*
- *In the northern part of Yala Glacier no stakes were installed because the area is inaccessible due to ice cliffs and steep slopes, which has been clarified in the text (comment Line 225).*

*Overview map:*

- *The Himalayan range, rivers and glacier area are shown, including the investigated glaciers and other glaciers mentioned in the discussion. Country boundaries are removed to adhere to the journal's guidelines.*

*Caption revised: "Figure 1: The study sites Rikha Samba and Yala glaciers showing the measurement sites and their location in the Himalayas. At all measurements sites stakes were installed. Snow pits were dug at the top stakes and at selected lower stakes provided snow was present. (a) For Rikha Samba Glacier RapidEye orthoimages from April 2010 were used for the background image and glacier outlines. The contour lines are derived from the SRTM-3 DEM. (b) For Yala Glacier GeoEye-1 orthoimages from January 2012 were used for the background image and in combination with dGNSS data for the glacier outlines. The contour lines are derived from the DEM2012 generated from the GeoEye-1 stereo images. (c) The overview map shows the location of the two investigated glaciers and other glaciers mentioned in the discussion section. The glacier inventory is from ICIMOD (Bajracharya et al., 2014)."*

Table 1: Order of characteristics? Glacier locations could be placed first; glacier type information rather below

*Table reorganized: "Mass balance information", "$ELA_0$" and "$AAR_0$" shifted to the end. "Average slope" added. "Glacier type" mentioned after "Climate" because it's directly related. In the caption added: "The balanced-budget equilibrium line altitude and accumulation area ration are denoted as $ELA_0$ and $AAR_0$." (cf. Cogley et al., 2011)*

Line 142: not clear: if the glacier extents further to north-west, why is this area not included in the analyses? Also, not clear: how can the glacier be separated along the flowline? What do you mean exactly?

*Sentence rephrased to clarify terrain, please also note the revised Figure 1: "The ice body extends further to north-west on a similar elevation range, with steep slopes, ice cliffs and rockfall areas. For the mass-balance analyses, Yala Glacier's drainage basin was separated from the adjacent ice body along the flowline."*

*In subsection "3.4.2 Glacier area and length" the methods is briefly explained: "On the north-west side, the glacier's drainage basin has been separated from the adjacent ice body along the flowline, using flow vectors drawn perpendicular to the contour lines derived from the DEM2012 (Cuffey and Paterson, 2010)."*

*Reference added: Cuffey, K. M. and Paterson, W. S. B.: The Physics of Glaciers, 4th ed. Butterworth-Heinemann/Elsevier, Oxford, 2010.*

Line 144: Do the 5% of the area also include the ice cliffs or only the steep slopes? What is the influence of the ice cliffs? (it is addressed in the Discussion, maybe add make a short separate section there, because it is a special feature).

*Sentence revised: "The glacier faces mainly southwest and the average slope is 25°. Numerous ice cliffs and steep slopes are distributed over the glacier area, but mainly in the northern part of the glacier."*

*The influence of the ice cliffs and steep slopes is now addressed in the section "5 Discussion", subsection "5.1.4 Steep slopes and ice cliffs". Knowledge about the influence of ice cliffs on the mass balance of clean glaciers is limited, mainly to glaciers from low and high latitudes (e.g. Kilimanjaro, Antarctica). The influence of ice cliffs has been studied in the Himalayas on debris covered glaciers, which is less meaningful for clean glaciers.*

*We analysed the slope angles and slope heights that can be represented by DEMs of various resolutions. We found that with a DEM of 30 m resolution, slopes can only be represented if steep slopes have a defined minimum slope height. The original statement that "slopes steeper than 50° only cover 5 % of the glacier area" is only valid for slope heights larger than 35 m. Smaller and steeper slopes are not represented in a DEM of 30 m resolution. Therefore, the statement has been deleted. In the Supplement section S4, the issue related to the representation of steep slopes in DEMs are addressed.*

Line 146: based on radar or other measurements?

*Revised: "measured by ground penetrating radar (GPR)" added*

Line 148: if rockfall covered parts of the glacier, why is it outside the defined outlines?

*Sentence revised: "In the 2015 Nepal earthquake, rockfall covered parts of the ice body, which are next to the defined outlines of Yala Glacier."*

*The revised Figure 1 and first paragraph contribute to the clarification of this sentence (please see comment L 142).*

Line 148/149: this transition zone from debris-covered ice to buried ice (dead ice?) and permafrost ground with ice should be shown on the map. And, how is a clear delineation of the glacier margin possible? Does this situation influence the massbalance observations?

*Sentences revised: "In the 2015 Nepal earthquake, rockfall covered parts of the ice body, which is next to the defined outlines of Yala Glacier. In these parts we find a transition from debris-covered glacier to possible permafrost with refrozen meltwater and buried ice."*

*Background: In a large area, in the vicinity of Yala Glacier a lot of ice of various appearance was observed between big boulders and debris. There is a transition of ice from obviously debris-covered glacier, glacier ice buried by rockfall and refrozen meltwater. However, in large parts the genesis of the ice is unclear. The occurrence of permafrost is very likely. Mapping such a transition zone (e.g. with geoelectrical measurements) was outside the scope our work.*

*Consequently, in the transition zone it is impossible to clearly delineated the glacier margin, which is a common problem. However, the adjacent terminus of Yala Glacier is not affected and the transition zone does not influence the glacier mass balance in this study because the area is outside of Yala Glacier's outlines.*

*Such ice transition zones can be observed near many other glaciers, and new ones will form with the continuous glacier retreat. The ice transition zone explain some of the difficulties to map the glacier area in the vicinity of Yala Glacier. At some locations this will influence the glacier mass balance, and in particular geodetic mass balance calculations that rely on glacier inventories. These are relevant aspects and the reason to mention the transition zone in this article, although there is no influence on our mass balance calculations of Yala Glacier.*

Line 149: Add a break and start new paragraph. → *corrected*

Line 150: Earth's → *deleted*

Line 150: the largest landslide so far documented (?)

*corrected: "Yala Glacier sits on a gneiss bedrock shelf, which forms part of the base from which a large landslide slipped (Weidinger et al., 2002, Takagi et al., 2007)."*

Line 152: ka → *corrected*

Line 153: What do you mean with high glaciation? Glaciation during the ice age(s) or glacier coverage until today. → *"recent high glaciation" corrected to "recent glaciation"*

Line 155: why possibly? From your own field experience, you can maybe directly assess the local meteorological conditions?

*Revised: "possibly" replaced by "probably" (see also next comment)*

*Based on the topography and personal observations the statement is a reasonable assumption. To limited degree meteorological measurements are available between 4000 m and 5500 m a.s.l., but not from above 6500 m a.s.l. on top of the mountain ranges.*

Lines 154-162: Move the aspects about the climate to the next section.

*Sentence rephrased and part of the information shifted to section "2.2 Climate": "The landslide left behind an open topography, with Yala Glacier located within and sheltered by the surrounding high mountains of the Langtang range (>6500 m a.s.l.)."*

Line 162: compared to Yala glacier? → *sentence deleted*

Line 169: is sparse → *corrected*

Line 173: "received": check tense → *past tense, referring to reanalysis data from 1957–2002*

Line 175: the meteo stations you refer to could be added to the map

*AWSs at Yala Glacier are shown in Fig. 1. Other stations are outside the map boundaries.*

Line 177: rather mean annual precipitation (also line 181) → *corrected*

Line 178: start new paragraph for Rikha Samba glacier

*In this section, the topics of the paragraphs are the general setting, precipitation, temperature, cloudiness, and wind. The paragraphs were kept as they were.*

Line 189/190: not clear: what do you mean here? How is it indicated? From the observed radiation?

*Sentence revised: "The sky in the Nepal Himalaya is generally clear in the post-monsoon and winter season (Fujita et al., 2001)." "as indicated by the incoming solar radiation" deleted.*

Lines 189-200 and entire sub-chapter 2: Regarding the overall structure, I suggest to bring first the more general facts about the setting incl. climate, and then specifically address the two glaciers.

*The structure of the section "2.2 Climate" is the general setting, precipitation, temperature, cloudiness, and wind. Each paragraph is introduced by indicating the climatic parameter addressed. The paragraph on precipitation has been improved by adding a better introductory sentence: "Precipitation has been analysed for the Langtang Valley and Rikha Samba Glacier based on reanalysis data and field measurements (Immerzeel et al., 2012; Racoviteanu et al., 2013; Fujita et al., 2001)."*

Line 199/200: does this apply in general or especially at Yala glacier?

*Sentence revised: "During monsoon from June to September the wind speeds at both glaciers are lower with a smaller variability."*

Line 200: wind speeds → *corrected*

Line 202: was monitored → *corrected*

Line 202: rather write "from autumn 2011 to. . ."

*This is an introductory paragraph for the following sections and the sentence is kept as it is. Details about the measurements follow below.*

Line 206: avoid abbreviation and "biannually"

*"HKH-CMP" deleted; the project is now introduced in the section "1 Introduction"*

*"Biannually" replaced by "twice a year"*

Line 207: measurements → *corrected*

Line 218: stakes → *corrected*

Line 219/220: unclear, should be further explained. Also, in line 221/222 you refer to ablation and accumulation occurring during the same time. For a better general understanding of the seasonal mass-balance evolution at the glaciers, it would be useful to have (1) a general statement about potential accumulation/ablation seasons and (2) how this affects the mass-balance measurements (and (3) how mass-balance measurements are performed in practice).

*In the section "Study area and climatic setting" in Table 1, it is stated that Yala and Rikha Samba glaciers are summer-accumulation type glaciers in a monsoon climate. In the introductory paragraph about the study area, sentences have now been added specifying the main characteristics of summer-accumulation type glaciers, mentioning the main accumulation and ablation seasons. The first paragraph in section "3.1.1 In situ mass balance measurements" has been modified. Many readers might be unfamiliar with summer-accumulation type glaciers and the related measurements. A brief description of summer-accumulation type glaciers, how it affects the mass balance measurements and how measurements are performed in practice is now provided in section S1 of the Supplement.*

*The introductory sentence about the study area now reads: "Both glaciers are summer-accumulation type glaciers (Ageta and Higuchi, 1984), which are characterized by an overlapping main accumulation and ablation season during the monsoon season (Fig. S1). A brief description of summer-accumulation type glaciers and mass-balance measurements is provided in the Supplement (section S1)."*

*The introductory paragraph for subsection "3.1.1 In situ mass balance" now reads: "The in situ mass balance was measured following Kaser et al. (2003), taking into consideration aspects in the ablation and accumulation area specific to summer-accumulation type glaciers (for details see Supplement, section S1). In the ablation area, the mass balance is measured with bamboo stakes. If snow is present, its depth is usually measured at each measurement site, and at selected stakes the snow density and profile are also recorded. In the accumulation area, snow pits are dug or cores taken, and the snow profile, depth and density recorded. Additionally, several snow probing measurements are taken. Bamboo stakes mainly mark the measurements sites, but in absence of snow pit data they are also used for the mass-balance calculation, in particular in the case of a negative mass balance. The snow pit measurements are only reliable if the previous measurement surface can be clearly identified, e.g. when marked with a sawdust layer. Difficulties arise in the accumulation area, if the cumulative ablation temporarily exceeds the cumulative accumulation during the measurement period (Fig. S2). The exceeding ablation is not represented in a snow pit measurement and likely impacts the sawdust layer. Stake readings are less reliable because the underlying snow and firn layers compact over time and may push or pull the stake up or down. "*

Line 221: can be clearly identified → *corrected*

Line 225: why is the network stretched along one line? See also my comment regarding Figure 1.

*The Paragraph has been rephrased to clarify why no measurements were taken in some parts of the glacier: "On Yala Glacier, the measurements stretch along a line established in the past by Japanese researchers (Fujita et al., 1998). In the lower part a few stakes were initially added in a transect. Since the glacier has been shrinking, a second row of stakes was installed parallel to the original line in November 2016, in an attempt to maintain measurements also in future when the glacier retreats beyond the current stake locations. In the northern and highest parts of the glacier no measurements were taken because steep terrain, crevasses and ice cliffs make access difficult."*

Line 229/230: The effect of the ice cliffs and steep slopes should be placed in the Discussion.

*The sentence has been deleted and is addressed in the discussion.*

Line 234: Snow depths and snow pits incl. snow density. . .

*Sentence rephrased: "Snow depth was probed, and the density measured in snow pits, but sawdust was spread only during few occasions and found only once, making accumulation measurements challenging."*

Line 238: kg m$^{-3}$ → *corrected (paragraph deleted here but kept and corrected in section "4 Results")*

Line 240ff: This section is very detailed (too detailed). You could just directly refer to Table S1.

*The section has been shortened. It is common to provide technical details about dGNSS surveys (e.g. Wagnon et al., 2020; Tshering and Fujita, 2016).*

Line 245: at best? → *"at worst" deleted*

Line 255: What are the velocity measurements used for?

*Glacier velocity data is supporting information for the interpretation of glacier data. It can give an indication about glacier dynamics and flow direction. As stated in the discussion chapter, the lowering ice velocities of Yala Glacier give an indication about the glacier's downwasting.*

Line 258: Explain that "Schneider" is a particular map name (Arbeitsgemeinschaft für vergleichende Hochgebirgsforschung?)

*Sentence revised: "The maps include the Survey of India, the so-called Schneider and the Nepal topographical maps published in 1965,…" Additional information about the map is provided in Table S2.*

*Background: In Nepal, the "Alpenvereinskarte Langthang Himal Ost" is known as the "Schneider map". The map was published within the framework of the Alpenvereinskartographie by the Oesterreichischer Alpenverein (Austrian Alpine Club) in 1990. The "Arbeitsgemeinschaft für vergleichende Hochgebirgsforschung" was not involved, and published different maps.*

Section 3.2 is interesting to read but very detailed. Maybe some of the information could be placed in a table instead to shorten the section.

*The section has been revised and shortened, sentences have been deleted, or statements are expressed more concisely. The Tables S2 and S3 in the Supplement provide more detailed information about maps and satellites. The shortened section now reads:*

*"For Yala Glacier, various maps were compared and evaluated for their suitability for area, volume and frontal change analysis. The maps include the Survey of India, the so-called Schneider and the Nepal topographical maps published in 1965, 1990 and 1995, the map by the Japanese Glaciological Expedition Nepal (GEN) map (Yokoyama, 1984) and glacier outlines from the ICIMOD glacier inventory of Nepal (Bajracharya et al., 2014; Table S2). The GEN map and glacier inventory data were used; however, despite good quality no other maps could be used because of transformation issues and inconsistencies. The GEN map is based on a ground photogrammetric field survey in 1981 (Yokoyama, 1984). The photo point was about 2 km from the glacier terminus in 1981 on a lower location; consequently, the exposing axis is almost parallel to the glacier surface. We found a distortion and mismatches at the ridge and at the south-east and north-west side of the glacier. We georeferenced the map with the GeoEye-1 orthoimage from 2012 to calculate the frontal variations but did not use it for area or geodetic mass-balance analyses.*

*Satellite images were used to delineate glacier outlines and termini of both glaciers, and to calculate the geodetic mass balance of Yala Glacier (Table S3). The SRTM-3 DEM (SRTM-3) is the third version of the DEM from the Shuttle Radar Topography Mission (SRTM) and is generated based on data from 2000. The spatial resolution is about 90 m, with an absolute vertical accuracy of ±16 m and a vertical reference to the WGS 84 EGM96 geoid (Rabus et al., 2003). The penetration of the SRTM C-band beam in snow, firn and glacier ice is an issue that results in a lower accuracy especially in the accumulation area (Kääb et al., 2012, Berthier et al., 2006). SRTM-3 was resampled to 30 m for the geodetic mass-balance calculation of Yala Glacier. The SRTM-1 DEM was used for the mass-balance analysis of Rikha Samba Glacier. It is based on the SRTM-3 data from 2000 but was released with an improved resolution of about 30 m.*

*The GeoEye-1 is a commercial high-resolution stereo satellite image with 0.5 m spatial and 8 bits per pixel radiometry resolutions. The stereoscopic images from 15 January 2012 were used to generate a DEM (DEM2012) for Yala Glacier to calculate the glacier-wide geodetic mass balance, and the orthoimage was used to delineate the outlines.*

*We used Landsat images for various purposes. A Landsat 8 image acquired on 18 November 2013 was used to collect horizontal reference (x, y) and the SRTM-3 for the vertical reference (z) for ground control points (GCP) to georeferenced the GeoEye-1 images, and tie points for DEM generation for Yala Glacier. A Landsat 7 Enhanced Thematic Mapper (ETM+) image from 2000 helped to identify the outlines of Yala Glacier for the geodetic mass balance and to analyse frontal variations. We analysed terminus changes of Rikha Samba Glacier using a Landsat 4, Landsat 7 ETM+ and two Landsat 5 Thematic Mapper (TM) images from the years 1989, 2001, 2006 and 2011, respectively. RapidEye images from 25 and 27 April 2010 were used to delineate the outlines of Rikha Samba Glacier.*

*A Hexagon KH-9 image from November 1974 was used for a frontal variation analysis of Yala Glacier. Other Hexagon images were found unsuitable for area and volume analysis because of void areas, or cloud and snow cover in the images at other times of the year. Additionally, it was difficult to delineate*

*the glacier at the north-west and south-east side without contour lines to derive the flowlines at that time.*

*For this study, we adopt the projection system WGS 1984, UTM Zone 44N and 45N for Rikha Samba and Yala glaciers, respectively. We used the local projection system called Modified Transverse Mercator, with false easting 500,000 m and scale factor of 0.9999 at the central meridian 84° E and 87° E for Rikha Samba and Yala glaciers, respectively."*

Line 305: not clear, rewrite the last sentence

*Sentence revised: "The DEM was used for to analyse the mass balance analysis of Yala Glacier with the geodetic and glaciological method."*

Line 332: Abbreviations should be written out when they appear for the first time.

*Corrected and Table 1 adjusted: "The glacier-wide mass balances, the equilibrium line altitude (ELA) and accumulation area ratio (AAR) were calculated based on the interpolated mass balance gradient derived from the point measurements following a similar method used by Wagnon et al. (2013)."*

Line 358: based on the measurements by whom? According to Figure 3, there are no data.

*Context changed and sentence rephrased: "For the accumulation area, no characteristic gradients could be identified because only few measurements were available."*

Line 358: balance year 1998/99? → *corrected*

Line 360ff: Are there large changes in the area over the observation period? If not, the difference between conventional and reference-surface balance is small and a corresponding note should be added in the text.

*We did not analyse area changes of Yala and Rikha Samba glaciers between 2011 and 2017.*

*Background: As documented in section "3.2 Maps, satellite images and DEMs" and Table S2, for Yala Glacier we made efforts to assess various maps and satellite images with the goal to analyse the geodetic mass balance or area changes. However, as mentioned we faced a range of issues such as snow cover, transformation and scale problems. While accurate maps and DEMs with high resolution and regular updates are a standard e.g. in Europe or North America, this is not possible in Nepal because of its extreme environment. In our work we faced repeatedly challenges such as distorted maps or inaccurate elevations, which made it hard to georeferenced satellite images for areas as small as Yala Glacier.*

Line 375/376: not fully clear what you mean with regression lines caused by outlying points

*Sentence rephrased:" The accuracy of the ELA and AAR were estimated by shifting the regression lines based on point measurements deviating from the initial regression line."*

Line 380: not clear: what has been separated? Why based on the contour line method? It is not evident from Figure 1 (map).

*Sentence revised: "On the north-west side, the glacier's drainage basin has been separated from the adjacent ice body along the flowline, using flow vectors drawn perpendicular to the contour lines derived from the DEM2012 (Cuffey and Paterson, 2010)."*

Line 402: Not clear: earlier in the text you state that the ELA is derived from the mass balance gradient; is it also observable in the field? "balanced ELA": write ELA0 as in other occasions in the paper and state at some point that ELA0 (AAR0) is the balanced ELA (AAR).

*Sentence revised and the term "balanced ELA" doesn't appear in the manuscript anymore: "The accumulation and ablation areas were separated by an estimated ELA of 5350 m a.s.l."*

*Background: for the calculation of the geodetic mass balance the process required the ELA to be estimated for the respective time period. This ELA differs from the one calculated from the annual mass balance gradient data and the $ELA_0$.*

Line 410: What is/was the purpose of this additional profile line? Was it used to calibrate the geodetic data? Was there a correction applied? Or was there any benefit for the in situ glaciological measurements?

*The profile line has been addressed in subsection "3.1.2 GNSS surveys", and the revised paragraph now reads: "The glacier surface profiles of Yala and Rikha Samba glaciers were repeatedly surveyed with dGNSS, along a longitudinal profile and three and two cross-profiles, respectively, but only data from May 2012 from Yala Glacier are presented here. Already Sugiyama et al., (2013) surveyed the profile line in 2009. The repeated measurements provide the opportunity to further analyse the mass balance with an independent complementing method (Wagnon et al., 2013, 2020)."*

Line 434: climate stations? → *Corrected: "…however, AWSs on and near…"*

Line 445: gradient for Yala Glacier from 2011/12 to 2016/17. Delete row with "1999" → *corrected*

Line 450: Not clear whether there was actually a measurement at Rikha Samba in 1999 or if this should be 2011. However, if the latter applies, the caption should read 2010 to 2017, or better 2010/11 to 2016/17.

*Caption revised: "Table 3: Mass balance (B) measured with the glaciological method, ELA, AAR and the lower and upper mass-balance gradient for Rikha Samba Glacier for the mass-balance years 1998/99, and from 2011/12 to 2016/17. We did not calculate the ELA and AAR for Rikha Samba Glacier for 2011/12, 2013/14 and 2014/15 due to the very few data points. For the mass-balance year 1998/99, the point measurements collected by Fujita et al. (2001) were used."*

*In the subsection "3.4.1 Point and glacier-wide mass balance", it is mentioned that data from Fujita et al. (2001) was used to calculate the glacier-wide mass balance for the mass balance year 1998/99. For clarification it is now also mentioned in the caption of Table 3.*

Line 463: The differences between seasonal and annual balances should be further explained (in the Discussion).

*In the section "5 Discussion" in subsection "5.1.2 Seasonal mass balance" the differences in annual and cumulative seasonal balance in 2011/12 and 2014/15 are now addressed.*

Lines 474-477: Repetition of what has previously been stated.

*Lines deleted in the "Data and Methods" section, but kept in the "Results" section.*

Line 486: m a.s.l. → *corrected*

Line 488: linear regression? → *corrected*

Line 493: seasonal and annual balances The consistent gradient for the glacier: an interesting finding that should be mentioned in the abstract.

*The sentences have been revised to better explain in words what can be seen in Figure 2. The described gradients show a common pattern, especially considering that the lack of sufficient data in the accumulation area prevented identifying a characteristic gradient in the accumulation area. The findings are not unusual and there is no need to describe them in the abstract. Examples of common mass balance gradients can be found in the Global Glacier Change Bulletins of the WGMS.*

*The revised sentences now read: "Figure 2 shows the characteristic gradients for the annual and seasonal balances of Yala Glacier that remain relatively constant over the investigated time period. However, additional measurements in higher elevations would have allowed to identify a gradient in the accumulation area for the annual and the summer balance."*

Line 499: Your statements here raise the question of why there are no measurements in the higher areas of the glacier(s). Are these areas too high? Too steep? Too dangerous? This should be specified and clarified at some point.

*In subsection "3.1.1 In situ mass balance", sentence revised: "In the northern and highest parts of the glacier no measurements were taken because steep terrain, crevasses and ice cliffs make access difficult."*

Lines 500-509: move this paragraph to the Discussion → *paragraph shifted*

Line 528: The thinning rate along the profile line is not clear (-1.1 m a-1); is it the mean along the line? Would a mean even make sense? What is the purpose of this profile line?

*The revised sentence is: "The mean thinning rate along the profile line is -1.1 ±0.13 m $a^{-1}$."*

*In the section "5 Discussion", subsection "5.1.1 Annual mass-balance rates" sentences have been added discussing the purpose of the mass balance of the profile lines: "The profile line has been surveyed repeatedly, the first time by Sugiyama et al. (2013) in 2009 and in subsequent years by our team. The future analysis of the geodetic mass balances along the profile lines and transects is planned as supporting and independent method for the analysis of the mass balance (Wagnon et al., 2020, 2013)."*

For such a long time period, only the mean makes sense. Along profile line Line 529: m a.s.l.

→ *corrected*

Line 538: Table 4 should be moved to the Discussion. → *done*

Line 540: In Table 4, you refer to WGMS (2019). The reference is missing and anyway the question is what the original source is. In the table, I would explicitly write "glaciological method".

*"direct measurements" replaced by "glaciological method"*

*The initial compilation of some of the data is based on the FoG dataversion 10.5904/wgms-fog-2019-12, when the most up to date data was only available from the FoG database. The FoG reference has been updated short before submitting the manuscript in early September 2020. In the meantime, new references have been published and are updated in the Table 6 according to best knowledge. These are:*

- *Wagnon et al., 2020, including supplementary information*
- *Mandal et al., 2020*

*Please note, for the geodetic mass balance calculation for Yala and Rikha Samba glaciers 2000–2016, the paper by Brun et al., (2017) has been provided for the description of the data. The actual values for the two glaciers have been retrieved from WGMS (2020a), no other source could be accessed.*

Lines 545-552: should be moved to the Discussion → *done*

Line 550: Maybe also add Chhota Shigri glacier (cf. Discussion) → *Chhota Shigri Glacier added*

Line 551: regarding Mera and Pokalde glaciers: where are these glaciers exactly located? They could/should be added to the overview map.

*The glaciers have been added in the overview map in Figure 1.*

Line 558/559: The comparison of the retreat rates of both glaciers is relative because the glaciers are different in size and form. For Yala glacier, it might also be more straightforward to assess area changes instead of length changes (?). Or to add area changes in addition to the length changes.

*The revised sentence now reads: "For Rikha Samba Glacier, between 1989 and 2013 the average retreat rate and total retreat was -18.0 m $a^{-1}$, and -431 m, respectively."*

*In this section, the results of the glacier length changes are presented, which have been reported for the FoG database in 2015. We discuss length changes in section "5 Discussion", subsection "5.1.3 Glacier length changes, flow and downwasting" and put them into context of past records. We are aware that glacier length changes depend on the topography, that their climate signal is delayed and indirect and depend on the glaciers' individual response times. The ECV product "glacier length" (Haeberli et al., 2007) has been changed to "glacier area" (GCOS, 2016) only in recent years and therefore we initially put priority on reporting glacier length changes. If it was easy, we would have also analysed area changes. It wasn't easy. As already stated in section "3.2 Maps, satellite images and DEMs", we assessed satellite images and maps but found them unsuitable for area analysis.*

Line 579: Figure 12 should be moved to the Discussion. → *Done*

Line 583: The displacement of the stakes is interesting and allows the link to the velocity measurements. → *Yes, thanks*

Line 585: why is the profile line from Sugiyama et al. added? And is this the same profile line as mentioned before?

*Profile line has been removed from the Figure because it is not relevant here. And yes, it is the same profile line that we surveyed repeatedly for further comparison.*

Line 595: All glaciers and benchmarks as used in the Discussion could be shown in the overview map. For clarification, it should also explicitly be stated

*In Figure 1, the overview map (c) has been revised, showing the investigated glaciers and the glaciers mentioned in the Discussion. In the Discussion, references to Fig. 1 is made.*

Line 614: The paragraph here as well as the subsequent ones refer always to geodetic mass balances; this should be clearly stated or structed in a clear way so that both observations types are not mixed up.

*The section Discussion has been restructured. Geodetic and in situ data are now explicitly mentioned.*

Line 637ff: What does this mean for assessing glacier mass balances over the entire Himalaya? Is it possible from the available data to draw general conclusions? The authors might elaborate a bit more on this question, which also allows to place the mass-balance data from Yala and Rikha Samba into a larger context and to streamline the Discussion (cf. my general comment above about the Discussion).

*The section Discussion has been restructured. The mass-balance data of Yala and Rikha Samba Glacier have been put into a larger context in the "5 Discussion", subsection "5.3 Comparison of in situ glacier mass balances in the Himalaya".*

*In the subsection "5.4 Bias by low-lying glaciers" the closing sentences have been reorganised and now read: "The bias introduced by low-lying glaciers result in the overestimation of negative mass balances in the region (Gardner et al., 2013). It highlights the importance of investigating large glacier elevation ranges, measuring mass balances in the accumulation areas and precipitation data in high altitudes."*

Line 640: Such a bias → *corrected*

Line 641: here the term mass budget is used, at other occasions the term mass balance is used. You may check for a consistent use throughout the paper. The same applies to the term net balance (e.g. in line 649 or Figure 14), which should be avoided (cf. the mass-balance glossary by Cogley et al., UNESCO, 2011).

*The terms "budget" has been replaced by "balance", and "net" has been replaced with a suitable term in the manuscript, except for "balanced-budget equilibrium line altitude (ELA$_0$) and balanced-budget accumulation area ration (AAR$_0$)" and "net accumulation", where the terms are used according to Cogley et al. (2011).*

Line 651/652: delete sentence → *deleted*

Line 660/661: 1980s, 1990s (see also line 680 and other occasions) Paragraph from line 649: It should be better worked out why this context is important for the discussion of the measurements presented in this study.

*Regarding 1980s, 1990s etc.: entire manuscript checked and corrected*

*The context is now addressed in the restructured section "5 Discussion", subsection "5.5 Interannual variability of winter precipitation and long-term trends of accumulation".*

Line 686: If you specifically address particular stakes with names, then these should also be labelled in the map. → *done in revised Figure 1*

Line 700: There is a break and hence a new paragraph should be started here.

*Obsolete after restructuring the section "5 Discussion"*

Line 701: why are the measurements on Yala glacier representative? Could you explain that a bit more in detail? → *sentence deleted*

Line 707: The general character of Yala (and Rikha Samba) glacier(s) should be stated at the beginning of the paper (it also relates to my previous comment regarding the ablation/accumulation season, see lines 219/220). In line 708, there is now a contradiction to the what has just been stated before. This should also be clarified.

*The discussion has been restructured and rephrased. The introductory sentence repeating the character of Yala Glacier and results has been deleted. Please also see response to comment 219/220.*

Line 715: during favorable conditions → *corrected: "in" deleted*

Line 723: Is this statement really needed for the discussion? → *Statement deleted*

Line 738ff: rather results, can be deleted here or just refer to Figure 11, but then also an explicit link to the mass-balance data should be made. On the other hand, the Little Ice Age extent is mentioned here for the first time, however, the location is not shown on the length change plot (either add or delete entirely).

*The discussion has been restructured and the frontal variations of the two glaciers are now discussed separately. LIA is now written out as Little Ice Age.*

*Please note, in the section "5 Discussion" we put our own results into the context of measurements from earlier years, based on studies conducted by Ono (1985), Yamada et al. (1992), Kappenberger et al. (1993), Fujita et al. (1998) and Fujita et al. (2001). In the section "Introduction" we refer to some of the respective work. Their results are not displayed in Fig. 9 and 10 (in section "results"; previously Fig. 10 and 11) but surely would have been interesting to see. The former glacier extents and data on moraines were not available to us in a georeferenced digital format. As addressed in the responses for comment "Section 3.2" and "Line 360ff", there are in general problems to georeferenced maps because of transformation issues. However, the respective references provided maps and photos. In the section "results", Table 4 (previously 5) has been corrected and the three entries with "Source: Fujita et al. 2001" have been deleted because these are not our results.*

*We submitted frontal variation data (former ECV products) to the FoG database and hence found them relevant to be discussed. The mass balances data are a direct climate signal and the glacier length changes are the delayed response to the mass changes. However, the glacier dynamics of Yala Glacier is "disturbed" because of the downwasting, which makes it problematic to use the response time equation by Jóhannesson et al. (1989)\*. The response time of Rikha Samba Glacier is according to Jóhannesson et al. (1989) approximately 43 years. We do not have mass balance data from several decades to draw conclusions about the observed glacier length changes.*

*\*Jóhannesson, T., Raymond, C., and Waddington, E. (1989). Time-scale for adjustment of glaciers to changes in mass balance. Journal of Glaciology, 35(121):355–369.*

Line 756: the mass-balance data shown only cover a few years, so it is probably more appropriate to write about an overall retreating. Or can you exclude that there were more balanced (or maybe even positive) years over the last couple of decades?

*Sentence revised: "Both Yala and Rikha Samba glaciers shrank and retreated in the last couple of decades. The geodetic mass balance of Yala Glacier showed a mass loss of -10.49 ±7.41 m w. e. from 2000 to 2012, at an annual rate of -0.74 ±0.53 m w.e. $α^{-1}$. The cumulative in situ mass balances for Yala and Rikha Samba Glacier were -4.80 ±0.69 m w.e. and 2.34 ±0.79 m w.e., and the annual mass-balance rates -0.80 ±0.28 m w.e. $α^{-1}$ and -0.39 ±0.32 m w.e. $α^{-1}$, respectively. From 1974 to 2016, Yala Glacier retreated 346 m, and from 1989 to 2013 Rikha Samba Glacier retreated 431 m."*

Line 760/line 770: Avoid new references or rather move to Discussion

*Conclusions revised and reference now also mentioned in the discussion*

Lines 764ff: Does this refer to Yala and/or Rikha Samba? → *statement deleted*

Line 772: m a.s.l. → *corrected*

Line 773: This aspect is not mentioned in the Discussion, isn't it? It is an important aspect because it also influences the glacier's mass balance (and there is also another comment above about conventional vs. reference-surface mass balance), therefore I would discuss it before and draw one conclusion that is then presented here at the end.

*The downwasting is addressed in "5 Discussion" and has been revised in the new subsection "5.1.3 Frontal variations, glacier flow and downwasting". Area changes were not assessed.*

*In the section "6 Conclusion" the revised statement now reads: "Yala Glacier experienced downwasting, indicated by the observed changing surface topography between 2011 and 2017 and decreasing ice flow velocities. Over the course of the years, most of the stakes could not be reinstalled at the original coordinates, either because of new crevasses, or significant changes of the surface features at the original site. The downwasting and small accumulation area at low elevation compromise the long-term monitoring of Yala Glacier."*

Line 784: Can you give a time horizon regarding the survival of the glacier? Or are there any projections for the region?

*A sentence is added addressing the expected survival of the glaciers: "Under the recent climate it can be expected that Yala Glacier will disappear over time but not Rikha Samba Glacier (Fujita and Nuimura, 2011)."*

Lines 786-793: In my view, these sentences can be deleted. → *deleted*

Lines 794ff: In fact, the inclusion of geodetic mass-balance data is an integral part of the monitoring strategy within the Global Terrestrial Network for Glaciers (GTN-G), to compare and eventually calibrate both glaciological and geodetic time series (which is not the case for Yala glacier). This should be clearly stated here. In turn, I would rather conclude how in your case the geodetic method helped to interpret the glaciological mass-balance measurements (and will eventually help to calibrate the series at a later stage).

*The "6 Conclusions" have been revised and a section "7 Recommendations" has been added. In the "6 Conclusion" we now state: "The glacier mass-balance programmes for the two glaciers have been designed using a comprehensive monitoring strategy following the international glacier monitoring strategy within GTN-G (WGMS, 2020b; Haeberli et al., 2000)." In "7 Recommendations" we now state: "Geodetic mass-balance analyses overlapping the timeframe of the glaciological measurements of Yala and Rikha Samba glaciers are needed. The complementing approaches assure keeping the annual signal of the glaciological measurements, and reduce the uncertainty introduced for example by unmeasured parts of the accumulation area or glacier steep slopes."*

Line 803/804: are available and are published

*Revised: "The data are available from the Fluctuations of Glaciers Database http://dx.doi.org/10.5904/wgms-fog-2021-xx (WGMS, 2021)."*

Line 805: delete "World Glacier Monitoring Service, Zurich, Switzerland" → *corrected*

Line 810: mass balance data → *corrected*

---

## Referee Report (RR1)

[referee-annotated manuscript omitted]

---

## Author Response (AR2)

**Authors' response to review reports**

We would like to thank an anonymous referee and Argha Banerjee for the careful second review and constructive comments that helped to further improve the manuscript essd-2020-272.

We responded to all comments raised by the referees point by point in blue italic font. We thoroughly checked the language of the entire manuscript and made various minor corrections as highlighted in the document in track-change mode. The line numbers refer to the "essd-2020-272-manuscript-version3.pdf" submitted in April 2021.

**Changes other than requested by referees**

*Line 434, Table 2: Values of mean, STD and sum corrected for BW+BS, the caption edited:*

Table 2: Mass balance (B) measured with the glaciological method, winter balance ( $B_W$ ), summer balance ( $B_S$ ), ELA, AAR and mass-balance gradient for Yala Glacier from 2011/12 to 2016/17. The summer balance from 2011/12 and winter balance from 2014/15 (\*) have not been reported to the WGMS and are discussed in subsection 5.1.2 Seasonal mass balance.

|           | В                | Bw               | Bs               | B w+ B s | ELA             |      | db/dz                                   |
|-----------|------------------|------------------|------------------|------------------------|-----------------|------|-----------------------------------------|
| B year    | (m w.e.)         | ( m w.e.) | ( m w.e.) | ( m w.e. )      | (m a.s.l.)      | AAR  | $(m \text{ w.e.} (100 \text{ m})^{-1})$ |
| 2011/12   | $-0.86 \pm 0.40$ | 0.16             | -0.20*           | -0.03                  | $5454\pm\!\!30$ | 0.28 | 1.14                                    |
| 2012/13   | $-0.01 \pm 0.29$ | 0.36             | -0.35            | 0.01                   | $5380 \pm 20$   | 0.48 | 0.99                                    |
| 2013/14   | $-0.61 \pm 0.27$ | 0.27             | -0.99            | -0.73                  | $5431 \pm 20$   | 0.35 | 1.18                                    |
| 2014/15   | $-1.18 \pm 0.26$ | 0.54*            | -1.12            | -0.59                  | $5510 \pm 40$   | 0.13 | 0.90                                    |
| 2015/16   | $-0.61 \pm 0.23$ | 0.19             | -0.79            | -0.60                  | $5444 \pm 20$   | 0.31 | 0.93                                    |
| 2016/17   | -1.54 ±0.20      | 0.20             | -1.75            | -1.54                  | 5518 ±20        | 0.12 | 1.10                                    |
| Mean      | $-0.80 \pm 0.28$ | 0.29             | -0.87            | -0.58                  | 5456            | 0.28 | 1.04                                    |
| STD       | 0.53             | 0.14             | 0.56             | 0.56                   | 52              | 0.14 | 0.12                                    |
| 2011-2017 | $-4.80 \pm 0.69$ | 1.72             | -5.21            | -3.48                  |                 |      |                                         |

Line 452, Figure 3 (right) slightly updated: massbalance years are now also in the format "201x/1y" instead of "201y", similar to Figure 2.

Figure 3: Point mass balance, gradients and hypsography of Rikha Samba Glacier for the mass-balance years 1998/99, and 2011/12 to 2016/17.

*Line* 577: *Figure 11 caption revised: "(Figure adapted from Sugiyama et al., 2013)" deleted. The Figure already uses different data including different base data, a different coordinate system, and different results.*

Line 654: Figure 12 has been redesigned and is now shown in colour. We show our own results and data from other sources as declared in the caption. The caption has been revised: "(adapted from Fujita et al. 1998)" has been deleted.

Figure 12: Altitudinal distribution of the surface flow speeds of Yala Glacier, surveyed in 1982 by Ageta et al. (1984), 1996 by Fujita et al. (1998), 2008 to 2009 by Sugiyama et al. (2013) and from 2012 to 2014 in this study.

Supplement: Figure S1 has been further redesigned using a common format. The caption has been revised: "(adapted from Ageta and Higuchi, 1984)" deleted.

**Responses to comments of Report #1, by anonymous referee #2 submitted on 12 May 2021**

**Overall assessment of re-submission:**

The authors made a great, if not exceptional, effort to answer the reviewers' questions in great detail, which eventually resulted in a substantial reformulation of large parts of the manuscript incl. changes made to Figures. There is also a largely extended supplementary section. With that, the manuscript is very comprehensive, and I do not see what else needs to be added.

As the underlying data is key for the present paper, I should also mention that the presentation of the data and their availability is now very clearly described in the paper (this point was brought up by both reviewers that assessed the first version of the paper). Another point referred to the extrapolation of the glacier mass balance to higher elevations, which the authors carefully addressed with a dedicated separate section.

Overall, the paper is very comprehensive and it includes really all (or nearly all) aspects that are relevant when assessing glaciological mass balances and especially when implementing a (new) glaciological mass-balance programme. The Introduction offers many references to other studies; it is almost a complete overview on previous (glaciological) work for the particular study region, which is rather long but I think it will also be useful for other studies, too. The following description of the study site and its climate is written very carefully (the sub-section on climate is very comprehensive, but it will be useful for readers who are less familiar with the region). Presentation of results and their discussion are very clear, and there is extensive supplementary material referred to. Finally, I think the new Recommendations section will be very useful, too.

In addition, I also noticed the following points that could be addressed before final publication:

Line 12: you could first list the mass balance and then the length changes, as the main focus of the paper is on (glaciological) mass balance.

Sentences rephrased:

"Here we present the methods and data of the directly measured annual mass balances for the first six mass-balance years for both glaciers from 2011/12 to 2016/17. For Yala Glacier we additionally present the directly measured seasonal mass balance from 2011 to 2017, and the mass balance from 2000 to 2012 obtained with the geodetic method. In additions, we analysed glacier length changes for both glaciers."

Line 15: It is not quite evident why there are two different observation periods for Yala Glacier (2000-2012 and 2011-2017), which also partly overlap (or maybe do not overlap, because the second period is actually 2011/12-2016/17 (?).

Sentences rephrased: "The directly measured average annual mass-balance rates of Yala and Rikha Samba glaciers are -0.80 ±0.28 m w.e. a-1 and -0.39 ±0.32 m w.e. a-1, respectively, from 2011 to 2017. The geodetically measured annual mass-balance rate of Yala Glacier based on digital elevation models from 2000 and 2012 is -0.74 ±0.53 m w.e. The cumulative mass loss for the period 2011 to 2017 for Yala and Rikha Samba glaciers is -4.80 ±0.69 m w.e. and -2.34 ±0.79 m w.e., respectively. The mass loss on Yala Glacier from 2000 to 2012 is -8.92 ±6.33 m w.e."

Line 18/19: "mass-balance rates"  $\rightarrow$  corrected

Line 26: The new FoG database version will be 2021-05 and it should appear very soon on the WGMS doi landing page (you may check here: https://wgms.ch/data\_databaseversions/)  $\rightarrow$  Thank you! Manuscript checked, cited values and references updated

Line 30: Good and interesting that you refer to the WMO(GCOS) headline indicators.  $\rightarrow$  Thank you

Line 32: You could also refer to Gärtner-Roer et al. 2019 (Worldwide assessment of national glacier monitoring and future perspectives, MRD, 39, A1-A11) about the national implementation of the glacier monitoring strategy.  $\rightarrow$  *Reference added in text and reference list. Thank you.*

Line 45: I would just write "glaciers" and not "clean glaciers". If you want to use the term "clean" in this context, you would have to first explain what you exactly mean.

On the other hand, it would open many other questions., e.g. what about glaciers that are partly debris-covered, and how would this affect the glaciological mass balance.  $\rightarrow$  corrected: "clean" deleted

Line 110: There is a kind of break in here, maybe you could link this last paragraph more to the previous Introduction.

Sentence modified: "In this article we focus on the mass balance and glacier length changes of Yala and Rikha Samba glaciers measured within the framework of the HKH-Cryosphere Monitoring Project."

Line 113/114: Not fully clear; what do you mean with "other supporting data beyond the scope of the WGMS FoG database"?

Sentence and "for" added. The sentences read now: "

At Yala Glacier we measured [...]. Additionally, we recorded supporting information such as flow velocity and direction. On Rikha Samba Glacier we assessed [...]. The methods are documented for these measurements and data submitted to the WGMS Fluctuations of Glaciers (FoG) database (WGMS, 2021), and for other supporting data beyond the scope of the WGMS FoG database."

Figure 1 is very useful for the broader context, incl. the overview map showing the location of other glaciers with mass-balance measurements.

You could add the date of the glacier outlines (= the same as the satellite image?).

The dates of the glacier outlines have been added in Fig. 1, besides having it mentioned in the caption.

Line 292: Sentence is logically not fully clear, as the DEM is part of the geodetic method (?). Or is it an additional DEM that you are referring to?

Sentence deleted. The information that the DEM2012 was used for the geodetic mass balance calculation is already provided in subsection "3.2 Maps, satellite images and DEMs". In subsection "3.4.1 Point and glacier-wide mass balance", the DEMs are now explicitly specified in Line 322 and the sentence reads now: "The elevations of the DEM2012 for Yala Glacier and the SRTM1 for Rikha Samba Glacier were applied to the regression equations to calculate the glacier-wide mass balance."

Line 320/21: You could explicitly state that Wagnon et al. used this methodology on Mera and Pokhalde glaciers.

"for Mera and Pokalde glaciers" added. The sentence reads now: "The glacier-wide mass balances, the equilibrium-line altitude (ELA) and accumulation-area ratio (AAR) were calculated based on the interpolated mass-balance gradient derived from the point measurements following a similar method used by Wagnon et al. (2013) for Mera and Pokalde glaciers."

Line 535: there is some repetition in this paragraph, when compared to the beginning of this section.

*The author didn't find any repetitions. However, in Line 558 the time interval we corrected to "2006 to 2011", instead "2011 to 2016".*

Subsection 4.1: 1st paragraph: in situ mass balance both glaciers; 2nd paragraph: seasonal balance Yala Glacier; 3rd paragraph: uncertainties of mass balance; 4th paragraph: densities; 5th paragraph: ELA and AAR; 6th paragraph: mass-balance gradients; 7th paragraph: observations snow; 8th paragraph: geodetic mass balance and cumulative mass balances

Subsection 4.2: 1st paragraph: glacier length changes; 2nd paragraph: glacier flow

Line 604: "Yala Glacier will disappear" → *corrected, "is" deleted*

Line 663: The conclusions of this discussion (about glacier length, flow, and downwasting) remains a bit elusive: Do you mean that there is a bias in the mass-balance measurements? I read in this paragraph that the detailed topographical analyses (length, flow, downwasting) are useful to better assess the mass-balance measurements. But this could be mentioned more explicitly.

**The paragraph has been revised and reads now:**

"From 2011 onwards, we observed that concave shapes on Yala Glacier's surface have become more pronounced, and that the glacier surface was downwasting, as observed at other glaciers (Ragettli et al., 2016; Sommer et al. 2020). Both the downwasting and enhanced concave shapes are a consequence of the decreased ice velocities, and indicate changes in the glacier dynamics. The downwasting of Yala Glacier can affect the mass balance and its monitoring in several ways, such as locally enhanced ablation and compromised representativeness of stake measurements. Ablation can be locally enhanced in bowl-shaped areas, where radiation is reflected, resulting in a positive feedback and higher ablation than in the surrounding area (Hock, 2005). Such concave surfaces with transitions to steep slopes became more pronounced, for example, between stakes S1 and S1B and near S5. Usually, stakes represent a characteristic type of glacier area. However, the representativeness of stake measurements is compromised over time when the glacier surface topography changes from an even surface to a very concave surface with steep slopes. The bias induced by reduced stake representativeness should be corrected later with help of complementing geodetic mass-balance analyses for the same timeframe (Zemp et a., 2013)."

Line 695: rather "Fujita and Nuimura (2011) found that..."  $\rightarrow$  corrected

Line 746: or: debris-free  $\rightarrow$  corrected, "clean" replaced with "debris-free"

Line 817: maybe reformulate this sub-title; the in situ measurements will always remain the same, but the mass balance will be extrapolated to regions without in situ measurements.

**The revised title of the subsection is now: "5.6 Extrapolation of mass-balance data to unmeasured areas"**

Line 854: Not fully clear: I understood that the glacier-wide mass balance is given for the entire glacier as shown in Figure 1. Hence, there is a quantification for these slopes (as long as they are in the respective glacier area/outline?), but this extrapolation (which nevertheless represents a quantification) may have a certain error. It is also a bit confusing because line 855 says that this error (underestimation of ablation) can be addressed with the geodetic method, which was actually done in the presented study. Or was the applied DEM not sufficient enough (as the following sentence might suggest?).

The paragraph has been rephrased and now reads: "The mass balance of the steep south-west-facing slopes on Yala Glacier could not be measured but have been quantified based on the linear regression equations from the in situ measurements. However, the ablation on steep slopes is possibly underestimated due to the orientation and the steepness of the slopes. This bias can be addressed with geodetic mass-balance analyses using the same time period as for the in situ measurements. The relevance of the steep glacier slopes in terms of area cannot be quantified neither for Yala Glacier, nor the glaciers in Nepal in general with DEMs of 30 m and 90 m resolution, respectively."

Please note, it is a standard to complement glaciological with geodetic mass-balance analyses. It is essential to use data from the same time period (e.g. Zemp et al., 2013 and Wagnon et al., 2020). In our study, the measurement period for the geodetic (2000–2012) and glaciological data (2011–2017) differs, as documented in section "3 Data and Methods", and cannot be used to complement each other.

Also, in the time interval from 2000 to 2012, the annual mass balance on every point on the glacier would have changed every year. From one year to the next, the mass change moved continuously further down on the glacier. Hence, the average annual mass-balance distribution from 2000 to 2012

*in the areas of the ice cliffs and steep slopes does not represent the annual mass-balance distribution for later years. Additionally, the uncertainty is relatively large.*

Geodetic mass-balance analyses can also be used for several applications, taking the purpose, base data (e.g. resolution, time interval) and uncertainties into consideration. One example we give in the discussion section, subsection "5.1.4 Steep slopes and ice cliffs": "To better understand and assess specifically the influence of the steep slopes and ice cliffs of the mass balance, geodetic thickness-change analyses based on **high-resolution surface elevations for short time intervals** could be used, in combination with energy-balance models (Joerg and Zemp, 2014)." Besides this, we write: "Complementing geodetic mass-balance measurements for the same timeframe help to correct the glacier-wide annual mass balances of Yala Glacier for biases such as introduced by steep slopes and ice cliffs (Zemp et al., 2013; Wagnon et al., 2020)."

Regarding the glaciological and geodetic mass-balance measurements it is important to consider what is actually measured, the role of the measured time interval, ice flow and resolution of DEMs used (see also Cogley et al., 2011). In the Table below a brief overview is provided for the different methods.

|                                                | Glaciological mass balance                                                                                                                                                                                                                                                                                                         | Geodetic mass balance                                                                                                                                                          |                                                                                                                                                                                                |  |  |
|------------------------------------------------|------------------------------------------------------------------------------------------------------------------------------------------------------------------------------------------------------------------------------------------------------------------------------------------------------------------------------------|--------------------------------------------------------------------------------------------------------------------------------------------------------------------------------|------------------------------------------------------------------------------------------------------------------------------------------------------------------------------------------------|--|--|
|                                                |                                                                                                                                                                                                                                                                                                                                    | high resolution                                                                                                                                                                | low resolution                                                                                                                                                                                 |  |  |
| Typical time         Annual, seasonal, monthly |                                                                                                                                                                                                                                                                                                                                    | Multiannual measurements                                                                                                                                                       |                                                                                                                                                                                                |  |  |
| interval                                       |                                                                                                                                                                                                                                                                                                                                    | Approx. 1–5 years                                                                                                                                                              | Approx. 10 years                                                                                                                                                                               |  |  |
| What is
measured:                           | Mass changes between the
glacier surface $s_0$ and $s_1$ , in the
time period between $t_0$ and $t_1$ , in
a relative reference system.
The measurements are taken in
reference to the glacier surface
and the used measurement
method, e.g. a stake or a snow pit
relative to the glacier surface. | Mass changes between the
the time period between to
reference system.
The elevations of the glacio
times are measured in an o                                      | e glacier surface $s_0$ and $s_1$ , in
$a$ and $t_1$ , in an absolute
er surfaces at two different
absolute coordinate system.                                                 |  |  |
| Mass balance:                                  | Only surface mass balance                                                                                                                                                                                                                                                                                                          | Surface, internal and basa                                                                                                                                                     | l mass balance                                                                                                                                                                                 |  |  |
| Ice flow:                                      | Ignored                                                                                                                                                                                                                                                                                                                            | Relevant but small                                                                                                                                                             | Relevant                                                                                                                                                                                       |  |  |
| Mass balance
distribution
on glacier:    | On high elevations on glacier:
positive balance (or less negative
than on low elevations)
On low elevations on glacier:
Negative balance (or less positive
than on high elevations)                                                                                                                                 | Generally, if time interval
is short enough and no
extreme events:
On high elevations on
glacier: positive balance
(or less negative than on
low elevations) | Generally:
Depends on past
climate/surface mass
balance and ice flow.
Anywhere on the glacier it
can be positive or
negative.                                                |  |  |
|                                                |                                                                                                                                                                                                                                                                                                                                    | On low elevations on
glacier: Negative
balance (or less positive
than on low elevations)                                                                              | Generally, over very long
time intervals, shrinking
glaciers have a very
negative mass balance in
the low elevations of a
glacier and less negative
mass balances higher up. |  |  |

Line 872: "on the following points"  $\rightarrow$  corrected, "the" added

Line 1205: Wagnon et al. 2020 is listed twice (it must be an important paper :-) )

**Corrected. We confirm, it's important. =)**

Line 1220: The WGMS (2020a) reference might be replaced with WGMS (2021), because the 2021 database version also includes all data of the 2020 version, and hence you can just refer to the latest

database version. (Unless you explicitly want to refer to some data that were in the 2020 database but are not any more or that have changed in the 2021 version; but I think, this (rare) case does not apply here). WGMS (2020b) would then be WGMS (2020).

*New FoG database 2021-05 checked, reference and values (Table 6) updated.*

**Further point that I have noticed:**

Usage of three nouns: for mass-balance rate or mass-balance monitoring etc., they are usually written with hyphen (not quite with all occurrences in the text), but other word combinations are written without (e.g. sea level rise).

We checked the manuscript again regarding the hyphenation. Please note, the rules of hyphenation of compound nouns vary. Here, we followed the ESSD grammar guidelines, terminology used by Cogley et al., 2011 (Glossary of mass balance and related terms), common rules found online (e.g. by Cambridge Dictionary, Capstone Editing), and tried to follow the hyphenation rules referee #2 used in earlier comments. We do not have a preference regarding the usage of hyphens.

**Responses to comments of Report #2, by Argha Banerjee, referee #1 submitted on 21 May 2021**

I congratulate the authors for an excellent revision. In my opinion the manuscript is at a stage where it can be published up to a few minor (but significant) corrections.

These are,

1. It may not be wise to report the winter mass balance data for 2012 and 2015 where there are large uncertainties such that the mass conservation principal was violated. Please see my detailed comments in the annotated manuscript.

In the section results, subsection "4.1 Mass balances, ELA, AAR and gradients", a remark has been added in Table 2 and its caption, as well as in caption of Fig. 4. Additionally, a statement has been added in the result section "4.1 Mass balances, ELA, AAR and gradients" and discussion section "5.1.2 Seasonal mass balance".

Table 2: Mass balance (B) measured with the glaciological method, winter balance ( $B_W$ ), summer balance ( $B_S$ ), ELA, AAR and mass-balance gradient for Yala Glacier from 2011/12 to 2016/17. The summer balance from 2011/12 and winter balance from 2014/15 (\*) have not been reported to the WGMS and are discussed in subsection 5.1.2 Seasonal mass balance.

|           | В                | Bw                | Bs                | Bw+Bs    | ELA           |      | db/dz                                   |
|-----------|------------------|-------------------|-------------------|----------|---------------|------|-----------------------------------------|
| B year    | (m w.e.)         | ( m w.e. ) | ( m w.e. ) | (m w.e.) | (m a.s.l.)    | AAR  | $(m \text{ w.e.} (100 \text{ m})^{-1})$ |
| 2011/12   | $-0.86 \pm 0.40$ | 0.16              | -0.20*            | -0.03    | $5454 \pm 30$ | 0.28 | 1.14                                    |
| 2012/13   | $-0.01 \pm 0.29$ | 0.36              | -0.35             | 0.01     | $5380 \pm 20$ | 0.48 | 0.99                                    |
| 2013/14   | $-0.61 \pm 0.27$ | 0.27              | -0.99             | -0.73    | 5431 ±20      | 0.35 | 1.18                                    |
| 2014/15   | $-1.18 \pm 0.26$ | 0.54*             | -1.12             | -0.59    | $5510 \pm 40$ | 0.13 | 0.90                                    |
| 2015/16   | $-0.61 \pm 0.23$ | 0.19              | -0.79             | -0.60    | $5444 \pm 20$ | 0.31 | 0.93                                    |
| 2016/17   | -1.54 ±0.20      | 0.20              | -1.75             | -1.54    | 5518 ±20      | 0.12 | 1.10                                    |
| Mean      | $-0.80 \pm 0.28$ | 0.29              | -0.87             | -0.58    | 5456          | 0.28 | 1.04                                    |
| STD       | 0.53             | 0.14              | 0.56              | 0.56     | 52            | 0.14 | 0.12                                    |
| 2011-2017 | -4.80 ±0.69      | 1.72              | -5.21             | -3.48    |               |      |                                         |

The caption of Fig. 4 now reads: "Winter, summer and annual mass balance of Yala Glacier and annual balance of Rikha Samba Glacier, calculated based on the respective gradients. In the mass-balance years 2011/12 and 2014/15, the sum of winter and summer balances differ significantly from the annual balances, likely due to a lack of data in higher elevations. The summer balance from 2011/12 and winter balance from 2014/15 have not been reported to the WGMS."

In the section results, subsection "4.1 Mass balances, ELA, AAR and gradients" the underlined sentence has been added: "The cumulated winter and summer balances largely sum up to the annual balances, except in 2011/12 and 2014/15 when the cumulated winter and summer balances underestimate the annual mass loss by -0.83 m and -0.59 m w.e. These discrepancies are discussed in section 5.1.2 Seasonal mass balance."

In the section discussion, subsection "5.1.2 Seasonal mass balance", the two underlined sentences have been added: "[...] The seasonal mass-balance measurements in June 2015 were taken under precarious conditions, and only stake measurements could be taken up to an elevation of 5217 m a.s.l., resulting in a higher uncertainty for the seasonal mass balances in 2014/15 and a possibly underrepresented accumulation in winter 2014/15. Hence, the winter balance for the mass-balance year 2014/15 has not been reported to the WGMS. [...] In autumn 2012, we calculated the least negative summer balance (-0.35 m w.e.), based on only three measurements and likely underestimating ablation. This could explain the underestimated annual mass loss of -0.83 m w.e. in the cumulative seasonal balance

compared to the annual balance of 2011/12. Consequently, the summer balance from the mass- balance year 2011/12 has not been reported to the WGMS."

Please also note comments and responses below for:

- Page 15, comments 1, 2
- Page 23, comment and responses

2. Your conclusion about the annual balance on Yala being insensitive to winter balance can be discussed in a bit more detail. Please see my detailed comment in the attached annotated manuscript. Basically you have concluded this based on 4 years of data (if we leave out the two years with large uncertainties) where annual balance stayed mostly negative. The conclusion may not hold in general.

Please note, it is characteristic for summer-accumulation-type glaciers that the biggest mass changes happen in summer. In section 2 we write: "Both glaciers are summer-accumulation-type glaciers (Ageta and Higuchi, 1984), which are characterized by an overlapping main accumulation and ablation season during the monsoon season (Fig. S1). A brief description of summer-accumulation-type glaciers and mass-balance measurements is provided in the Supplement (section S1)."

During the investigation period, the winter balance was positive and the summer balance was negati

---

## Author Response (AR3)

**Authors' response to comments**

*We would like to thank the topical editor Reinhard Drews for accepting the manuscript for publication, his kind words and for the comments for further improvement of the article. We responded to all comments raised point by point in blue italic font below.*

**Topical Editor Decision: Publish subject to technical corrections** (28 Jun 2021) by Reinhard Drews
Comments to the Author:
Dear Authors,

Thank you for your revisions in response to the re-review. Both reviewers have positively evaluated this version of the paper. I agree with their evaluation and accept the paper for ESSD. Congratulations. I complement you for a detailed response-to-review which treated the reviewer's comments with much respect.

In my re-read of the paper I found some smaller issues that require attention, but most those can be hopefully dealt with straightforwardly.

Kind regards & congratulations,
-- Reinhard Drews

l 42. "misinformation" is pretty strong wording referring to IPPC AR4. Maybe "overstating" ? Also, what's the point of citing an old IPCC report, what happened to this statement in the fifth report? I think this sentenced can be removed without much loss of information. If you insist on maintaining it include the latest IPPC report.
*There was an issue in the IPCC AR4 regarding a statement about glaciers in the Himalayas and a small team looked into it in 2009 (discussions on cryolist in 2009). As a result, Cogley et al. (2010) tracked down the source of the issue and used the word "misinformation" to address it, hence, it seems a legitimate word to be used.*
*Following the discussions two aspects became clear: (1) this specific information in the IPCC AR4 didn't satisfy the requirements of science, and (2) there was not sufficient information available regarding the actual state of the glaciers in this region. A substantial amount of funding became available to address this knowledge gap. In this regard, the AR4 took an important role in the Himalayas compared to other IPCC reports. Newly gained and more accurate knowledge is integrated in later IPCC reports but it was AR4 that boosted glacier research and capacity building in the HKH region, which is the reason for referring to it in the following sentence. Both sentences have been kept as they were.*

*Cogley, J. G., Kargel, J. S., Kaser, G., and van der Veen, C. J.: Tracking the source of glacier misinformation, Science, 327(5965), 522, https://doi.org/10.1126/science.327.5965.522-a, 2010.*

Fig 1b include scalebar in legend.
*Corrected, scalebar shifted.*

l. 235 I don't quite understand how snow compaction can "push" a stake up.
*We write "…the underlying snow and firn layers compact over time and [the layers] may push or pull the stake up or down." Hence, it is the snow and firn layers that cause the stake to move. The compaction happens in the snow and firn layers. However, the stake cannot be compacted with the snow and firn, and is usually somehow frozen into firn or ice layers. Consequently, the stakes are pushed or pulled up or down, depending on the forces, where the compaction happens, where the stake is frozen into and where space is available.*

l. 262 remove "so-called" (also elsewhere. I am never quite sure what this adds.)
*Based on the comment of a reviewer, "so-called" has been added in L262, and removed again now. In L310, "so-called" deleted, and in L688, "so-called preferable ELAs" kept.*

l. 383 If GPS data from 2012 are available, why weren't they used as GCPs in Section 3.3?
*We generated the DEM2012 using GCPs with help of T. Bolch and colleagues in February 2014. Up to that time we had only very limited access to dGNSS devices, and limited time to conduct dGNSS surveys besides the mass balance measurements and trainings of about 25 participants. In May 2012, we conducted various dGNSS surveys (Table S2), but no GCP survey that should have been conducted on stable non-glaciated terrain.*

l. 423 SRTM 3 has an accuracy of plus minus 16 m (l. 274) and was used as GCP to offset the stereographic DEM. I am not sure how this uncertainty can then melt down to 7.4 m.
*We corrected the sentences: "To assess the uncertainty of the thickness change, we estimated the vertical precision of the DEMs by calculating the normalized median absolute deviation (NMAD), which is ±7.41 m (Holzer et al., 2015; Höhle and Höhle, 2009). The uncertainty of the geodetic mass balance is the root of the sum of each squared error term, which consist of the NMAD and the uncertainty for the ice density of ±60 kg m$^{-3}$ (Huss, 2013)."*

*The accuracy of ±16 m is the vertical accuracy of the elevation of the SRTM3 (absolute system). The uncertainty of ±7.4 m is the uncertainty of the thickness change of the two DEMs SRTM3 and DEM2012 (relative system). To generate the DEM2012 (slave DEM), we used the SRTM3 as master DEM, and for that we used the elevation of the SRTM3 for the GCPs.*

*Reference added: Höhle, J. and Höhle, M.: Accuracy assessment of digital elevation models by means of robust statistical methods. ISPRS J. Photogramm. Remote Sens., 64, 398–406, https://doi.org/10.1016/j.isprsjprs.2009.02.003, 2009.*

Fig 2 Label panels with (a)-(d). Replace this with description "far left".
*Corrected, "far left" replaced with (a), headings of panels deleted, caption edited.*

[Figure]

**Figure 2: The glacier hypsography (a), and the m**ass balances and gradients for the annual, winter and summer mass balance for Yala Glacier from 2011–2017 **(b-d).**

Fig 4/13 add x-axis label "years"
*Corrected for both Figures 4 and 13.*
*Fig 10: "years" added and y-axis labelled with "Cumulative frontal variation (m)"*

Fig 7 I could be wrong but I suspect that at least some of this thickness change is due to the pm 16 m of SRTM uncertainty.

*Yes, the uncertainties of the DEMs influence the thickness change, which is reflected in the uncertainty of the glacier wide thickness change of ±7.41 m, calculated with the NMAD. Please also note response for L 423.*

Fig7 I suggest to divide the thickness difference by the 8 year time period to get "per year" thickness change which is a quantity more familiar for most.

*It is common to either display thickness differences as total value or annual rate. We prefer to display the information as total value. Examples of articles that display the total thickness change:*

- *Goerlich, F., Bolch, T., and Paul, F.: More dynamic than expected: an updated survey of surging glaciers in the Pamir, Earth Syst. Sci. Data, 12, 3161–3176, https://doi.org/10.5194/essd-12-3161-2020, 2020.*
- *Liu et al.: Recent Accelerating Glacier Mass Loss of the Geladandong Mountain, Inner Tibetan Plateau, Estimated from ZiYuan-3 and TanDEM-X Measurements, Remote Sens. 2020, 12, 472. https://doi.org/10.3390/rs12030472, 2020.*
- *Bolch, T., Pieczonka, T., Mukherjee, K., and Shea, J.: Brief communication: Glaciers in the Hunza catchment (Karakoram) have been nearly in balance since the 1970s, The Cryosphere, 11, 531–539, https://doi.org/10.5194/tc-11-531-2017, 2017.*
- *Holzer, N., Vijay, S., Yao, T., Xu, B., Buchroithner, M., and Bolch, T.: Four decades of glacier variations at Muztagh Ata (eastern Pamir): a multi-sensor study including Hexagon KH-9 and Pléiades data, The Cryosphere, 9, 2071–2088, https://doi.org/10.5194/tc-9-2071-2015, 2015.*
- *Pieczonka et al. (2013) Heterogeneous mass loss of glacier in the Aksu-Tarim Catchment revealed by 1976 KH-9 Hexagon and 2009 SPOR-5 stereo imagery, Remote Sensing of Environment 130, 233–244, http://dx.doi.org/10.1016/j.rse.2012.11.020, 2013.*

L 540 there is a "per year" missing after "-3.8 m"
*Corrected in L 540 and L 741*

Fig S1 insert "months of the year" as x-label
*Corrected*

S2 define if you refer to horizontal or vertical dGNSS accuracy.
*Corrected: "Horizontal accuracy" instead "Accuracy measurements"*

***Other changes***
*L 425: "were" instead of "are"*
*L 824: value of the geodetic mass balance corrected (value was by accident from thickness change):*
*"…geodetic mass balance of Yala Glacier showed a mass loss of -8.92 ±6.33 m w. e. …"*